# Cytoarchitectonic, receptor distribution and functional connectivity analyses of the macaque frontal lobe

Lucija Rapan[1]*, Sean Froudist-Walsh[2,3], Meiqi Niu[1], Ting Xu[4], Ling Zhao[1], Thomas Funck[1], Xiao-Jing Wang[2], Katrin Amunts[1,5], Nicola Palomero-Gallagher[1,5]

[1]Institute of Neuroscience and Medicine INM-1, Research Centre Jülich, Jülich, Germany; [2]Center for Neural Science, New York University, New York, United States; [3]Bristol Computational Neuroscience Unit, Faculty of Engineering, University of Bristol, Bristol, United Kingdom; [4]Center for the Developing Brain, Child Mind Institute, New York, United States; [5]C. & O. Vogt Institute for Brain Research, Heinrich-Heine-University, Düsseldorf, Germany

**Abstract** Based on quantitative cyto- and receptor architectonic analyses, we identified 35 prefrontal areas, including novel subdivisions of Walker's areas 10, 9, 8B, and 46. Statistical analysis of receptor densities revealed regional differences in lateral and ventrolateral prefrontal cortex. Indeed, structural and functional organization of subdivisions encompassing areas 46 and 12 demonstrated significant differences in the interareal levels of $\alpha_2$ receptors. Furthermore, multivariate analysis included receptor fingerprints of previously identified 16 motor areas in the same macaque brains and revealed 5 clusters encompassing frontal lobe areas. We used the MRI datasets from the non-human primate data sharing consortium PRIME-DE to perform functional connectivity analyses using the resulting frontal maps as seed regions. In general, rostrally located frontal areas were characterized by bigger fingerprints, that is, higher receptor densities, and stronger regional interconnections. Whereas more caudal areas had smaller fingerprints, but showed a widespread connectivity pattern with distant cortical regions. Taken together, this study provides a comprehensive insight into the molecular structure underlying the functional organization of the cortex and, thus, reconcile the discrepancies between the structural and functional hierarchical organization of the primate frontal lobe. Finally, our data are publicly available via the EBRAINS and BALSA repositories for the entire scientific community.

*For correspondence:
l.jankovic-rapan@fz-juelich.de

## Editor's evaluation

Rapan et al. report a new multi-modal parcellation of the macaque frontal cortex based on cytoarchitectural division complemented with functional connectivity and neurochemical data. This builds on prior highly influential maps that subdivide the cortex based on anatomical fingerprints, both confirming these prior reports and defining new subdivisions. As such, this is a fundamental contribution with compelling results that can guide future neuroscientific research into the function of the frontal lobes.

## Introduction

The anterior portion of the primate frontal lobe, known as the prefrontal cortex (PFC), is a region notably involved in the higher cognitive functions (*Fuster, 2008*). It has been a focus region of numerous functional studies in human and monkey brains. Research involving non-human primates

**Figure 1.** Schematic drawing of the medial, lateral, and orbital surfaces of the macaque prefrontal cortex depicting parcellations according to (**A**) *Walker, 1940*, and (**B**) *Carmichael and Price, 1994*. Macroanatomical landmarks are marked with red dashed lines; *cgs,* cingulate sulcus; *ias,* inferior arcuate sulcus; *ps,* principal sulcus; *ros,* rostral orbital sulcus; *sas,* superior arcuate sulcus.

plays a vital role in the medical progress and scientific applications due to their close evolutionary relation to humans, but also due to ethical standards which do not allow all the vital material and data to be acquired directly from human brains (*DeFelipe, 2015*). In particular, macaque monkeys are the most widely used primate species in neurobiological research (*Passingham, 2009*). As a series of comparative analyses have shown, they share a similar basic architectonic plan to that of the human brain (*Petrides et al., 2012*; *Petrides and Pandya, 1994*; *Petrides and Pandya, 1999*; *Petrides and Pandya, 2002*; *Petrides and Pandya, 2009*).

Early cytoarchitectonic studies of the monkey cerebral cortex encountered the same issues and limitations as those of the human cortex with regard to both methodological and nomenclatural issues. Methodological limitations include small sample size, usually single of only a few cases, analysis of a single modality, and a subjective approach to the detection of cortical borders due to their identification by pure visual inspection. The nomenclature issue seems to be problematic as well since it not only affects comparability between different maps, but also translational analyses and identification of homolog areas in the human brain. The most influential cytoarchitectonic map of the monkey PFC was published by *Walker, 1940*, who used the numerical nomenclature introduced by Brodmann in his human brain map (*Brodmann, 1909*), although he did not compare the cytoarchitecture of the human and macaque monkey prefrontal regions in detail. *Walker, 1940* labelled the frontopolar cortex of the monkey as area 10 and added areas 46 and 45 (*Figure 1*), which were not indicated in Brodmann's map of the monkey frontal cortex (*Brodmann, 1905*). Thus, Walker's (*Walker, 1940*) parcellation scheme became the basis for future microparcellation and anatomical–connectional studies with anterograde and retrograde tracers, as well as in physiological microstimulation studies (e.g. *Barbas and Pandya, 1989*; *Carmichael and Price, 1996*; *Morecraft et al., 2012*; *Petrides and Pandya, 2006*). This research led to a 'golden era' of experimental neuroanatomy with various research groups focused on the analysis of a specific region of interest (ROI) in the monkey brain, for example, the orbitofrontal (*Barbas, 2007*; *Carmichael and Price, 1994*), dorsolateral prefrontal (*Petrides, 2005*; *Petrides and Pandya, 1999*; *Preuss and Goldman-Rakic, 1991*), and ventrolateral PFC (*Gerbella et al., 2007*; *Petrides and Pandya, 2002*; *Preuss and Goldman-Rakic, 1991*).

The development of a quantitative approach to the analysis of cytoarchitecture in the entire human brain sections enabled statistical validation of visually detectable cortical borders and thus an objective approach to brain mapping (*Schleicher et al., 2009*; *Schleicher and Zilles, 1990*). Furthermore, an implementation of the analyses, which include multiple architectonical modalities, also enabled

a more comprehensive characterization of the cortical parcellation. Specifically, quantitative in vitro multireceptor autoradiography has been revealed as a powerful tool to describe the important aspects of the brain's molecular and functional organization since neurotransmitters and their receptors are known to play an important role in a signalling process (*Impieri et al., 2019*; *Palomero-Gallagher et al., 2009*; *Zilles et al., 2002*). Concentrations of receptors for classical neurotransmitter systems vary between different cortical areas; hence, the area-specific balance of different receptor types ('receptor fingerprint') subserves its distinct functional properties. Quantification of heterogeneous receptors distribution throughout the cerebral cortex enables the identification and characterization of principal subdivisions such as primary sensory, primary motor, and hierarchically higher sensory or multimodal areas (*Palomero-Gallagher and Zilles, 2019*; *Zilles and Palomero-Gallagher, 2017b*). Multivariate analyses of the receptor fingerprints demonstrate not only structural but also functionally significant clustering of cortical areas (*Zilles and Amunts, 2009*). Therefore, this multimodal approach to cortical mapping provides detailed insights into the relationship between cytoarchitecture (which highlights the microstructural heterogeneity) and neurotransmitter receptor distributions (which emphasize the molecular aspects of signal processing) in the healthy non-human primate brain. It constitutes an objective and reliable tool which provides basic information of functional networks and precisely defined anatomical structures.

In vivo neuroimaging of the non-human primates has been advancing rapidly due to increased collaboration and data sharing (*Milham et al., 2018*; *Milham et al., 2020*). Primate imaging is a promising approach to link between precise electrophysiological and neuroanatomical studies of the cortex and distinct functional networks observed in humans. However, integration of neuroimaging data with high-quality postmortem anatomical data has been problematic since these results have not been conveyed in a common coordinate space. In recent years, several digital macaque atlases have been created (*Bezgin et al., 2012*; *Frey et al., 2011*; *McLaren et al., 2009*; *Moirano et al., 2019*; *Reveley et al., 2017*; *Van Essen et al., 2012*) based on the previous parcellations. Indeed, maps of *Carmichael and Price, 1994*; *Petrides and Pandya, 2002*; *Petrides, 2005* and *Preuss and Goldman-Rakic, 1991*, used in atlas of *Saleem and Logothetis, 2012*, have been brought into stereotaxic space by *Reveley et al., 2017*. However, macaque maps, which are currently available to the in vivo neuro-imaging researchers, do not contain information about receptor densities. Such information enables identification of the chemical underpinnings of functional activity and connectivity observed in vivo.

The primary aim of this study was to identify and characterize prefrontal areas based a quantitative cyto- and receptor architectonic approach, and to create a 3D statistically validated parcellation scheme in stereotaxic space. Since the functional connectivity analysis revealed a tight coupling between posterior prefrontal and premotor areas, and, also the fact that receptors play a key role in signal transduction, we hypothesized that this tight relationship would be associated with similarities in neurochemical composition. Thus, we decided to also include our previously published receptor fingerprints of (pre)motor areas (*Rapan et al., 2021*) in the multivariate analyses. Importantly, the densities of prefrontal and (pre)motor areas were all obtained from the same brains. All data are made available to the community in standard Yerkes19 surface via the EBRAINS repository of the Human Brain Project and the BALSA platform.

## Results

### Cytoarchitectonic analysis

The systematic identification of 35 prefrontal areas of every 20th coronal histological section of the brain DP1, as well as silver body-stained sections of brains 11530, 11539, 11543, resulted in a map containing the location and extent of all areas, and their relationships with macroanatomical landmarks is clearly depicted in *Figure 2*. Additionally, *Table 1* was created to depict the relationship between areas defined by Rapan and colleagues (this study; *Rapan et al., 2021*) and referenced maps used here.

Additionally, *Figure 2—figure supplements 1 and 2* show the characteristic macroanatomical features (i.e. dimples and sulci) of the macaque frontal lobe, used here to delineate our ROIs. The PFC is separated from the motor areas by the well-defined arcuate sulcus (*arcs*), which branches dorsally into the superior arcuate sulcus (*sas*) and ventrally into the inferior arcuate sulcus (*ias*), thus forming a letter Y on the lateral surface of the hemisphere. Ventrally, PFC is limited by the lateral fissure (*lf*),

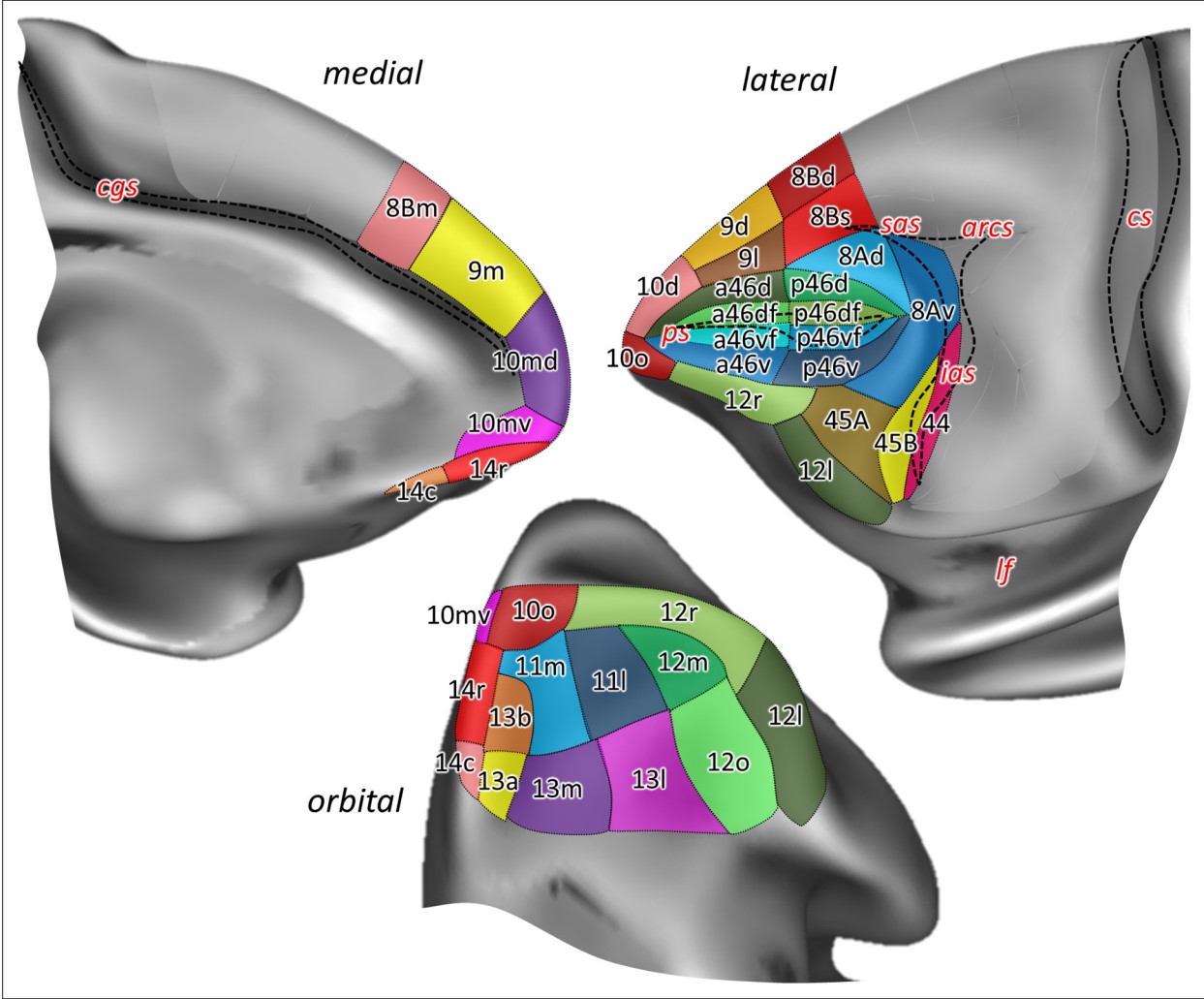

**Figure 2.** Position and extent of the prefrontal areas on the medial, lateral, and orbital views of the Yerkes19 surface. The files with the parcellation scheme are available via EBRAINS platform of the Human Brain Project (https://search.kg.ebrains.eu/instances/Project/e39a0407-a98a-480e-9c63-4a2225ddfbe4) and the BALSA neuroimaging site (https://balsa.wustl.edu/study/7xGrm). Macroanatomical landmarks are marked in red letters, while black dashed lines mark fundus of sulci. arcs, spur of the arcuate sulcus; cgs, cingulate sulcus; cs, central sulcus; ias, inferior arcuate sulcus; lf, lateral fissure; ps, principal sulcus; sas, superior arcuate sulcus.

The online version of this article includes the following figure supplement(s) for figure 2:

**Figure supplement 1.** Macroanatomical landmarks (sulci labelled in red letters and dimples in green) shown on the lateral surface of the two related species of macaque monkey used in the present architectonic analyses.

**Figure supplement 2.** 2D flat map, based on the macroanatomical landmarks of every 40th section, displays orbital, medial, and dorsolateral hemispheric views with all defined areas within the macaque frontal lobe.

which represents the border with temporal areas, whereas on the medial surface, the cingulate sulcus (*cgs*) separates PFC from the limbic cortex. Another prominent feature on the lateral aspect of the PFC in the macaque monkey brain is the well-defined principal sulcus (*ps*), which starts rostrally within the frontopolar region and ends caudally within the arcuate convexity (*Figure 2—figure supplement 2*). These prominent macroanatomical features are recognizable in both macaque species (*Macaca mulatta* – brain ID DP1, and *Macaca fascicularis* – brain IDs rh11530, rh11539, and rh11543) studied here, as well as on the Yerkes19 surface used as a template for our 3D map (*Figure 2—figure supplement 1*).

In contrast, the orbitofrontal surface is characterized by a more variable sulcal pattern, comprised of lateral (*lorb*) and medial orbital sulcus (*morb*). In brain DP1 they are shown as two parallel, sagittally oriented sulci in the left hemisphere, while in the right hemisphere these sulci are partially connected

**Table 1.** A list of cortical areas identified by the different authors (***Walker, 1940***; ***Petrides and Pandya, 1994***; ***Petrides and Pandya, 2002***; ***Preuss and Goldman-Rakic, 1991***; ***Morecraft et al., 2012***; ***Caminiti et al., 2017***), whose maps were used as references for the present analysis, compared to areas identified by Rapan and colleagues.

'*a46*', areas a46d, a46df, a46vf, a46v; '*p46*', areas p46d, p46df, p46vf, p46v; '*p46d*', areas p46d, p46df; '*p46v*', areas p46v, p46vf.

| Walker vs. Rapan | | Preuss & Goldman-Rakic vs. Rapan | | Carmichael & Price vs. Rapan | |
|---|---|---|---|---|---|
| | 10d | | 10d | | 10d |
| | 10md | | 10md | | 10md |
| | 10mv | | 10mv | **10**m | 10mv |
| **10** | 10o | **10** | 10o | **10o** | 10o |
| | Rostral part of 'a46', 11m, 14r, 13b | | Rostral part of a46d and a46v | | |
| | 9d | | 9d | | |
| | 9l | **9d** | 9l | | |
| **9** | 9m | **9m** | 9m | *n.a.* | |
| **8B** | 8Bd | | 8Bd | | |
| | 8Bs | **8Bd** | 8Bs | | |
| | 8Bm | **8Bm** | 8Bm | | |
| | Caudal part of 9d, 9l, and 9m | | Caudal part of 9d, 9l, and 9m | *n.a.* | |
| **8A** | 8Ad | **8Ar** | 8Ad, 8Av, 45A, caudal part of 'p46' | | |
| | 8Av | **8Am** | 8Ad | | |
| | Caudal part of 'p46' | **8Ac** | 8Av | *n.a.* | |
| | a46' | **46r** | a46df, a46vf | | |
| | p46' | **46dr** | a46d, p46d, ventral part of 9l | | |
| | Dorsal part of 12r; ventral part of 9l | **46vr** | a46v, p46v, dorsal part of 12r | | |
| | Rostroventral part of 8Ad; rostrodorsal part of 45A | **46d** | a46df, p46df | | |
| **46** | | **46v** | a46vf, p46vf | *n.a.* | |
| | 45A | **45** | 45B, 44 | | |
| | 45B | | | | |
| **45** | Rostroventral part of 8Av | | | *n.a.* | |
| *n.a.* | | *n.a.* | | *n.a.* | |
| | 12r | | 12r | **12r** | 12r |
| | 12m | | 12l | **12m** | 12m, 12o |
| | 12l | **12vl** | Rostral part of 45A | **12l** | 12l |
| **12** | 12o | | | **12o** | 12o |
| | Part of 45A; 13l | | | | |
| | 13m | **13M** | 13m | **13b** | 13b |
| **13** | 13l | **13L** | 13l | **13a** | 13a |
| | | | | **13m** | 13m |
| | | | | **13l** | 13l |

*Table 1 continued on next page*

*Table 1 continued*

| Walker vs. Rapan | | Preuss & Goldman-Rakic vs. Rapan | | Carmichael & Price vs. Rapan | |
|---|---|---|---|---|---|
| | 11m | | 11m | **11**m | 11m |
| 11 | 11l | 11 | 11l | **11**l | 11l |
| | Part of 12m, ventral part of 12l | | | | |
| | 14r | **14**A | 14r, 10o, 10mv, 11m, 13b | **14**r | 14r |
| 14 | 14c | **14**M | 14r, 14c | **14**c | 14c |
| | Part of 11m; 13b, 13a | **14**L | 14r, 14c, 13b, 13a | | |

| Petrides & Pandya vs. Rapan | | Morecraft vs. Rapan | | Caminiti vs. Rapan | |
|---|---|---|---|---|---|
| | 10d | | 10d | | 10d |
| | 10md | | 10md | | 10md |
| | 10mv | | 10mv | | 10mv |
| 10 | 10o | 10 | 10o | 10 | 10o |
| | Rostral part of a46d and a46v; ventral part of 12r | | Rostral part of a46d and a46v | | Rostral part of a46d and a46v |
| | 9d | | 9d | | 9d |
| | 9l | **9** | 9l | **9**l | 9l |
| 9 | 9m | **9**m | 9m | **9**m | 9m |
| | 8Bd | | 8Bd | | 8Bd |
| | 8Bs | **8Bd** | 8Bs | | 8Bs |
| **8B** | 8Bm | **8Bm** | 8Bm | **8B** | 8Bm |
| | Caudal part of 9d, 9l, and 9m | | Caudal part of 9d, 9l, and 9m | | Caudal part of 9d, 9l, and 9m |
| **8Ad** | 8Ad | **8Ad** | 8Ad | **8Ad** | 8Ad |
| **8Av** | 8Av | **8Av** | 8Av | **8Av** | 8Av |
| | Caudal part of 'p46' | | Caudal part of 'p46' | | Caudal part of 'p46' |
| **46** | a46' | **46** | a46' | **46dr** | a46d, a46df |
| **9/46d** | p46d' | **9/46d** | p46d' | **46vr** | a46v, a46vf |
| **9/46v** | p46v' | **9/46v** | p46v' | **46dc** | Caudal part of a46d and a46df, 'p46d' |
| | | | | **r46vc** | Caudal part of 'a46v', rostral part of 'p46v' |
| | | | | **c46vc** | p46v, p46vf |
| **45A** | 45A | **45** | 45A | **45A** | 45A |
| **45B** | 45B | | | **45B** | 45B |
| **44** | 44 | **44** | 44, F5s | *n.a.* | |
| | 12r | | 12r | **r12r** | 12r |
| | 12l | **47/12** | 12l | **i12r** | 12r |
| | 12m | | | **c12r** | 12r, rostral part of 12l and 45A |
| **47/12** | 12o | | | **12l** | 12l |
| | | | | **12m** | 12m, 12o |

*Table 1 continued on next page*

*Table 1 continued*

| Petrides & Pandya vs. Rapan | | Morecraft vs. Rapan | | Caminiti vs. Rapan | |
|---|---|---|---|---|---|
| | | | | 12o | 12o |
| 13 | 13m | *n.a.* | | 13a/13b | 13a, 13b |
| | 13l | | | 13m/13l | 13m, 13l |
| 11 | 11l, part of 12r and 12m | *n.a.* | | 11m | 11m |
| | | | | 11l | 11l, 11m |
| | 14r | 14 | 14r | | 14r |
| 14 | 14c | | 14c | | 14c |
| | Caudal part of 10mv; 13a, 13b | | Caudal part of 10mv | 14 | 10mv |

forming a letter H (*Figure 2—figure supplement 2*). Though not as deep as sulci, there are several dimples within the PFC, for example, the anterior dimple (*aspd*) in its rostral part, and more caudally, the posterior dimple (*pspd*) in the dorsal PFC. Finally, ventral to the *ps* the inferior principal dimple (*ipd*) was recognizable only in the right hemisphere of DP1. The appearance of these dimples in three *M. fascicularis* brains is rather variable. Since the Yerkes19 atlas is based on structural MRI scans of 19 adult macaques, these dimples are missing from its surface (*Figure 2—figure supplement 1*).

As specified in the 'Materials and methods' section, previously published architectonic literature and nomenclature conventions were used as a starting point for the cytoarchitectonic analysis. All borders detected by visual inspection were then tested by image analysis and statistical validation, and the most distinguishing cytoarchitectonic features of the identified subdivisions belonging to the same area are summarized in *Table 2*.

## Frontopolar and orbital areas

The most rostral tip of the primate brain is occupied by the so-called frontal polar region (largely occupied by Walker's area 10), where we identified four distinct areas (*Figures 2 and 3A*): that is, area 10d (dorsal) located on the dorsolateral surface of the frontal pole, areas 10mv (medioventral) and 10md (mediodorsal) on its medial surface, and 10o (orbital) on its most ventral aspect, occupying the rostral portion of the ventromedial gyrus. With a well-developed layer IV, this entire region represents a highly granular cortex, with slight differences in its appearance between the four defined areas, whereby medial areas 10md and 10mv show a slightly thinner layer IV compared to adjacent areas 10d and 10o, respectively (*Figure 3B*). Unlike the rest of area 10, area 10d has more densely packed layers II and V, with small-sized pyramids, whereas in the medial (10md/10mv) and orbital (10o) portions characteristic larger pyramids could be recognized in the upper part of layer V. 10mv can be distinguished from the neighbouring areas 10md and 10o by the much thinner appearance of its layer V. Additionally, the border between layers II and III is clearly visible in area 10o, but not in 10mv (*Figure 3B*). *Figure 3C* shows the result of the statistical validation of these newly defined subdivisions of area 10, as well as of the corresponding borders with adjacent areas.

Twelve areas within the orbitofrontal and ventrolateral cortex (*Figures 2 and 4A*; *Figure 4—figure supplements 1 and 2*) were identified: two are located within Walker's area 14 (14r and 14c), four are within Walker's area 13 (13b, 13a, 13m, and 13l), two are in Walker's area 11 (11m and 11l), and four are within Walker's area 12 (12r, 12m, 12l, and 12o). Moving posteriorly along the ventromedial gyrus, granular cortex of area 10o transitions into dysgranular area 14r and further caudally into agranular area 14c. Similar to areas 14, subdivisions of area 13, which are found on the medial wall of the *morb*, show rostro-caudal differences in the appearance of their layer IV, that is, rostral area 13b is granular, whereas caudal area 13a is dysgranular (*Figure 4B*). However, unlike 14r and 14c, areas 13b and 13a have bilaminar layer V. Laterally, on the orbitofrontal gyrus, granular areas 11m and 11l occupy its rostral portion, while caudally dysgranular areas 13m and 13l are located, just rostral to the agranular

**Table 2.** Prominent cytoarchitectonic features highlighted for all 35 identified prefrontal areas.

| Area | Layer IV | Cytoarchitecture | |
|------|----------|------------------|---|
| 10d | | Small-size pyramids in III/V; dense granular layers II/IV | |
| 10md | | Wide, pale layer V | |
| 10mv | | Prominent middle-size pyramids in V | |
| 10o | Granular | Prominent layer II | |
| 14r | Dysgranular | well-developed layer II; columnar pattern in IV-V | |
| 14c | Agranular | Pale layer III | |
| 11m | | Sublamination of V (Va/Vb); cell clusters in Va | |
| 11l | Granular | Sublamination of V (Va/Vb) | |
| 13b | Granular | Columnar pattern in IV-V | |
| 13a | | Sublamination of V (Va/Vb) | |
| 13m | | Sublamination of V (Va/Vb); layer Va wider than Vb | |
| 13l | Dysgranular | Sublamination of V (Va/Vb); both layers of comparable width | |
| 12r | Dysgranular | No sublamination of V | |
| 12m | | Sublamination of V (Va/Vb) | |
| 12l | Granular | Sublamination of V (Va/Vb) | |
| 12o | Dysgranula | No sublamination of V | |
| 9m | | Sublamination of V (Va/Vb) | |
| 9d | | Gradient in cell-size within III; sublamination of V (Va/Vb); pale layer Vb is wider in 9d than 9l | |
| 9l | Granular | Gradient in cell size within III; sublamination of V (Va/Vb) | |
| a46d | | | Well-developed layer II |
| a46df | | | Scattered middle-sized pyramids in lower layer III |
| a46vf | | | Scattered middle-sized pyramids in layer III |
| a46v | Granular | Scattered middle-sized pyramids in upper layer V | Prominent layer II, but not as in a46d |
| p46d | | | Well-developed layer II; densely packed cells in layer III |
| p46df | | | Densely packed cells in layer III; scattered middle-sized pyramids in lower layer III |
| p46vf | | | Scattered middle-sized pyramids in layer III |
| p46v | Granular | Cells more uniform in size throughout the cortex | Prominent layer II, but not as in p46d |
| 8Bm | | Layer VI pale compared to dorsal subdivisions | |
| 8Bd | | Dark, prominent layer II | |
| 8Bs | Dysgranular | Small size pyramids in III and V compared to 8Bd | |
| 8Ad | | Upper layer III pale | |
| 8Av | Granular | Lower layer III pale; highly granular cortex | |
| 45A | | Middle-sized pyramids in layer III | |
| 45B | Granular | Layer IV less developed | |
| 44 | Dysgranular | Few larger pyramids scattered in layer V | |

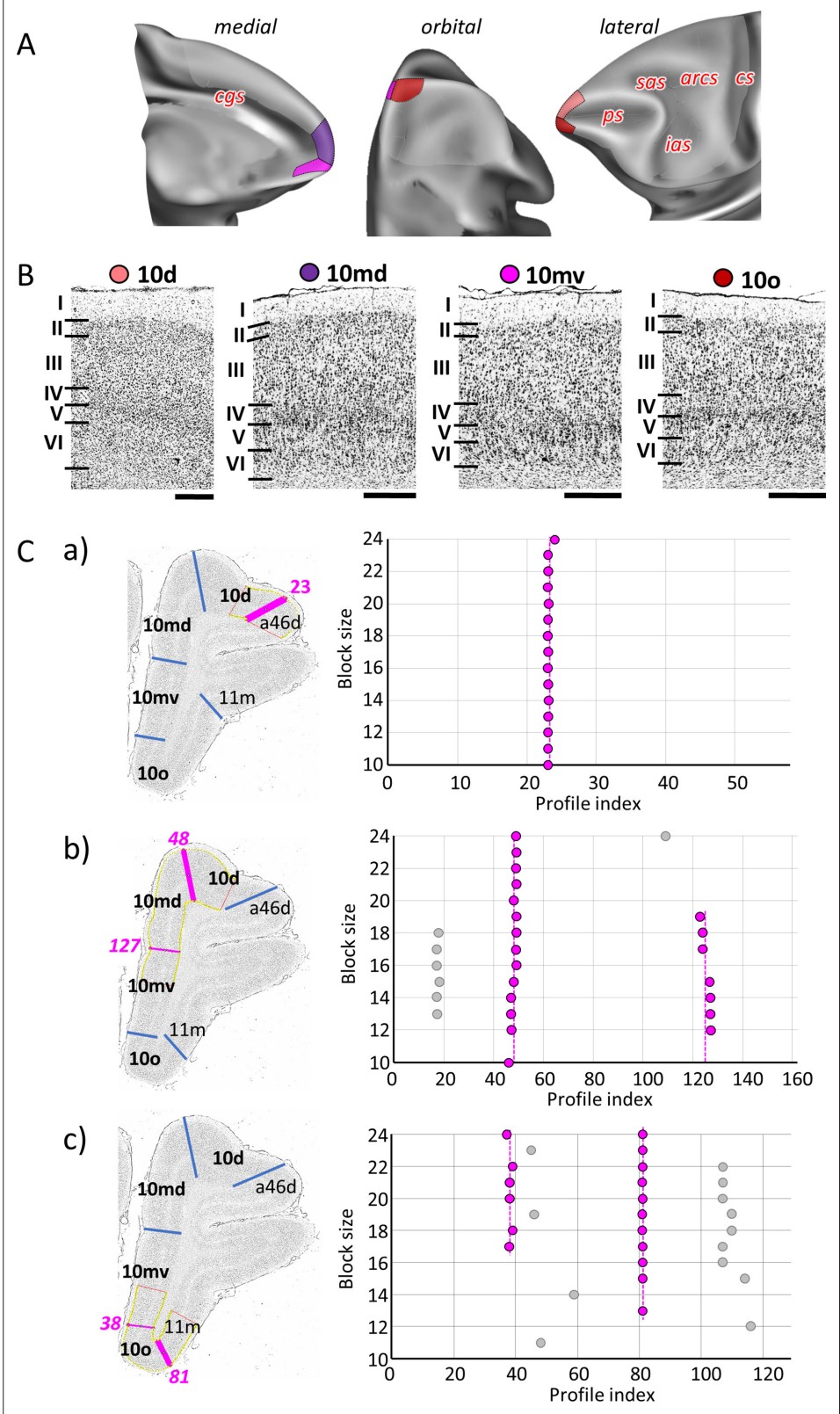

**Figure 3.** Quantitative analysis of the cytoarchitecture of Walker's area 10 (**Walker, 1940**). (**A**) Position and extent of subdivisions of Walker's area 10 within the hemisphere are displayed on orbital, lateral, and medial views of the Yerkes19. Macroanatomical landmarks are marked in red letters. (**B**) High-resolution photomicrographs show cytoarchitectonic features of areas 10d, 10md, 10mv, and 10o. Each subdivision is labelled by a coloured

*Figure 3 continued on next page*

*Figure 3 continued*

dot, matching the colour of the depicted area on the 3D model. (**C**) We confirmed cytoarchitectonic borders by a statistically testable method, where the Mahalanobis distance (MD) was used to quantify differences in the shape of profiles extracted from the region of interest. Profiles were extracted between outer and inner contour lines (yellow lines drawn between layers I/II and VI/white matter, respectively) defined on grey-level index (GLI) images of the histological sections (left column). Pink lines highlight the position of the border for which statistical significance was tested. The dot plots (right column) reveal that the location of the significant border remains constant over a large block size interval (highlighted by the red dots). (a) depicts analysis of the border between areas 10d and a46d (profile index 23); (b) depicts analysis of the border delineating dorsally located subdivisions, 10d and10md (profile index 48), as well as the medial border segregating dorsal and ventral subdivision, 10md and 10mv (profile index 127); and (c) depicts analysis of the borders between ventrally positioned subdivisions of the frontal polar region, 10mv and 10o (profile index 38) and 10o and 11m (profile index 81). Scale bar 1 mm. Roman numerals indicate cytoarchitectonic layers. arcs, spur of the arcuate sulcus; cgs, cingulate sulcus; cs, central sulcus; ias, inferior arcuate sulcus; ps, principal sulcus; sas, superior arcuate sulcus.

---

insular region. The main difference among the subdivisions of area 11 is the pattern of cells in sublayer Vb, which is occasionally broken into aggregates of cells in area 11m, but continuous in area 11l. Similar, difference between 13m and 13l is related to the sublaminas V; that is, in 13m layer Va is wider that Vb, whereas in 13l both layers are of comparable width (*Figure 4B*). On the ventrolateral surface, the four subdivisions of Walker's area 12 are distinguished by the degree of granularity of layer IV, and the size and distribution pattern of the pyramids in layers III and V (*Figure 4B*). The most rostral area on the medioventral surface of the prefrontal cortex, 12r, is a dysgranular cortex with characteristic columnar aspect in layers III and V. Area 12m, located on the lateral wall of the *lorb*, has a bipartite layer V and a well-developed layer IV which distinguishes it from surrounding areas 12r and 13l. Area 12o, located medial to 12l on the caudal medioventral convexity, has a thin and weakly stained layer IV, and no obvious sublamination in layer V. Area 12l is granular cortex with clear subdivisions in layer V (*Figure 4B*).

## Medial and dorsolateral areas

The dorsal portion of the prefrontal cortex directly abutting area 10 of Walker is occupied by his area 9, within three distinct areas were identified (*Figures 2 and 5A*): area 9m, located on the medial surface between areas 10md rostrally and 8Bm caudally, is followed dorsally by area 9d, which in turn is delimited laterally by 9l (directly adjacent to area 46). Areas 9d and 9l are limited rostrally by area 10d and caudally by areas 8Bd and 8Bs, respectively. All subdivisions of area 9 are characterized by the low packing density and width of layer III, and the sublamination of layer V with a prominent Va containing relatively large pyramidal cells and a sparsely populated Vb, which distinguishes them from neighbouring areas (*Figure 5B*). This contrast between layers Va and Vb is particularly conspicuous in area 9l, thus clearly highlighting its border with area 9d (*Figure 5C*). Area 9d can be distinguished from 9l by its wider, pale layer V. The most recognizable feature of areas 9d and 9l, which is not visible in area 9m, is the gradual increase in the size of layer III pyramids, with largest cells found close to layer IV (*Figure 5B*).

As mentioned above, the dorsal portion of the most posterior part of the PFC is occupied by three subdivisions of Walker's area 8B (*Figures 2 and 6A*): area 8Bm is located on the medial hemispheric surface, delimited caudally by the premotor cortex and rostrally by area 9m; area 8Bd is located on the dorsal surface along the midline; 8Bs is a newly identified area found on the cortical surface lateral to 8Bd and reaching the fundus of the *sas*. Walker's area 8A occupies the cortex surrounding the most caudal portion of the *ps*, where it abuts areas p46. Here we identified area 8Ad dorsally, which extends into the ventral wall of the *sas*, reaching its fundus, and area 8Av ventrally, extending into the rostral wall of the *ias*, and also reaching its fundus (*Figures 2 and 6A*). Subdivisions of area 8B are dysgranular, whereas subdivisions of area 8A present a clearly developed layer IV (*Figure 6B*). Area 8Bm is more weakly laminated than 8Bd and 8Bs, but presents a columnar organization not visible in the lattermost areas. Area 8Bd is characterized by a more densely packed layer II and by lager pyramids in layers III and V than areas 8Bm or 8Bs. Both subdivisions of area 8A have a clear laminar structure, with a well-developed layer IV, which is especially wide and dense in 8Av (*Figure 6B*). All borders were statistically validated by the quantitative cytoarchitectonic analysis (*Figure 6C*; *Figure 7—figure supplement 1* and *Figure 8—figure supplement 2*).

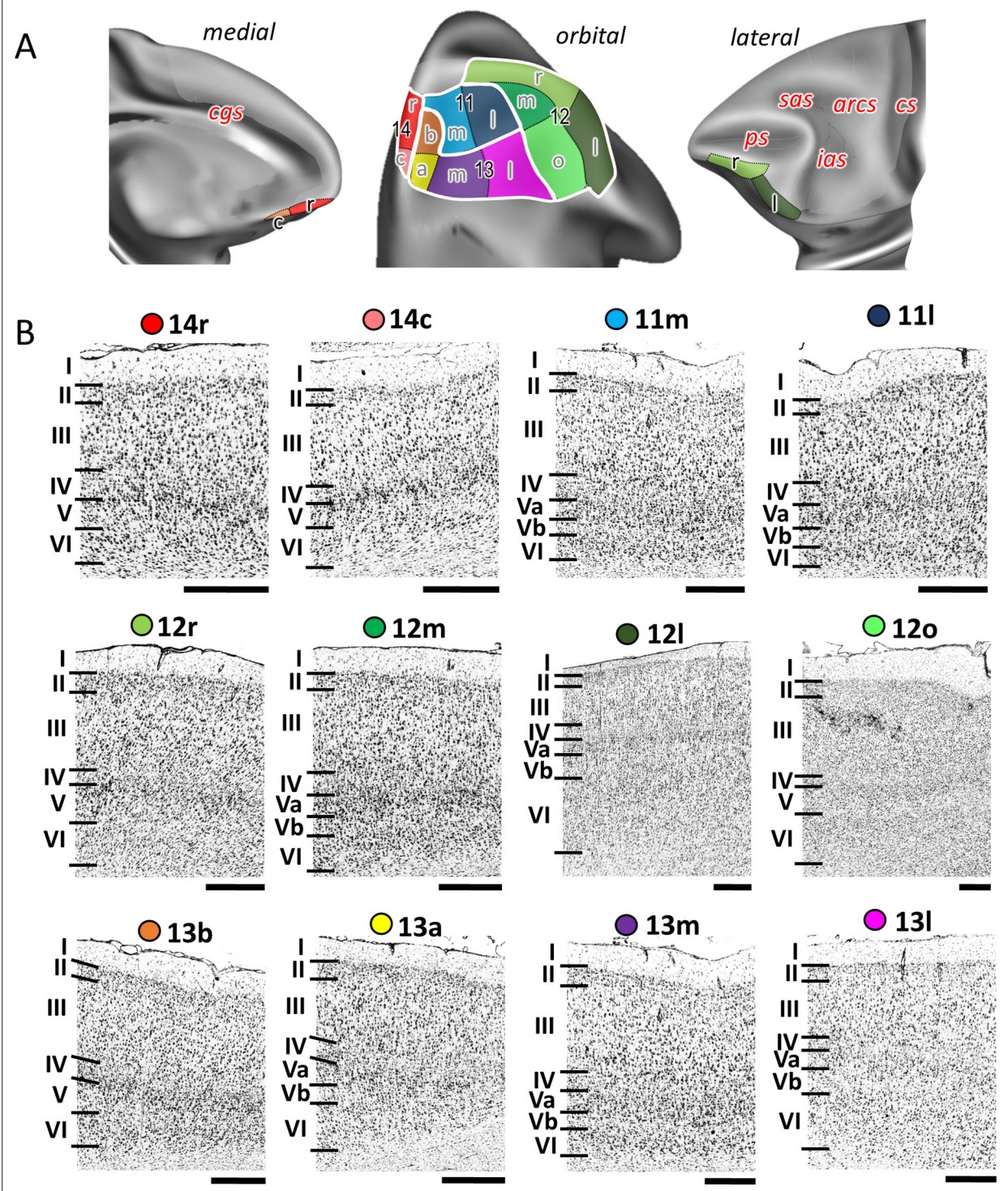

**Figure 4.** Cytoarchitecture of orbitofrontal areas.

(**A**) Position and extent of the orbitofrontal areas within the hemisphere are displayed on orbital, lateral, and medial views of the Yerkes19. Macroanatomical landmarks are marked in red letters. (**B**) High-resolution photomicrographs show cytoarchitectonic features of orbitofrontal 14r, 14c, 11m, 11l, 12r, 12m, 12l, 12o, 13b, 13a, 13m, and 13l. Each subdivision is labelled by a coloured dot, matching the colour of the depict area on the 3D model. Scale bar 1 mm. Roman numerals (and letters) indicate cytoarchitectonic layers. arcs, spur of the arcuate sulcus; cgs, cingulate sulcus; cs, central sulcus; ias, inferior arcuate sulcus; ps, principal sulcus; sas, superior arcuate sulcus.

The online version of this article includes the following figure supplement(s) for figure 4:

*Figure 4 continued on next page*

*Figure 4 continued*

**Figure supplement 1.** Statistically testable borders (pink lines) confirmed by the quantitative analysis for the rostral orbital and ventrolateral areas 14r, 13b, 11m, 11l, 12m, and 12r.

**Figure supplement 2.** Statistically testable borders (pink lines) confirmed by the quantitative analysis for the caudal orbital and ventrolateral areas 14c, 13a, 13m, 13l, and 12o.

A mosaic of distinct areas was identified within Walker's area 46 which encompasses our areas a46d, a46df, a46vf, a46v, p46d, p46df, p46vf, and p46v (*Figures 2 and 7A*; *Figure 7—figure supplements 1 and 2*). Such segregation results from a principal subdivision of area 46 into areas located within the anterior portion of the *ps* (the 'a46-areas') and those found in its posterior portion (the 'p46-areas'), as well as differences between areas located on the dorsal (the '46d-areas') and ventral (the '46v-areas') shoulders of the sulcus, or around its fundus (the '46f-areas'), depicted on our schematic drawing of the *ps* (*Figure 7A*). Cytoarchitectonically, 'a46' and 'p46' areas can be distinguished by differences in the size of layer III and V pyramids, which are smaller in the posterior than in the anterior areas (*Figure 7B*). Dorsal subdivisions of area 46 have a wider and more densely packed layer II than the ventral areas, which, in turn, have more a more prominent layer IV, and larger cells in layers V and VI. Areas located around the fundus of the ps, that is, areas a46df/46vf and p46df/46vf, are additionally characterized by a clear border between layer VI and the white matter (*Figure 7B*).

## Caudal ventral areas

Rostral to the ventral premotor cortex, we identified areas 44, 45A, and 45B (*Figures 2 and 8A*; *Figure 8—figure supplements 1 and 2*) belonging to the ventral granular PFC. Area 44 can be found along the deeper portion of the ventral wall of the *ias*, and encroaching onto its dorsal wall, where it abuts area 45B. The border between areas 45B and 45A was consistently found at the tip of the *ias*, whereby area 45A occupies the prearcuate convexity. Dysgranular areas 44 and granular area 45B can also be distinguished by differences in layer V which presents larger pyramids in the former than in the latter area (*Figure 8B*). Layer IV of 45A is wider than that of 45B. Additionally, layer III pyramids tend to build clusters in area 45B, but not in 45A (*Figure 8B*).

## Receptor architectonic analysis

The regional and laminar distribution patterns of 14 distinct receptor types were characterized throughout the macaque prefrontal cortex for each cytoarchitectonically defined area (with the exception for 13a and 14c due to technical limitations) by means of receptor profiles. Silver-stained sections from the corresponding receptor brain were aligned with the receptor autoradiographs at the same macroanatomic level in order to enable comparison of cytoarchitectonic border positions with receptor distribution patterns. Not all receptors show each areal border, and not all borders are equally clearly defined by all receptor types. Changes in receptor distribution patterns confirmed cytoarchitectonically identified borders, but did not reveal further subdivisions within the PFC.

In detail, neurotransmitter receptors display distinct laminar distribution patterns, which are preserved across all examined areas for most receptor types with the notable exception of the $M_2$ receptors (*Figure 9*; *Figure 9—figure supplements 1–3*). In some areas $M_2$ receptors present a single maximum in layer V (10mv, 10o, 14r, 13b, subdivisions of areas 11 and 46). Other areas present a bimodal pattern, with maxima in layers III and V. In some cases, both maxima are of comparable intensity (13m, 13l, subdivisions of area 12), and in other areas the maximum in layer III is clearly higher than that in layer V (10d, 10md, 44, and subdivisions of areas 9, 8B, 8A, and 45). Kainate receptors also constitute a notable exception because they are the only ones consistently presenting higher densities in the deeper than in the superficial cortical layers. The $\alpha_1$ and 5-HT$_{1A}$ receptors stand out due to their bimodal laminar distribution, with the highest of the two maxima located in the superficial layers. The remaining receptors present a rather unimodal laminar distribution pattern, whereby the width and position of the maximum varies depending on the receptor type. The $D_1$ receptor reaches its maximum density in subcortical structures and a relatively homogeneous distribution throughout the neocortex.

Absolute receptor densities (averaged over all cortical layers) varied by several orders of magnitude depending on the receptor type (*Table 3*; *Figure 10—figure supplement 1* and *Figure 11—figure supplement 1*). Highest absolute values were found for the GABA$_B$ receptor (2644 fmol/mg in

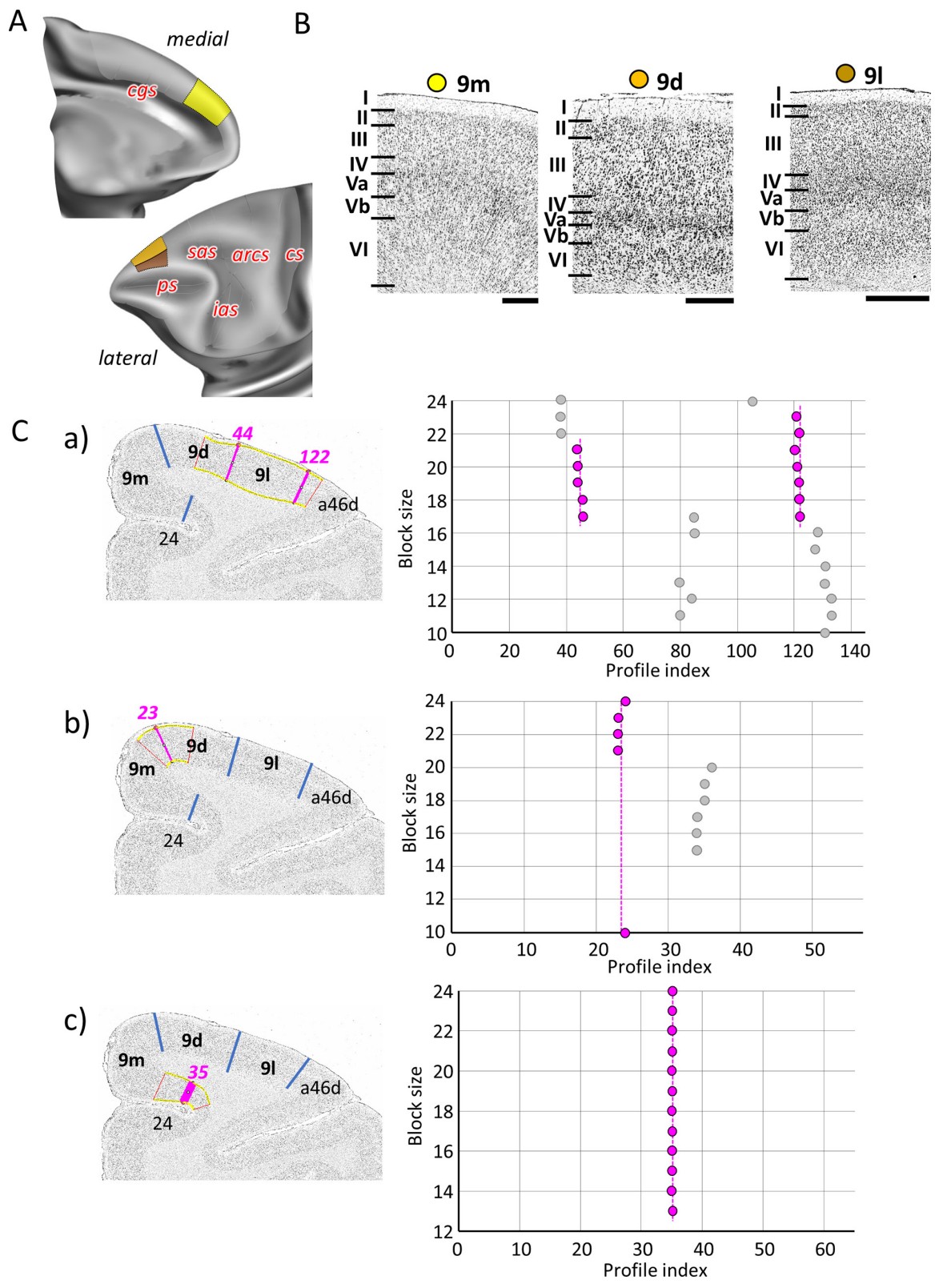

**Figure 5.** Quantitative analysis of the cytoarchitecture of Walker's area 9 (*Walker, 1940*). (**A**) Position and extent of the rostral medial and dorsolateral prefrontal areas within the hemisphere are displayed on lateral and medial views of the Yerkes19. Macroanatomical landmarks are marked in red letters. (**B**) High-resolution photomicrographs show cytoarchitectonic features of areas 9m, 9d, and 9l. Each subdivision is labelled by a coloured dot, matching the colour of the depict area on the 3D model. (**C**) We confirmed cytoarchitectonic borders by statistically testable method (for details see *Figure 3*).

*Figure 5 continued on next page*

*Figure 5 continued*

(a) depicts analysis of the borders between area a46d and 9l (profile index 122), as well as 9l and 9d (profile index 44); (b) depicts analysis of the border between dorsal and medial subdivision, 9d and 9m (profile index 44); and (c) depicts analysis of the border distinguishing medial subdivision 9m from cingulate cortex, area 24 (profile index 35). Scale bar 1 mm. Roman numerals (and letters) indicate cytoarchitectonic layers. arcs, spur of the arcuate sulcus; cgs, cingulate sulcus; cs, central sulcus; ias, inferior arcuate sulcus; ps, principal sulcus; sas, superior arcuate sulcus.

11l) and lowest densities for the $D_1$ receptor (67 fmol/mg in 9l). Considerable differences in absolute densities were also found within a single neurotransmitter system. For example, highest muscarinic cholinergic densities were found for the $M_1$ receptor (between 1152 fmol/mg in 12m and 708 fmol/mg in 8Av) and lowest for the $M_2$ receptor (between 223 fmol/mg in 13l and 134 fmol/mg in 14r). In general, lowest receptor densities were measured in subdivisions of areas 8B and 8A, which consequently displayed the smallest fingerprints of all PFC areas. Conversely, highest receptor densities were mainly located in orbitofrontal and frontopolar areas (*Figures 10 and 11*; *Figure 10—figure supplement 1* and *Figure 11—figure supplement 1*).

Out of all prefrontal areas examined here, we found that the frontopolar region (i.e. areas 10) is characterized by the highest density of kainate and $GABA_A$/BZ densities (*Table 3*). Changes in the laminar pattern of $GABA_A$, $M_1$, $M_2$, $\alpha_1$, and $5HT_{1A}$ receptors most clearly highlight the cytoarchitectonically defined borders within area 10 (*Figure 9*; *Figure 9—figure supplements 1 and 2*). Differences in the size of fingerprints particularly reflect the dorsoventral subdivision, with smaller sized fingerprints in areas 10d/10md compared to 10mv/10o (*Figure 10*; *Figure 10—figure supplement 1*). Both ventrally positioned subdivisions of area 10 (i.e. areas 10mv and 10o) differed significantly from caudally adjacent area 14r, though not always for the same receptor types (*Table 4*). Area 14r presented significantly lower AMPA and $GABA_A$ receptor densities than 10mv and 10o, respectively. Additionally, $GABA_A$/BZ densities in 10mv and 10o were significantly higher than in 14r. Likewise, dorsal subdivisions of area 10 presented a differential pattern of significant receptor densities compared to neighbouring areas. Areas 10d and 10md contain significantly higher kainate and NMDA receptor densities, respectively, than caudally adjacent subdivisions of area 9.

Within the orbitofrontal cortex (OFC), laminar distribution patterns of kainate, $GABA_A$, $GABA_B$, $M_1$, $M_2$, and $M_3$ receptors most clearly reflect the cytoarchitectonically identified areas 14r, 13b, 11m, and 11l, whereas caudal orbital areas 13m and 13l are highlighted by the laminar distribution of kainate, $GABA_A$, $\alpha_1$, $M_2$, $M_3$, and $5HT_{1A}$ receptors (*Figure 9*; *Figure 9—figure supplements 1 and 2*). Particularly areas 14r and 12l stand out due to the shape and size of their fingerprints (*Figure 10*; *Figure 10—figure supplement 1*). Area 14r is characterized by the lowest $GABA_A$/BZ and $M_2$ densities within PFC, but is among areas with the highest $GABA_B$ and $\alpha_1$ levels (*Table 3*). In addition to the above described differences with frontopolar areas, 14r contains significantly lower $GABA_A$ and $M_3$ densities than area 11m (*Table 4*). Rostral orbital region occupied by the subdivisions 11m and 11l measured highest concentration levels for $M_3$ among all prefrontal areas, and dysgranular areas 13m and 13l have the highest levels of AMPA, $M_2$, and $\alpha_2$ in regard to all other orbital areas (*Table 3*). Significant differences between 11l and neighbouring areas were only found for the $GABA_B$ densities in area 13m, whereas 11m differed significantly from areas 14r and 10o in its $GABA_A$ and $M_3$ and its kainate densities, respectively (*Table 4*).

Within Walker's area 12, differences between rostral ventrolateral areas 12m and 12r are best delineated by changes in the laminar distribution patterns of AMPA, $GABA_A$, $5HT_{1A}$, $M_1$, and $M_3$ receptors, whereas the border between caudal subareas 12o and 12l is most clearly revealed by the laminar distribution pattern of kainate, $GABA_A$, $\alpha_1$, $M_2$, $M_3$, and $5HT_{1A}$ receptors (*Figure 9*; *Figure 9—figure supplements 1 and 2*). In general terms, 12r has the highest and 12l the lowest densities measured within Walker's area 12, and in the size of their fingerprints (*Figure 10*; *Figure 10—figure supplement 1*). Medially positioned areas (12m and 12o) have significantly higher $\alpha_2$ receptor densities than laterally positioned areas (12r and 12l). For the lateral areas we also found significant differences in the rostro-caudal direction, whereby 12r has significantly higher $GABA_A$ densities than 12l. Additionally, 12r contains significantly higher AMPA receptor densities than dorsally adjacent areas a46v and p46v. Area 12r also contains significantly higher $GABA_A$, $GABA_B$, and $M_3$ receptor densities than does p46v (*Table 4*).

Differences in receptor architecture also revealed a novel cytoarchitectonic subdivisions of Walker's areas 9 and 8B. In particular, the borders between areas 9m, 9d, and 9l are most clearly reflected in

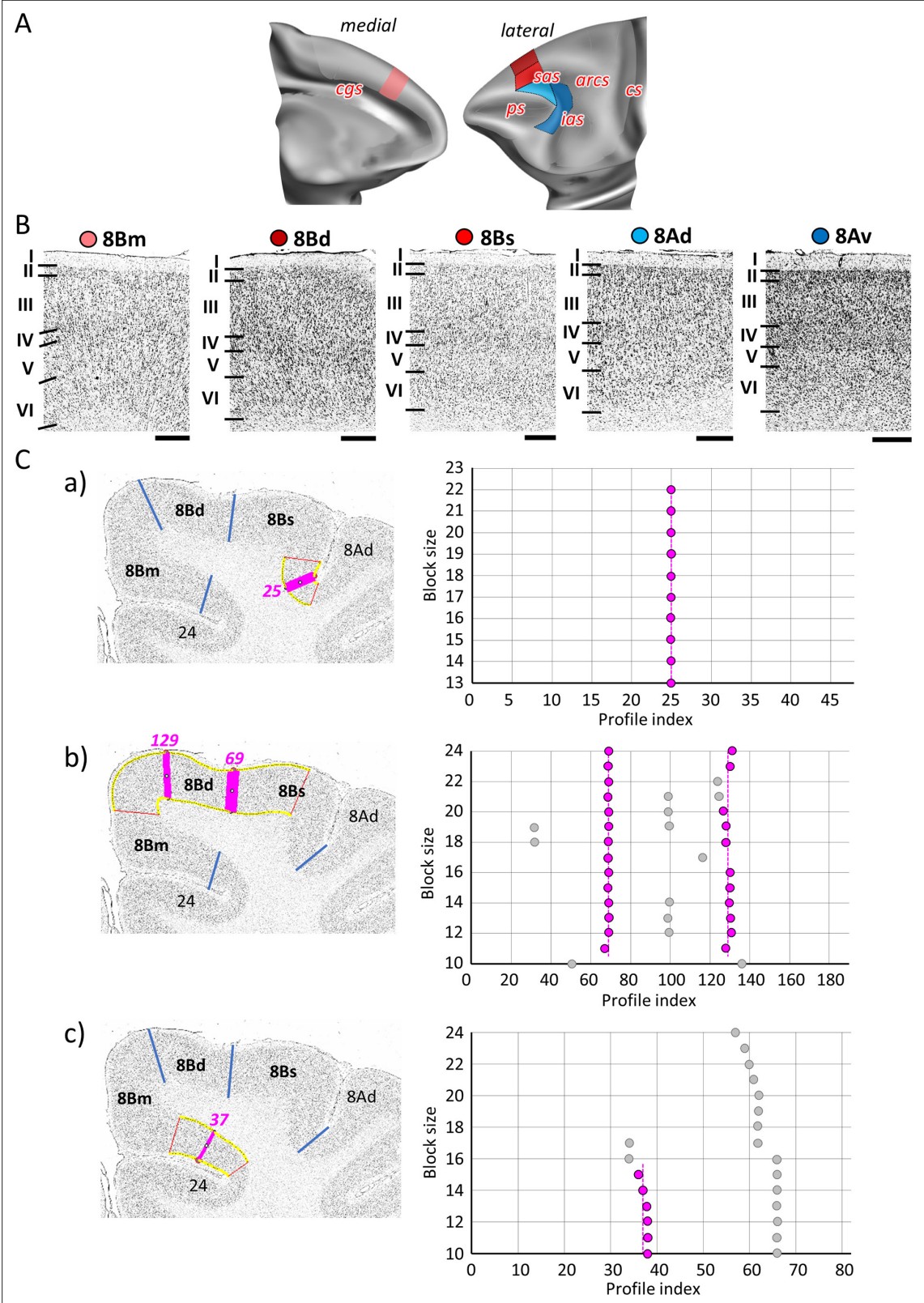

**Figure 6.** Quantitative analysis of the cytoarchitecture of Walker's area 8B (***Walker, 1940***). (**A**) Position and the extent of the caudal medial and dorsolateral prefrontal areas within the hemisphere are displayed on lateral and medial views of the Yerkes19. Macroanatomical landmarks are marked in red. (**B**) High-resolution photomicrographs show cytoarchitectonic features of areas 8B (8Bm, 8Bd, 8Bs) and 8A (8Ad, 8Av). Each subdivision is labelled by a coloured dot, matching the colour of the depict area on the 3D model. (**C**) We confirmed cytoarchitectonic borders of new 8B subdivisions by

*Figure 6 continued on next page*

**Figure 6 continued**

statistically testable method (for details see **Figure 3**). (a) depicts analysis of the border that separates new subdivisions 8Bs from neighbouring area 8Ad (profile index 25); (b) depicts analysis of the borders which delineate area 8Bd from surrounding areas 8Bs and 8Bm (profile index 69), as well as 8Bd and 8Bm (profile index 129); and (c) depicts analysis of the border distinguishing medial subdivision 8Bm from cingulate cortex, area 24 (profile index 37). Statistically testable borders for area 8Ad (adjacent to p46d) shown in **Figure 7—figure supplement 2** and for area 8Av borders can be seen in the **Figure 8—figure supplement 2**. Scale bar 1 mm. Roman numerals (and letters) indicate cytoarchitectonic layers. arcs, spur of the arcuate sulcus; cgs, cingulate sulcus; cs, central sulcus; ias, inferior arcuate sulcus; ps, principal sulcus; sas, superior arcuate sulcus.

the laminar distribution patterns of kainate, NMDA, $GABA_A/BZ$, $M_3$, $\alpha_2$, $5HT_{1A}$, and $5HT_2$ receptors (**Figure 9**; **Figure 9—figure supplements 1 and 3**). Subdivision of area 8B into 8Bm, 8Bd, and 8Bs is clearly revealed by the differences in the laminar distribution patterns of AMPA, kainate, $M_1$, $M_3$, and $5\text{-}HT_{1A}$ receptors (**Figure 9**; **Figure 9—figure supplements 1 and 2**). Newly defined area 8Bs contains the lowest kainate density out of all prefrontal areas, whereas area 8Bd presents the lowest NMDA and $GABA_A$ receptor densities within the PFC. In general, subdivisions of Walker's area 9 contain higher receptor densities than those of his area 8B (**Table 3**), and this is reflected in their slightly larger fingerprints (**Figure 11—figure supplement 1**). There are also pronounced differences in the shape of the fingerprints, and this becomes particularly obvious when observing the normalized fingerprints (**Figure 11**). Areas 9d and 9l show significantly higher kainate, NMDA, $GABA_A$, and $M_3$ receptor densities than their caudal counterparts within area 8B (i.e. 8Bd and 8Bs, respectively). Additionally, $\alpha_2$ and $5\text{-}HT_{1A}$ densities are significantly higher in 9d than in 8Bd (**Table 4**). Area 8Bs has significantly lower kainate receptor levels than laterally adjacent area 8Ad. The border between areas 8Bs and 8Ad is also revealed by differences in the laminar distribution pattern of kainate, $M_1$, $\alpha_1$, $5\text{-}HT_{1A}$, and $5\text{-}HT_2$ receptors (**Figure 9**; **Figure 9—figure supplements 1–3**).

The border between the dorsal and ventral subdivisions of Walker's area 8A (i.e. 8Ad and 8Av) is most clearly indicated by laminar differences in the distribution of kainate, $GABA_A$, $GABA_B$, $M_2$, and $\alpha_1$ receptors. Area 8Av was characterized by the lowest density of AMPA, $GABA_B$, $M_1$, $M_3$, $\alpha_1$, $\alpha_2$, and $5HT_{1A}$ receptors out of all areas analysed here (**Table 3**), thus for this area the size of the fingerprint was the smallest in the PFC (**Figure 11**; **Figure 11—figure supplement 1**). Area 8Av has significantly lower kainate, $\alpha_1$, and $5HT_{1A}$ receptor densities than 8Ad. It also has significantly lower densities of kainate, $M_3$, and $\alpha_2$ than neighbouring area 45A, of AMPA, NMDA, $\alpha_1$, $\alpha_2$, and $5HT_{1A}$ receptors than area 45B, as well as of kainate, $M_3$, $\alpha_2$, and $5HT_{1A}$ receptors than area p46v (**Table 4**).

Subdivisions of Walker's area 46 within and around the ps identified by cytoarchitectonic analysis were revealed by the following differences in receptor architecture. Changes in the laminar distribution patterns of AMPA, kainate, $GABA_A$, $GABA_B$, $GABA_A/BZ$, and $M_3$ receptors most clearly reveal delineation of subdivisions within Walker's area 46 for both anterior and posterior subareas (**Figure 9**; **Figure 9—figure supplements 1 and 2**). In general, higher densities were found in areas located around the fundus of ps than in those located on its dorsal and ventral shoulders, and higher muscarinic cholinergic densities were found in all anterior subdivisions of area 46 than in their caudal counterparts (**Table 3**). Furthermore, differences in the fingerprints of anteriorly located subdivisions of area 46 and their corresponding posterior counterparts were greater for areas located on the shoulder (e.g. when comparing a46d and p46d) than for areas located around the fundus (e.g. when comparing a46df and p46df; **Figure 11**; **Figure 11—figure supplement 1**). Along the entire length of the ps we found significantly higher $\alpha_2$ receptor densities in areas located around its fundus than the adjacent areas on the shoulder (**Table 3**). Interestingly, significant differences in kainate receptors were found only for anterior areas, whereby they were higher in a46d and a46v than in a46df and a46vf, respectively (**Table 4**).

Cytoarchitectonic borders between areas 45A, 45B, and 44 are clearly reflected by changes in the laminar distribution pattern of kainate, $GABA_B$, $GABA_A/BZ$, $M_1$, $M_2$, $\alpha_1$, and $5\text{-}HT_{1A}$ receptors (**Figure 9**; **Figure 9—figure supplements 1 and 2**). The size of the normalized receptor fingerprints increases gradually when moving from area 45A through 45B to 44 (**Figure 11**). Area 45A contains significantly higher kainate levels compared to 45B (**Table 4**). Out of all prefrontal areas, area 44 had highest concentration levels recorded for $5HT_2$ receptors. Furthermore, whereas area 44 presents one of the highest $5\text{-}HT_{1A}$ receptor densities within the PFC, area 45A contains the second lowest PFC density of this receptor type, and 45B only an intermediate to low value (**Table 3**), and these differences are reflected in the unique shaped normalized fingerprint of area 44 (**Figure 11**).

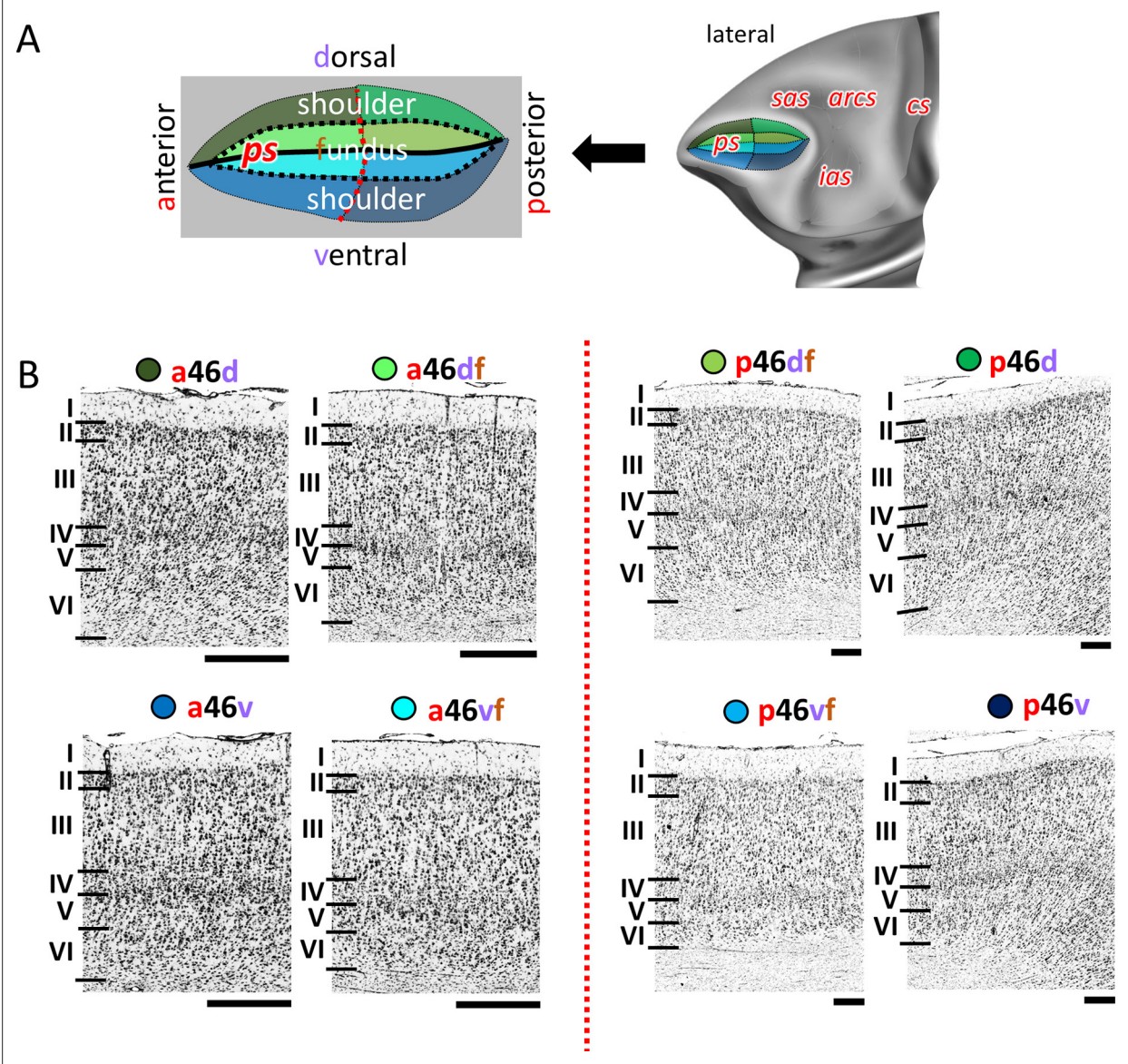

**Figure 7.** Cytoarchitecture of Walker's area 46 (**Walker, 1940**). (**A**) Position and the extent of areas located within and around the *ps*, are displayed on lateral view of the Yerkes19. Additionally, schematic drowning demonstrates how identified subdivisions are labelled with letters highlighted in red. Macroanatomical landmarks are marked in red letters. Black line indicates fundus, black dotted line marks border between shoulder and fundus region, and red dotted line separates anterior and posterior portion of sulcus. (**B**) High-resolution photomicrographs show cytoarchitectonic features of anterior areas of 46 (a46d, a46df, a46vf, a46v) and posterior ones (p46d, p46df, p46vf, p46v), separated by red dashed line. Each subdivision is labelled by a coloured dot, matching the colour of the depict area on the 3D model. Scale bar 1 mm. Roman numerals indicate cytoarchitectonic layers. arcs, spur of the arcuate sulcus; cs, central sulcus; ias, inferior arcuate sulcus; ps, principal sulcus; sas, superior arcuate sulcus.

The online version of this article includes the following figure supplement(s) for figure 7:

**Figure supplement 1.** Statistically testable borders (pink lines) confirmed by the quantitative analysis for the rostral region of the ps, occupied by the anterior subdivisions of area 46; a46d, a46df, a46vf, and a46v.

**Figure supplement 2.** Statistically testable borders (pink lines) confirmed by the quantitative analysis for the caudal region of the ps, occupied by the posterior subdivisions of area 46; p46d, p46df, p46vf, and p46v.

## Functional connectivity analysis

In addition to distinct cyto- and receptor architectonic features, areas have also been characterized by their unique functional connectivity pattern. To facilitate the description and interpretation of our results, we created summary figures emphasizing interareal connections (between subdivisions

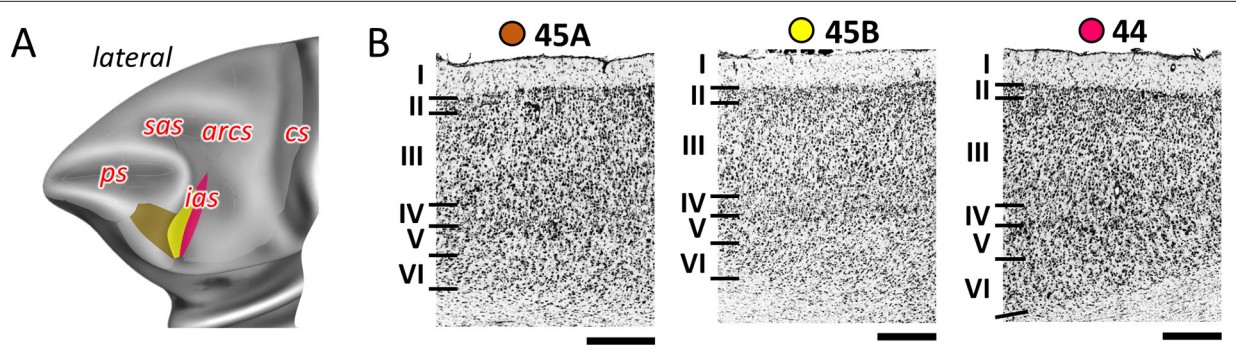

**Figure 8.** Cytoarchitecture of areas 44 and 45. (**A**) Position and the extent of the posterior ventrolateral areas within the hemisphere are displayed on lateral view of the Yerkes19. Macroanatomical landmarks are marked in red letters. (**B**) High-resolution photomicrographs show cytoarchitectonic features of areas 44 and 45 (45A, 45B). Each subdivision is labelled by a coloured dot, matching the colour of the depict area on the 3D model. Scale bar 1 mm. Roman numerals indicate cytoarchitectonic layers. arcs, spur of the arcuate sulcus; cs, central sulcus; ias, inferior arcuate sulcus; ps, principal sulcus; sas, superior arcuate sulcus.

The online version of this article includes the following figure supplement(s) for figure 8:

**Figure supplement 1.** Statistically testable borders (pink lines) confirmed by the quantitative analysis for the caudal ventrolateral area 12l and dorsally adjacent area 45A.

**Figure supplement 2.** Statistically testable borders (pink lines) confirmed by the quantitative analysis for the caudal ventrolateral cortex; areas 8Av, 45B, and 44.

belonging to same area) as well as the most prominent connectivity correlation patterns of each area. Indeed, the results of the analysis of the functional correlation of each identified frontal area with a total of 138 areas of the prefrontal, cingulate, premotor, motor, somatosensory, parietal, and occipital cortex, previously identified by our group. Whereas a parcellation of the temporal cortex comes from Lyon atlas of Kennedy and colleagues (*Markov et al., 2014*). Connectivity patterns of prefrontal areas (including their intra-areal correlations) are depicted in *Figures 12–15*. In addition, same schematic summary of functional connectivity results for premotor and motor areas is shown in *Figures 16–18*.

### Areas 10

Lateral frontopolar areas 10d and 10o present more restricted functional connectivity pattern than medial areas 10md and 10mv (*Figure 12*), apart from the weak correlation between 10d and areas a46d and a46v. Contrary, medial areas 10md and 10mv share strong connectivity with cingulate cortex, that is, dorsally located area 10md with p32, while ventral area 10mv was correlated with s32 and to a lesser extent with p32. Further differences are found since 10mv is strongly correlated to orbital area 14r, while this is not case with 10md. In contrast, 10md has connectivity with dorsal and lateral PFC. Within the frontal polar region, dorsal areas 10d and 10md are more strongly correlated to each other than to their ventral counterparts, which are also strongly connected to each other (*Figure 12*).

### Areas 14

Rostral area 14r has more prominent functional correlation with medial PFC (area 10mv) and anterior cingulate cortex (ACC) than with caudally located area 14c, which is strongly correlated with caudal orbital (area 13a) and rostral cingulate area 25. Subdivisions of area 14 show weaker connectivity among each other than to their corresponding adjacent areas (*Figure 12*).

### Areas 11

Subdivisions of area 11 displayed strong functional connectivity to each other and to their surrounding areas, that is, 11l and its laterally neighbouring areas 12r, 12m, and 12o, whereas area 11m was more strongly correlated with medially adjacent area 13b, and to a lesser extent with area 13l. Finally, both areas revealed connectivity with ventrolateral area 45A (*Figure 12*).

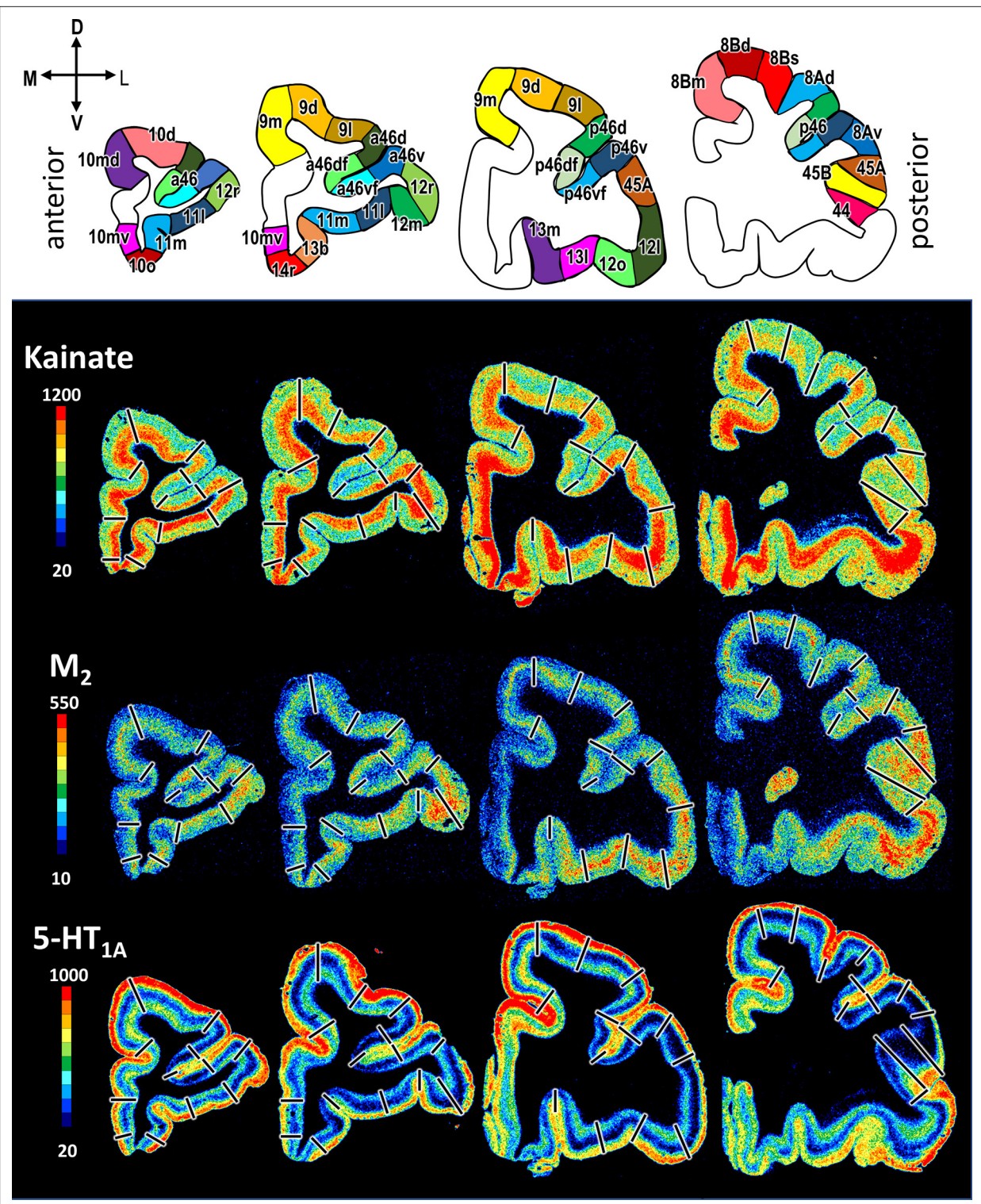

**Figure 9.** Exemplary sections depicting the distribution of kainate, M₂ and 5-HT₁ₐ receptors in coronal sections through a macaque hemisphere. The colour bar, positioned left to the autoradiographs, codes receptor densities in fmol/mg protein, and borders are indicated by black lines. The four schematic drawings at the top represent the distinct rostro-caudal levels and show the position of all prefrontal areas defined. C, caudal; D, dorsal; R, rostral; V, ventral.

The online version of this article includes the following figure supplement(s) for figure 9:

*Figure 9 continued on next page*

*Figure 9 continued*

**Figure supplement 1.** Exemplary sections depicting the distribution of the remaining receptor types, that is, of glutamate (AMPA, kainate, NMDA) and gamma-aminobutyric acid (GABA) (GABA$_A$, GABA$_B$, GABA$_A$-associated benzodiazepine binding sites – BZ) receptors, in coronal sections through a macaque hemisphere.

**Figure supplement 2.** Exemplary sections depicting the distribution of the remaining receptor types, that is, acetylcholine (M1, M2, M3) and noradrenalin ($\alpha$1, $\alpha$2) receptors in coronal sections through a macaque hemisphere.

**Figure supplement 3.** Exemplary sections depicting the distribution of the remaining receptor types, that is, serotonin (5HT$_2$) and dopamine (D$_1$) receptors in coronal sections through a macaque hemisphere.

### Areas 13

Among subdivisions of area 13, we found that areas 13a and 13m have most restricted connectivity pattern, whereby most rostral area 13b and laterally positioned area 13l show opposite trend. Interestingly, area 13a revealed weakest interconnectivity to 13l, but rather strong connections to adjoining areas 13b and 14c, whereby the strongest connectivity for area 13l is found to be with surrounding areas 13m and 12o. Additionally, area 13l revealed connectivity to posterior prefrontal region, in particular to areas 12l, 45A, p46d, and p46v (*Figure 12*).

### Areas 12

Within the orbitofrontal region, subdivisions of area 12 presented a widespread functional connectivity pattern. This was particularly true for area 12r, which showed strong correlation to lateral areas 46, ventral areas 45A and 45B, as well as a correlation, although weaker, with premotor areas F5 and temporal polysensory areas STPi, PBc, and LB. Interareal connectivity pattern showed a weak correlation between area 12l and the rest of the area 12, which share strong functional connectivity among each other. In contrast, the strongest connections of 12l are found with areas 45A, 13l, and p46v (*Figure 12*).

### Areas 9 and 8B

On the dorsolateral prefrontal cortex, rostro-caudal differences can be recognized between functional connectivity pattern of areas 9m, 9d, and 9l rostrally, and more caudally located areas 8Bm, 8Bd, and 8Bs, which displayed a more widespread connectivity pattern with various distinct areas in the prefrontal, pre(motor), parietal, medial occipital, and temporal cortex (*Figure 13*). While dorsal and lateral subdivisions of areas 9 and 8B are strongly intercorrelated, medial areas 9m and 8Bm showed a stronger connection to their medial neighbouring areas, that is, 9m to its adjacent cingulate area 24c, and 8Bm to surrounding areas a24'c and F6. Among all subdivisions of area 9, only medial area 9m shows functional connectivity with premotor cortex, in particular areas F6, F3, F2v, and F5s. Connectivity pattern of area 9d is restricted within prefrontal region; this is not true for 9m and 9l, which revealed connectivity with parietal area Opt and temporal areas STPr and STPi. Moreover, area 9m is rather correlated to anterior and mid-cingulate areas, whereas 9d has connection to posterior cingulate area d23a/b. All subdivisions of area 8B share strong functional connectivity with their surrounding prefrontal areas, parietal area Opt, and premotor areas F6 and F7. But opposite is found in regard to their connectivity with frontopolar and orbital areas. Additionally, only area 8Bd did not show connectivity with temporal areas. On the other hand, area 8Bs revealed functional connectivity with primary motor cortex, that is, areas 4a and 4m, as well as with transitional somatosensory area TSA and medial occipital region, that is, areas V6Adm and V6Avm (*Figure 13*).

### Areas 46

Rostro-caudal differences in functional connectivity patterns were also found for the subdivisions of lateral prefrontal area 46, whereby posterior subdivisions showed a more widespread connectivity pattern across the brain. Within the *ps*, the anterior and posterior subdivisions of area 46 have a similar intraregional organization. Specifically, while dorsal subdivisions have strong connection to each other, as well as with areas '46vf,' most ventrally located areas a46v and p46v revealed to have stronger connection to their counterparts '46vf' than with corresponding dorsal subdivisions. Interestingly, connectivity between areas '46v' and dorsal areas 46 is weaker in the rostral than in the caudal portion of the *ps*. Correlation with parietal areas Opt and LIP, and temporal STP areas is noticed

**Table 3.** Absolute receptor densities (mean ± SD) in fmol/mg protein.
BZ, GABA$_A$-associated benzodiazepine binding sites.

| Area | AMPA | Kainate | NMDA | GABA$_A$ | GABA$_B$ | BZ | M$_1$ | M$_2$ | M$_3$ | $\alpha_1$ | $\alpha_2$ | 5-HT$_{1A}$ | 5-HT$_2$ | D$_1$ |
|---|---|---|---|---|---|---|---|---|---|---|---|---|---|---|
| 10d | 591 | 858 | 1430 | 1697 | 1970 | 2151 | 995 | 141 | 880 | 507 | 337 | 623 | 340 | 93 |
| SD | 161 | 116 | 260 | 162 | 542 | 829 | 230 | 35 | 117 | 75 | 68 | 169 | 75 | 20 |
| 10md | 586 | 895 | 1470 | 1651 | 2095 | 2307 | 1012 | 154 | 856 | 494 | 327 | 628 | 357 | 90 |
| SD | 106 | 90 | 177 | 168 | 495 | 783 | 274 | 45 | 112 | 48 | 48 | 151 | 60 | 20 |
| 10mv | 628 | 903 | 1612 | 1680 | 2254 | 2451 | 1063 | 145 | 894 | 471 | 334 | 666 | 320 | 86 |
| SD | 130 | 66 | 151 | 199 | 606 | 839 | 332 | 35 | 124 | 94 | 56 | 214 | 67 | 18 |
| 10o | 569 | 909 | 1523 | 1723 | 2336 | 2327 | 1068 | 150 | 923 | 470 | 342 | 682 | 350 | 82 |
| SD | 76 | 50 | 190 | 160 | 612 | 774 | 313 | 45 | 105 | 76 | 76 | 233 | 59 | 12 |
| 14r | 470 | 818 | 1442 | 1427 | 2482 | 1715 | 921 | 134 | 833 | 497 | 297 | 583 | 323 | 86 |
| SD | 81 | 107 | 255 | 162 | 424 | 542 | 385 | 35 | 118 | 109 | 95 | 119 | 44 | 15 |
| 11m | 604 | 771 | 1585 | 1762 | 2476 | 1975 | 1094 | 159 | 965 | 473 | 342 | 549 | 357 | 92 |
| SD | 100 | 65 | 139 | 142 | 466 | 218 | 200 | 64 | 132 | 50 | 40 | 167 | 60 | 27 |
| 11l | 623 | 807 | 1562 | 1876 | 2644 | 2066 | 1050 | 159 | 944 | 462 | 351 | 529 | 357 | 96 |
| SD | 111 | 123 | 113 | 235 | 478 | 247 | 228 | 54 | 101 | 46 | 45 | 116 | 51 | 29 |
| 13b | 489 | 820 | 1548 | 1615 | 2311 | 1901 | 1039 | 166 | 897 | 480 | 350 | 562 | 355 | 93 |
| SD | 44 | 103 | 223 | 120 | 452 | 431 | 263 | 57 | 104 | 73 | 75 | 206 | 57 | 22 |
| 13m | 753 | 856 | 1499 | 1622 | 1908 | 1864 | 1059 | 206 | 918 | 485 | 417 | 527 | 357 | 78 |
| SD | 67 | 111 | 122 | 126 | 429 | 269 | 121 | 94 | 130 | 21 | 21 | 138 | 50 | 11 |
| 13l | 713 | 756 | 1498 | 1683 | 2057 | 2052 | 1054 | 223 | 826 | 461 | 404 | 460 | 351 | 70 |
| SD | 95 | 60 | 187 | 180 | 240 | 303 | 148 | 78 | 108 | 15 | 26 | 107 | 43 | 4 |
| 12r | 659 | 854 | 1406 | 1843 | 2412 | 1991 | 1026 | 180 | 922 | 439 | 306 | 540 | 350 | 86 |
| SD | 122 | 120 | 121 | 283 | 312 | 307 | 301 | 72 | 96 | 38 | 52 | 88 | 51 | 9 |
| 12m | 598 | 799 | 1533 | 1792 | 2222 | 1873 | 1152 | 202 | 918 | 481 | 379 | 504 | 354 | 86 |
| SD | 136 | 55 | 175 | 246 | 353 | 421 | 262 | 74 | 108 | 48 | 71 | 103 | 45 | 22 |
| 12l | 630 | 840 | 1400 | 1494 | 2010 | 1789 | 824 | 182 | 780 | 491 | 320 | 531 | 351 | 71 |
| SD | 112 | 73 | 126 | 221 | 483 | 417 | 347 | 75 | 132 | 82 | 43 | 163 | 48 | 6 |
| 12o | 670 | 817 | 1527 | 1579 | 2142 | 2102 | 888 | 209 | 832 | 484 | 401 | 541 | 384 | 89 |
| SD | 165 | 97 | 158 | 267 | 414 | 436 | 174 | 64 | 149 | 32 | 66 | 87 | 61 | 20 |
| 9m | 607 | 818 | 1224 | 1460 | 2048 | 1864 | 868 | 168 | 760 | 508 | 307 | 629 | 359 | 89 |
| SD | 125 | 84 | 252 | 352 | 235 | 449 | 196 | 33 | 79 | 50 | 49 | 136 | 55 | 22 |
| 9d | 584 | 766 | 1341 | 1633 | 2312 | 2081 | 1050 | 176 | 841 | 515 | 355 | 642 | 362 | 92 |
| SD | 154 | 72 | 206 | 338 | 235 | 478 | 177 | 34 | 80 | 40 | 59 | 81 | 61 | 24 |
| 9l | 554 | 711 | 1311 | 1582 | 2173 | 1972 | 1029 | 164 | 822 | 497 | 361 | 594 | 366 | 67 |
| SD | 151 | 56 | 230 | 324 | 260 | 464 | 143 | 31 | 91 | 38 | 47 | 64 | 54 | 21 |
| a46d | 527 | 810 | 1247 | 1609 | 1993 | 1821 | 981 | 187 | 819 | 462 | 318 | 521 | 354 | 90 |
| SD | 138 | 81 | 197 | 253 | 189 | 349 | 234 | 40 | 114 | 68 | 65 | 86 | 66 | 26 |
| a46df | 559 | 667 | 1348 | 1663 | 2071 | 1898 | 1083 | 176 | 860 | 478 | 384 | 466 | 355 | 94 |
| SD | 126 | 44 | 124 | 219 | 170 | 444 | 160 | 45 | 79 | 60 | 61 | 94 | 80 | 29 |
| a46vf | 619 | 679 | 1427 | 1752 | 2291 | 1873 | 1124 | 180 | 894 | 484 | 395 | 497 | 376 | 93 |
| SD | 126 | 81 | 102 | 297 | 280 | 352 | 161 | 47 | 94 | 47 | 39 | 88 | 76 | 30 |
| a46v | 502 | 808 | 1339 | 1614 | 2068 | 1908 | 1017 | 187 | 856 | 440 | 319 | 496 | 349 | 87 |
| SD | 67 | 61 | 167 | 281 | 200 | 406 | 235 | 52 | 85 | 52 | 35 | 79 | 58 | 17 |
| p46d | 563 | 785 | 1187 | 1449 | 1934 | 1786 | 889 | 185 | 771 | 439 | 300 | 484 | 364 | 81 |
| SD | 103 | 50 | 318 | 259 | 231 | 286 | 257 | 48 | 84 | 70 | 30 | 77 | 35 | 29 |
| p46df | 592 | 692 | 1305 | 1649 | 2049 | 1978 | 1000 | 176 | 812 | 453 | 388 | 478 | 373 | 85 |
| SD | 102 | 40 | 254 | 268 | 177 | 256 | 241 | 43 | 84 | 78 | 47 | 86 | 42 | 22 |
| p46vf | 613 | 671 | 1369 | 1726 | 2295 | 2138 | 998 | 163 | 834 | 467 | 395 | 528 | 381 | 88 |
| SD | 115 | 71 | 225 | 315 | 315 | 383 | 230 | 41 | 115 | 74 | 67 | 107 | 48 | 24 |
| p46v | 519 | 758 | 1241 | 1444 | 1956 | 1814 | 810 | 170 | 783 | 416 | 321 | 461 | 361 | 81 |
| SD | 49 | 67 | 207 | 279 | 213 | 284 | 294 | 34 | 74 | 88 | 43 | 98 | 43 | 23 |
| 8Bm | 528 | 731 | 1018 | 1216 | 1888 | 1958 | 806 | 178 | 667 | 472 | 273 | 508 | 351 | 83 |
| SD | 136 | 128 | 438 | 217 | 267 | 236 | 173 | 31 | 87 | 70 | 49 | 80 | 32 | 27 |

*Table 3 continued on next page*

*Table 3 continued*

| Area | AMPA | Kainate | NMDA | GABA$_A$ | GABA$_B$ | BZ | M$_1$ | M$_2$ | M$_3$ | $\alpha_1$ | $\alpha_2$ | 5-HT$_{1A}$ | 5-HT$_2$ | D$_1$ |
|------|------|---------|------|----------|----------|------|------|------|------|------|------|------|------|------|
| 8Bd | 481 | 641 | 973 | 1195 | 1896 | 2136 | 832 | 195 | 680 | 466 | 263 | 437 | 362 | 89 |
| SD | 92 | 106 | 346 | 151 | 173 | 385 | 131 | 41 | 92 | 73 | 70 | 89 | 47 | 28 |
| 8Bs | 494 | 570 | 1047 | 1232 | 1901 | 1931 | 831 | 164 | 682 | 436 | 304 | 484 | 356 | 88 |
| SD | 99 | 54 | 348 | 209 | 389 | 134 | 117 | 47 | 117 | 75 | 67 | 106 | 56 | 23 |
| 8Ad | 528 | 694 | 1108 | 1219 | 1972 | 1795 | 870 | 158 | 685 | 438 | 272 | 450 | 359 | 82 |
| SD | 115 | 65 | 322 | 200 | 143 | 301 | 227 | 37 | 139 | 67 | 36 | 82 | 43 | 29 |
| 8Av | 440 | 591 | 1017 | 1205 | 1703 | 1807 | 708 | 163 | 603 | 347 | 257 | 262 | 323 | 79 |
| SD | 94 | 102 | 264 | 202 | 264 | 369 | 268 | 36 | 174 | 112 | 64 | 109 | 67 | 25 |
| 45A | 550 | 733 | 1235 | 1461 | 1846 | 1810 | 880 | 168 | 734 | 422 | 321 | 394 | 358 | 75 |
| SD | 97 | 61 | 165 | 186 | 280 | 378 | 244 | 52 | 62 | 106 | 47 | 126 | 47 | 19 |
| 45B | 601 | 588 | 1310 | 1472 | 1955 | 1911 | 972 | 147 | 705 | 442 | 372 | 499 | 378 | 88 |
| SD | 150 | 54 | 271 | 286 | 301 | 249 | 317 | 29 | 120 | 65 | 75 | 166 | 58 | 30 |
| 44 | 595 | 592 | 1310 | 1520 | 2065 | 1756 | 957 | 154 | 697 | 475 | 402 | 638 | 385 | 93 |
| SD | 162 | 86 | 277 | 220 | 233 | 294 | 339 | 22 | 164 | 79 | 70 | 253 | 57 | 27 |

throughout areas 46; however, these connections are particularly strong for 'p46' areas. Finally, areas 'p46d' show connectivity with primary motor cortex and somatosensory areas TSA and 3bm, which is not case with areas 'p46v' (*Figure 14*).

## Areas 8A, 44, and 45

Within the most posterior portion of the lateral prefrontal cortex, areas 8Ad and 8Av revealed widespread connectivity pattern with region around *ias*, as well as with the cingulate, temporal, somatosensory, and parietal cortex (*Figure 15*). While both areas express similar connectivity pattern across cortex, we found that area 8Ad was more strongly connected with prefrontal area 8Bs and parietal area Opt. In contrast, area 8Av revealed stronger connection with prefrontal areas 45B and 44, as well as premotor area F4s and temporal TPt. Ventrolateral areas 45A and 45B have strong interconnection to each other, as well as to surrounding prefrontal and premotor areas (*Figure 15*). However, while 45B has widespread connectivity throughout the medial and inferior parietal cortex, this was not true for 45A. Instead, we found that area 45A has rather strong correlation with numerous orbital areas. Unlike areas 45, the more posteriorly located area 44 does not show strong correlation with auditory core region within the temporal cortex, but exhibits a wider connectivity pattern which also includes somatosensory cortex (i.e. areas 3al, 3bl, and 3bm) and primary motor area 4p (*Figure 15*).

## Premotor areas

Medial premotor areas F6 and F3 have strongest connectivity with each other and their respective adjacent areas, that is, F6 with prefrontal area 8Bm and F3 with primary motor area 4m. In general, both areas revealed to have widespread functional connectivity across the brain. Concretely, with the posterior prefrontal, lateral premotor, cingulate, and parietal areas, but connections of posterior area F3 are more extensive across primary motor, somatosensory, and temporal region than F6 (*Figure 16*).

All subdivisions of area F7 revealed to have strong connection with surrounding premotor areas and posterior prefrontal areas 8B, 8A, and 'p46.' While strongest connection is shown between F7d and F7i, the weakest one is noticed between F7d and F7s. Interestingly, most dorsal area F7d showed most restricted connectivity pattern, while opposite was true for most lateral area F7s, located on the dorsal wall of the *ias*. This area displayed widespread connectivity across primary motor, somatosensory, parietal, and temporal cortex (*Figure 16*). Caudally neighbouring to areas F7, on the dorsal premotor cortex, subdivisions of area F2 have relatively strong connection to each other, but the strongest connection of F2v was rather displayed with adjacent areas, located within the spur of the arcuate sulcus, F7s and F4s. Also, connectivity pattern of F2v is more widespread across cingulate, parietal, and temporal regions than of F2d. Finally, only F2v revealed connection with somatosensory cortex, that is, areas TSA, 2 and 3bl (*Figure 16*).

Similar to connectivity trend shown in dorsal counterparts, subdivisions of areas F5 and F4, located within the arcuate sulcus (i.e. F5s and F4s respectively), displayed more extensive connectivity patterns

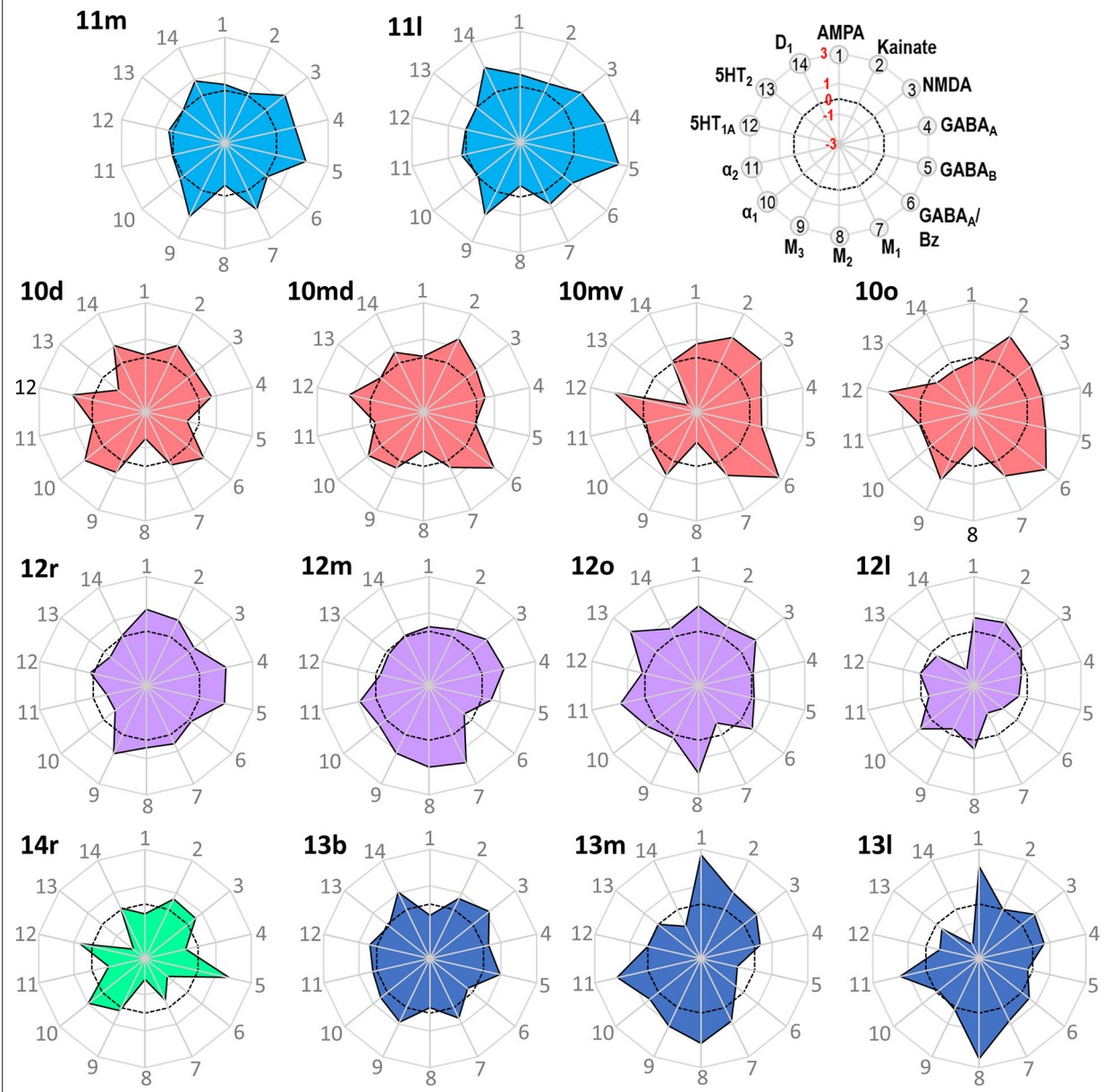

**Figure 10.** Normalized receptor fingerprints of the frontopolar and orbital areas. Black dotted line on the plot represents the mean value over all areas for each receptor. Receptors displaying a negative z-score are indicative of absolute receptor densities which are lower than the average of that specific receptor over all examined areas. The opposite is true for positive z-scores. Labelling of different receptor types, as well as the axis scaling, is identical for each area plot, which is specified in the polar plot on the top of the figure.

The online version of this article includes the following figure supplement(s) for figure 10:

**Figure supplement 1.** Receptor fingerprints of the frontopolar and orbital areas.

compared to their respective subdivisions on the ventral premotor surface (*Figure 17*). While areas F5s and F4s have strong correlation to their respective dorsal subdivisions F5d and F4d, connectivity to the ventral subdivisions is weaker; this is particularly true for correlation between F4s and F4v. Interestingly, we found correlation between F5v and auditory core region within the temporal lobe. Also, we found strong correlation between primary area 4p and ventral premotor region, which was the strongest for areas F4d and F4v (*Figure 17*).

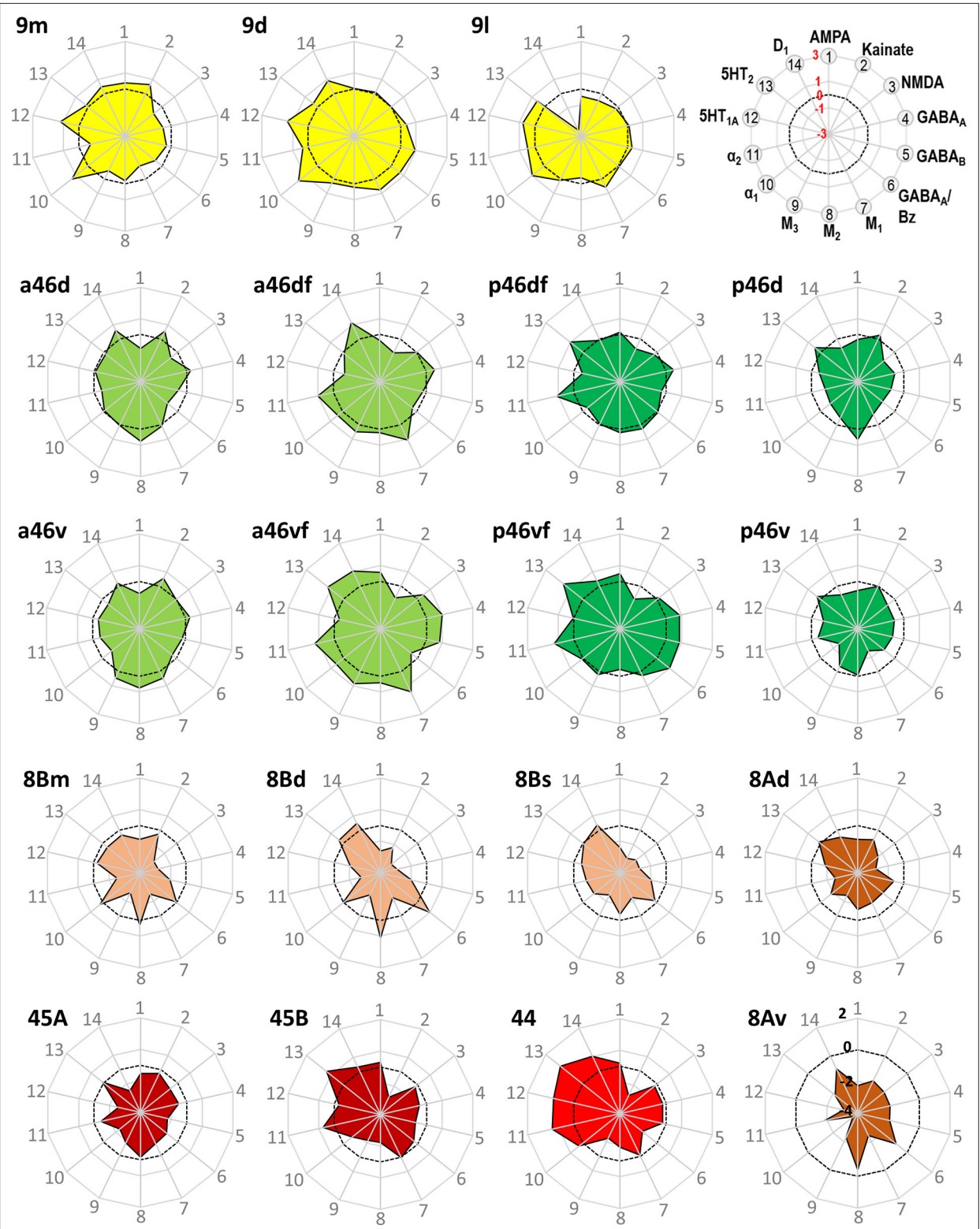

**Figure 11.** Normalized receptor fingerprints of the medial, dorsolateral, lateral, and ventrolateral areas. Black dotted line on the plot represents the mean value over all areas for each receptor. Receptors displaying a negative z-score are indicative of absolute receptor densities which are lower than the average of that specific receptor over all examined areas. The opposite is true for positive z-scores. Labelling of different receptor types, as well as the axis scaling, is identical for each area plot, which is specified in the polar plot on the top of the figure. Due to the low receptor densities measured in area 8Av, scaling for its fingerprint is adjusted and shown directly on the corresponding polar plot.

*Figure 11 continued on next page*

*Figure 11 continued*

The online version of this article includes the following figure supplement(s) for figure 11:

**Figure supplement 1.** Receptor fingerprints of the medial, dorsolateral, lateral, and ventrolateral areas.

## Primary motor areas

Subdivisions of area 4 have the strongest correlation with surrounding areas of premotor and somatosensory cortex. In particular, area 4m with medial premotor area F3 and somatosensory 3am and 3bm; area 4a with dorsal premotor areas F2d and F2v and most posterior area 4p with ventral premotor (F4d and F4v) and somatosensory areas 1, 3al, and 3bl. Additionally, 4p has strong correlation with rostral areas PF, PFG, and PFop of the inferior parietal lobule, as well as with intraparietal area AIP. In general, primary motor region revealed widespread connectivity with posterior prefrontal and cingulate areas, but also with parietal and temporal cortex (*Figure 18*).

## Hierarchical clustering and principal component analyses

The hierarchical cluster analysis (*Figure 19*) revealed differences in size of receptor fingerprints between areas occupying its most rostral portion (found in clusters 1 and 2) from the more caudally positioned prefrontal areas and (pre)motor areas (found in clusters 3–5). The five main clusters, which were identified by the k-means analysis, are mostly composed of neighbouring areas, but also group areas that do not share common borders and occupy different regions of the hemisphere.

- **Cluster 1** is the largest cluster and encompasses the most rostrally located prefrontal areas. It includes all subdivisions of frontal polar cortex (10d, 10md, 10mv, 10o), all subdivisions of area 46 located in the depth of *ps* (a46df, a46vf, p46df, p46vf), anterior subdivisions of 46 located on the shoulders of *ps* (a46d, a46v), rostral orbital areas (11m, 11l, 13b, 14r, 12r, 12m), as well as dorsal and medial areas 9d and 9m. In particular, areas a46df and a46vf are more similar to medial area 12m than to their posterior counterparts (p46df and p46vf, respectively), while areas a46d and a46v showed a greater similarity with orbital areas 13b and 14r and with mediodorsal areas 9m and 9d. Additionally, rostral orbital areas 11m and 11l grouped with laterally adjacent area 12r.
- **Cluster 2** is constituted of the posterior orbital areas 13m, 13l, 12o, and 12l, as well as dorsal area 9l and premotor area F5v.
- **Cluster 3** encompasses areas positioned most posteriorly in the prefrontal cortex (8Bd, 8Bm, 8Bs, 8Ad, p46d, p46v, 45A, 45B, 44) and premotor areas (F3, F6, F2d, F7d, F5d, F5s). The subdivisions of area 8B grouped closely with area 8Ad, which displayed the highest similarity to the medially adjacent area 8Bs, and premotor area F7d, which is the most similar to 8Bd. Medial premotor areas F6 and F3 clustered with caudal premotor area F2d, whereas the posterior subdivisions of area 46 located on the shoulder of the *ps*, areas p46d and p46v, did so with area 45A. Areas 45B and 44 share most similarities in receptor densities with the adjacent premotor areas F5s and F5d.
- **Cluster 4** comprises all subdivisions in and the around the spur of the arcuate sulcus, that is, prefrontal area 8Av and premotor areas F7s, F7i, F2v, F4s, F4d, and F4v.
- **Cluster 5** is the most homogeneous cluster of all since it consists of subdivisions of the primary motor cortex, areas 4p, 4a, and 4m.

A principal component analysis was carried out to reduce the 14-dimensional space resulting from the analysis of 14 different receptors area to a 2-dimensional plot (*Figure 20*). Differences in the first principal component revealed a rostro-caudal trend driven by the gradual decrease in size of the receptor fingerprints. Consequently, subdivisions of area 4 (4m, 4a, and 4p) are segregated from the rest of the frontal areas since their fingerprints are the smallest among all analysed areas (present data, *Rapan et al., 2021*). In contrast, areas of clusters 1 and 2 present the highest receptor concentration levels. The second principal component further segregated primary motor areas (cluster 5) from the premotor ones (clusters 4 and 3), as well as rostral prefrontal areas (clusters 1 and 2) from the posterior ones (cluster 3) (*Figure 20*). The first and second principal components did not segregate areas located in clusters 1 and 2.

**Table 4.** FDR-corrected p-values for the post hoc tests (i.e. third-level tests; p-values were corrected for 258 comparisons per receptor type).

No p-values are provided for the $M_1$, $M_2$, 5-HT$_2$, or $D_1$ receptors because they did not reach the level of significance in the second-level test. Green background highlights significant pairs of adjacent prefrontal areas in the macaque brain. *p<0.05, **p<0.01, ***p<0.001.

| | AMPA | Kainate | NMDA | GABA$_A$ | GABA$_B$ | BZ | M$_3$ | $\alpha_1$ | $\alpha_2$ | 5-HT$_{1A}$ |
|---|---|---|---|---|---|---|---|---|---|---|
| 10d - 10md | 0.9393 | 0.5591 | 0.8028 | 0.8776 | 0.6976 | 0.7871 | 0.7553 | 0.9104 | 0.866 | 0.9753 |
| 10d - 9d | 0.9041 | 0.1142 | 0.5721 | 0.8364 | 0.1413 | 0.8728 | 0.6135 | 0.9104 | 0.5692 | 0.9081 |
| 10d - 9l | 0.618 | 0.0091** | 0.4329 | 0.5871 | 0.4474 | 0.7277 | 0.4173 | 0.9549 | 0.4603 | 0.746 |
| 10d - a46d | 0.3472 | 0.4435 | 0.194 | 0.7085 | 0.9711 | 0.3149 | 0.3845 | 0.5571 | 0.6929 | 0.1842 |
| 10md - 10mv | 0.6304 | 0.8867 | 0.3033 | 0.9407 | 0.4908 | 0.8415 | 0.586 | 0.7554 | 0.8435 | 0.7195 |
| 10md - 9m | 0.8231 | 0.1508 | 0.0461* | 0.1826 | 0.8701 | 0.1242 | 0.1456 | 0.8313 | 0.5417 | 0.9816 |
| 10mv - 10o | 0.4391 | 0.9801 | 0.5458 | 0.8064 | 0.7872 | 0.8587 | 0.7441 | 0.9973 | 0.8276 | 0.9081 |
| 10mv - 14r | 0.0386* | 0.149 | 0.2167 | 0.1305 | 0.3529 | 0.0291* | 0.3522 | 0.7752 | 0.2444 | 0.3143 |
| 10o - 11m | 0.7018 | 0.0056** | 0.6676 | 0.8425 | 0.5291 | 0.2996 | 0.5396 | 0.9549 | 0.9936 | 0.0666 |
| 10o - 14r | 0.168 | 0.1227 | 0.5525 | 0.0366* | 0.5751 | 0.0291* | 0.1793 | 0.7645 | 0.1471 | 0.2115 |
| 11l - 11m | 0.8207 | 0.5126 | 0.8931 | 0.4519 | 0.4832 | 0.8721 | 0.7807 | 0.9104 | 0.7881 | 0.8618 |
| 11l - 12m | 0.8207 | 0.9666 | 0.9271 | 0.7045 | 0.058 | 0.7409 | 0.7964 | 0.8085 | 0.3854 | 0.8686 |
| 11l - 12r | 0.5848 | 0.3727 | 0.2291 | 0.8932 | 0.2739 | 0.8721 | 0.7446 | 0.7645 | 0.1325 | 0.917 |
| 11l - 13l | 0.2408 | 0.6732 | 0.9223 | 0.4866 | 0.0523 | 0.9766 | 0.2814 | 0.9549 | 0.1427 | 0.7352 |
| 11l - 13m | 0.1005 | 0.4105 | 0.9256 | 0.3035 | 0.0104* | 0.8678 | 0.9487 | 0.7645 | 0.0781 | 0.9081 |
| 11m - 13b | 0.0988 | 0.3998 | 0.8028 | 0.3063 | 0.4593 | 0.8728 | 0.2991 | 0.9549 | 0.8403 | 0.917 |
| 11m - 13l | 0.1593 | 0.9801 | 0.8261 | 0.8911 | 0.208 | 0.8652 | 0.19 | 0.9795 | 0.0925 | 0.6198 |
| 11m - 13m | 0.06 | 0.1684 | 0.8261 | 0.698 | 0.0554 | 0.9766 | 0.7943 | 0.8541 | 0.0465* | 0.997 |
| 11m - 14r | 0.0688 | 0.4928 | 0.2895 | 0.0159* | 0.9809 | 0.489 | 0.0347* | 0.812 | 0.149 | 0.8153 |
| 12l - 12o | 0.7396 | 0.7221 | 0.5477 | 0.7785 | 0.7117 | 0.5449 | 0.591 | 0.9338 | 0.0323* | 0.9837 |
| 12l - 12r | 0.7423 | 0.84 | 0.9808 | 0.0261* | 0.0824 | 0.7523 | 0.0613 | 0.3864 | 0.6495 | 0.9869 |
| 12l - 45A | 0.2779 | 0.0773 | 0.2152 | 0.8729 | 0.4924 | 0.9984 | 0.5606 | 0.148 | 0.9231 | 0.0933 |
| 12m - 12o | 0.4391 | 0.8664 | 0.9223 | 0.1851 | 0.7851 | 0.7313 | 0.2335 | 0.9973 | 0.6306 | 0.7877 |
| 12m - 12r | 0.4191 | 0.3735 | 0.3465 | 0.8326 | 0.4936 | 0.8721 | 0.9602 | 0.5104 | 0.0176* | 0.7772 |
| 12m - 13l | 0.1742 | 0.7207 | 0.9867 | 0.7785 | 0.7649 | 0.7496 | 0.4295 | 0.929 | 0.5069 | 0.8618 |
| 12o - 12r | 0.9669 | 0.5144 | 0.4782 | 0.0923 | 0.2583 | 0.8587 | 0.2335 | 0.5575 | 0.004** | 0.9881 |
| 12o - 13l | 0.5736 | 0.6021 | 0.9649 | 0.5306 | 0.9429 | 0.9901 | 0.9049 | 0.929 | 0.8128 | 0.6789 |
| 12r - a46v | 0.0151* | 0.3743 | 0.6393 | 0.0962 | 0.0824 | 0.8738 | 0.3415 | 0.9973 | 0.7023 | 0.6442 |
| 12r - p46v | 0.0427* | 0.0659 | 0.2246 | 0.0032** | 0.019* | 0.7409 | 0.0347* | 0.7253 | 0.6634 | 0.3438 |
| 13b - 14r | 0.8536 | 0.973 | 0.4654 | 0.2172 | 0.4936 | 0.7277 | 0.339 | 0.88 | 0.1052 | 0.9081 |
| 13l - 13m | 0.7624 | 0.2909 | 0.9979 | 0.8563 | 0.7452 | 0.8587 | 0.4298 | 0.8565 | 0.8354 | 0.6937 |
| 44A - 45B | 0.9416 | 0.9648 | 0.9808 | 0.8425 | 0.677 | 0.8415 | 0.933 | 0.6727 | 0.4447 | 0.089 |
| 45A - 45B | 0.5714 | 0.0122* | 0.6278 | 0.97 | 0.7593 | 0.8721 | 0.7363 | 0.7902 | 0.1275 | 0.2574 |
| 45A - 8Av | 0.0988 | 0.0062** | 0.095 | 0.1219 | 0.5291 | 0.9928 | 0.0476* | 0.0857 | 0.0401 | 0.0853 |

*Table 4 continued on next page*

*Table 4 continued*

| | AMPA | Kainate | NMDA | GABA$_A$ | GABA$_B$ | BZ | M$_3$ | α$_1$ | α$_2$ | 5-HT$_{1A}$ |
|---|---|---|---|---|---|---|---|---|---|---|
| 45A - p46v | 0.7274 | 0.6956 | 0.9363 | 0.9604 | 0.6792 | 0.9901 | 0.4861 | 0.9549 | 0.9686 | 0.4794 |
| 45B - 8Av | **0.0335*** | 0.9801 | **0.0327*** | 0.0914 | 0.3129 | 0.8721 | 0.1754 | **0.0238*** | **0.0004**** | **0.0016**** |
| 8Ad - 8Av | 0.2009 | **0.0487*** | 0.5458 | 0.9852 | 0.1933 | 0.9897 | 0.2412 | **0.0155*** | 0.6929 | **0.0073**** |
| 8Ad - 8Bs | 0.7142 | **0.0183*** | 0.7149 | 0.9407 | 0.8209 | 0.7836 | 0.9833 | 0.9978 | 0.2807 | 0.7062 |
| 8Ad - p46d | 0.6185 | 0.0705 | 0.5546 | 0.123 | 0.9152 | 0.9984 | 0.2024 | 0.9795 | 0.358 | 0.7062 |
| 8Av - p46v | 0.2667 | **0.0009***** | 0.0726 | 0.1047 | 0.2099 | 0.9915 | **0.0036**** | 0.1038 | **0.0344*** | **0.0043**** |
| 8Bd - 8Bm | 0.6165 | 0.1226 | 0.7936 | 0.9194 | 0.9698 | 0.7409 | 0.9038 | 0.937 | 0.8403 | 0.4665 |
| 8Bd - 8Bs | 0.8684 | 0.2213 | 0.6066 | 0.8663 | 0.968 | 0.7386 | 0.9602 | 0.7048 | 0.2297 | 0.6243 |
| 8Bd - 9d | 0.1213 | **0.0168*** | **0.0031**** | **0.0011**** | **0.0469*** | 0.9557 | **0.0155*** | 0.3477 | **0.0044**** | **0.004**** |
| 8Bm - 9m | 0.2744 | 0.1202 | 0.1303 | 0.115 | 0.5171 | 0.9071 | 0.1863 | 0.5663 | 0.2868 | 0.1364 |
| 8Bs - 9l | 0.385 | **0.0058**** | **0.0364*** | **0.0083**** | 0.2099 | 0.9766 | **0.0362*** | 0.1957 | 0.084 | 0.1598 |
| 9d - 9l | 0.6967 | 0.3221 | 0.8516 | 0.7657 | 0.5923 | 0.8587 | 0.7964 | 0.8085 | 0.8602 | 0.6144 |
| 9d - 9m | 0.7704 | 0.3551 | 0.3881 | 0.2172 | 0.2099 | 0.6636 | 0.2048 | 0.9384 | 0.1121 | 0.9095 |
| 9l - a46d | 0.7246 | 0.054 | 0.6553 | 0.8908 | 0.4226 | 0.7544 | 0.9602 | 0.5726 | 0.1595 | 0.3769 |
| a46df - a46d | 0.6801 | **0.004**** | 0.4699 | 0.7705 | 0.7808 | 0.8728 | 0.5621 | 0.833 | **0.0257*** | 0.5572 |
| a46df - a46vf | 0.3688 | 0.8465 | 0.5764 | 0.5843 | 0.3129 | 0.9857 | 0.6279 | 0.9549 | 0.747 | 0.7573 |
| a46df-p46df | 0.6714 | 0.6574 | 0.7815 | 0.9612 | 0.9519 | 0.8721 | 0.528 | 0.6964 | 0.9208 | 0.9138 |
| a46d-p46d | 0.6434 | 0.6648 | 0.6831 | 0.3038 | 0.8504 | 0.9781 | 0.5283 | 0.7053 | 0.5933 | 0.7062 |
| a46vf - a46v | 0.0688 | **0.0105*** | 0.5349 | 0.3464 | 0.3066 | 0.9766 | 0.5895 | 0.4481 | **0.0101*** | 0.9936 |
| a46vf - p46vf | 0.9393 | 0.9003 | 0.6864 | 0.9146 | 0.968 | 0.489 | 0.402 | 0.7902 | 0.9948 | 0.7508 |
| a46v - p46v | 0.8536 | 0.3958 | 0.5219 | 0.2731 | 0.677 | 0.8721 | 0.287 | 0.7048 | 0.9504 | 0.7352 |
| p46df - p46d | 0.7061 | 0.0724 | 0.3835 | 0.1953 | 0.638 | 0.7277 | 0.5781 | 0.8386 | **0.003**** | 0.9546 |
| p46df - p46vf | 0.7953 | 0.7199 | 0.6601 | 0.6934 | 0.226 | 0.7501 | 0.7768 | 0.8638 | 0.8326 | 0.6022 |
| p46vf - p46v | 0.1742 | 0.0982 | 0.3824 | 0.0563 | 0.0746 | 0.3428 | 0.4663 | 0.3193 | **0.0146*** | 0.4608 |

## Discussion

In this study, we provide a detailed parcellation of the macaque prefrontal cortex (apart from the cingulate cortex as a part of the limbic system), and which encompasses 35 cyto- and receptor architectonic areas. The new parcellation scheme integrates and refines former maps of the PFC, particularly concerning area 46 of Walker, and includes novel subdivisions of areas 10 (10mv, 10md, and 10d), 9 (9d and 9l) and 8B (8Bd and 8Bs). It is shown on a 2D flat map to facilitate comparison with previous maps (*Barbas and Pandya, 1989*; *Caminiti et al., 2017*; *Carmichael and Price, 1994*; *Morecraft et al., 2012*; *Petrides and Pandya, 1994*; *Petrides and Pandya, 2002*; *Preuss and Goldman-Rakic, 1991*; *Walker, 1940*), and, in addition, *Table 1* was created as an overview of Rapan's areas (this study; *Rapan et al., 2021*) in regard to the previous borders of referenced maps. Borders were also transferred to the Yerkes19 template (*Donahue et al., 2016*) to enable an architectonically informed analysis of functional connectivity in the macaque brain.

When analysing changes in receptor densities from area to area, the receptor fingerprints revealed differences across the frontal lobe when moving from rostral to caudal portions. Rostrally located areas contained higher receptor densities, thus bigger receptor fingerprints, than more caudally located areas. These differences in the size of receptor fingerprints seem to be the main force driving

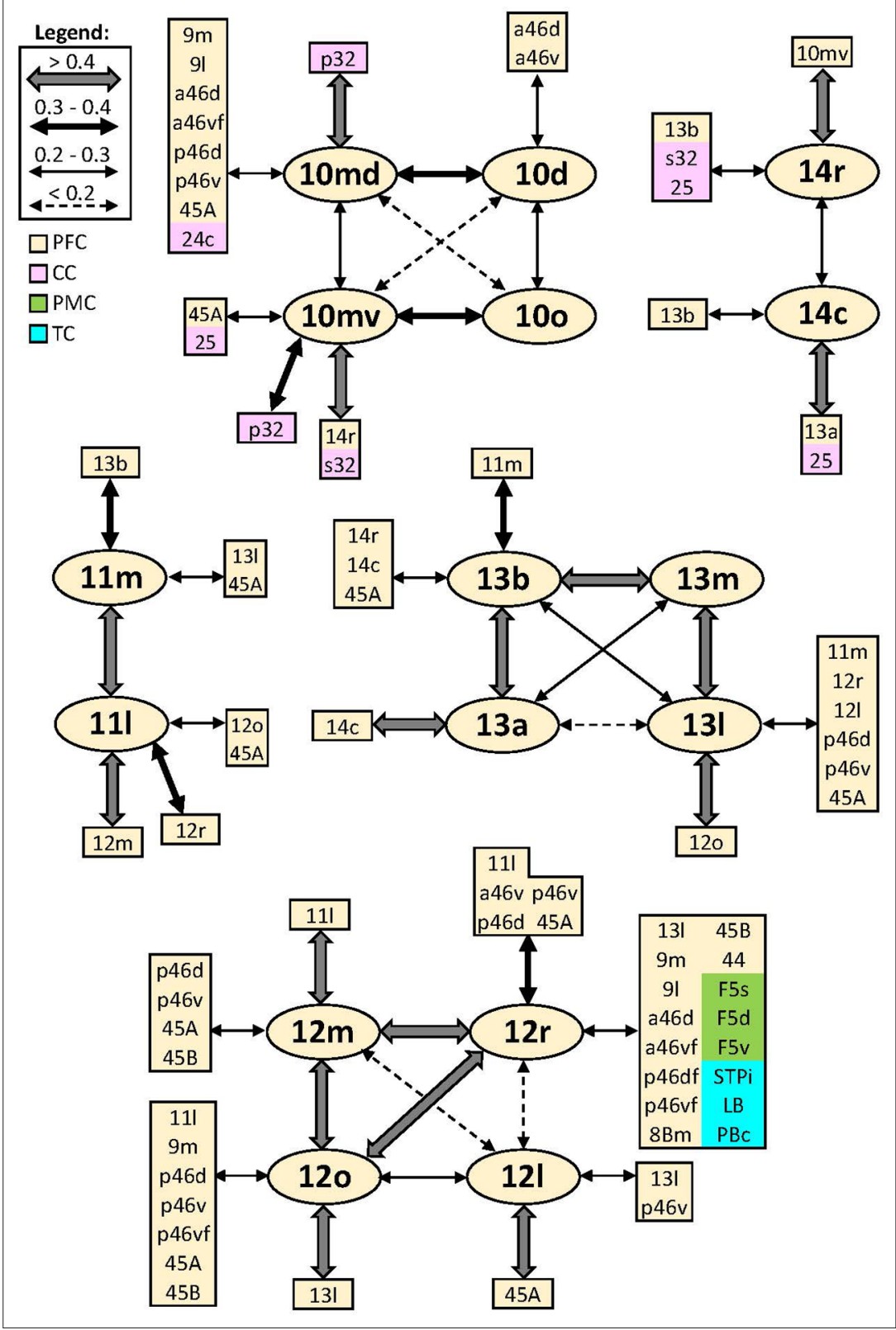

**Figure 12.** Schematic summary of the functional connectivity analysis between subdivisions of areas 10, 14, 11, 13, and 12. Legend shows the strength of the functional connectivity coefficient (z) is coded by the appearance (wider-thinner-doted) of the connecting arrows. Areas related to different brain regions are marked on the scheme with distinct colours; prefrontal cortex (PFC) in light yellow, cingulate cortex (CC) in pink, premotor cortex (PMC) in light green, and temporal cortex (TC) in light blue.

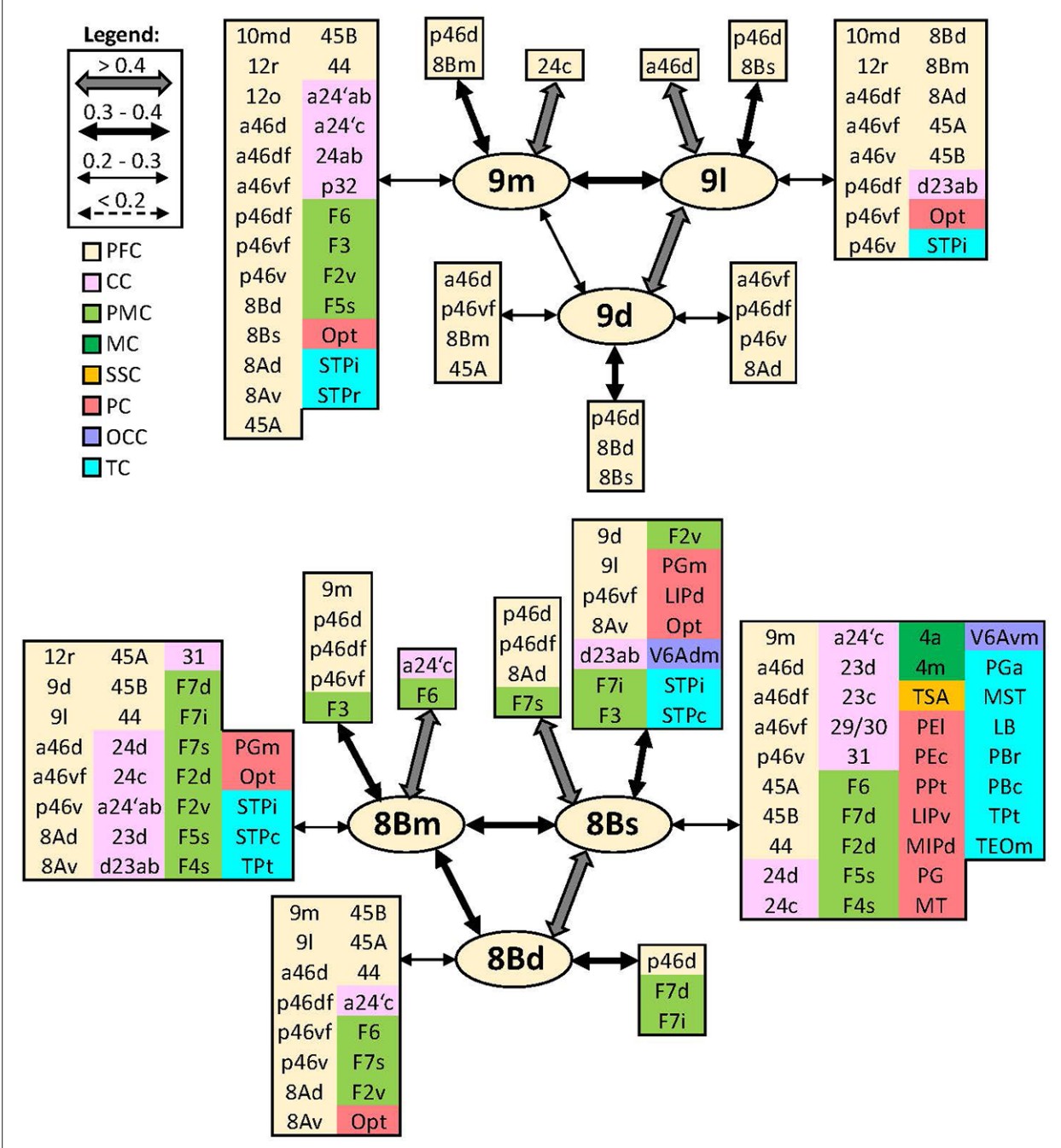

**Figure 13.** Schematic summary of the functional connectivity analysis between subdivisions of areas 9 and 8B. Legend shows the strength of the functional connectivity coefficient (z) is coded by the appearance (wider-thinner-doted) of the connecting arrows. Areas related to different brain region are marked on the scheme with distinct colours; prefrontal cortex (PFC) in light yellow, cingulate cortex (CC) in pink, premotor cortex (PMC) in light green, motor cortex (MC) in dark green, somatosensory cortex (SSC) in orange, parietal cortex (PC) in red, occipital cortex (OCC) in purple, and temporal cortex (TC) in light blue.

clustering of areas as revealed by the multivariate analyses. The heterogeneity within macaque frontal lobe is not only reflected by its architecture and molecular structure, but also by its functional diversity. The analysis of the functional connectivity revealed that posterior subdivisions of area 46 ('p46'), 45, 44, and 8A displayed the most extensive connectivity patterns within the frontal region, as well as with distinct cortical regions across the brain. Although not widespread pattern as for areas mentioned above, within the OFC only area 12r displayed connectivity pattern which included also remote

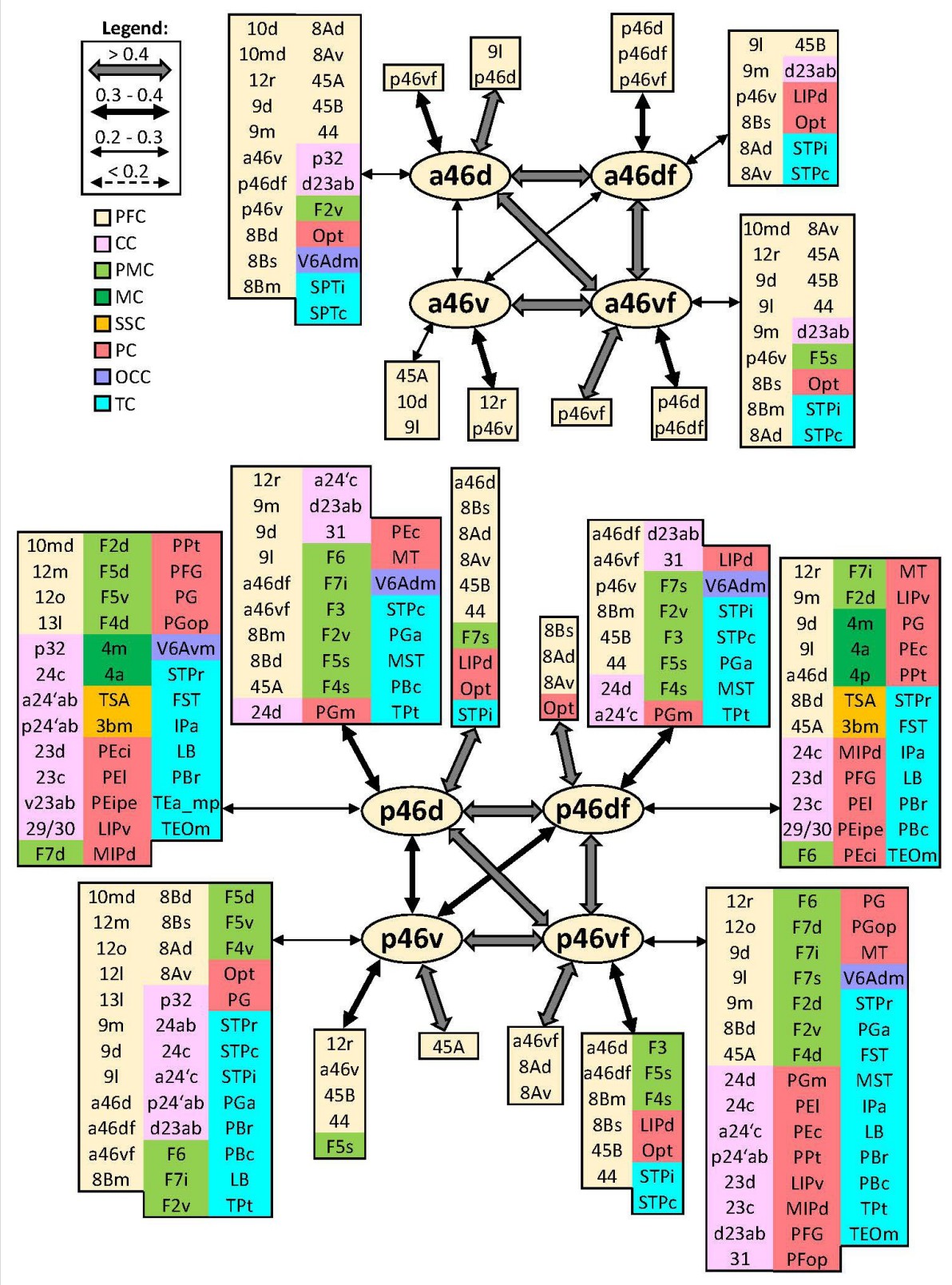

**Figure 14.** Schematic summary of the functional connectivity analysis between subdivisions of areas 46, rostral areas 'a46,' and caudal ones 'p46'. Legend shows the strength of the functional connectivity coefficient (z) is coded by the appearance (wider-thinner-doted) of the connecting arrows. Areas related to different brain region are marked on the scheme with distinct colours; prefrontal cortex (PFC) in light yellow, cingulate cortex (CC) in pink, premotor cortex (PMC) in light green, motor cortex (MC) in dark green, somatosensory cortex (SSC) in orange, parietal cortex (PC) in red, occipital cortex (OCC) in purple, and temporal cortex (TC) in light blue.

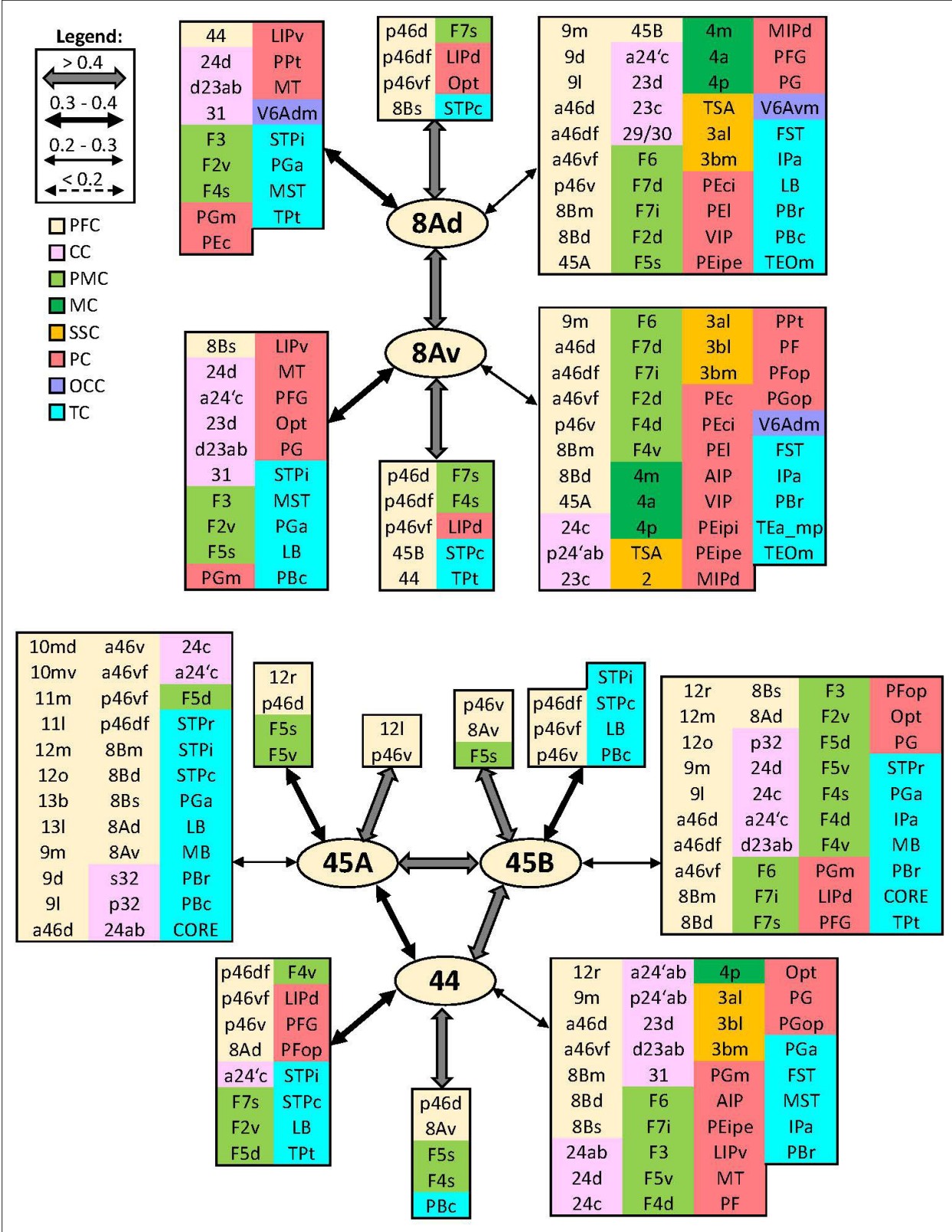

**Figure 15.** Schematic summary of the functional connectivity analysis between subdivisions of areas 8A and 45, and area 44. Legend shows the strength of the functional connectivity coefficient (z) is coded by the appearance (wider-thinner-doted) of the connecting arrows. Areas related to different brain region are marked on the scheme with distinct colours; prefrontal cortex (PFC) in light yellow, cingulate cortex (CC) in pink, premotor cortex (PMC) in light green, motor cortex (MC) in dark green, somatosensory cortex (SSC) in orange, parietal cortex (PC) in red, occipital cortex (OCC) in purple, and temporal cortex (TC) in light blue.

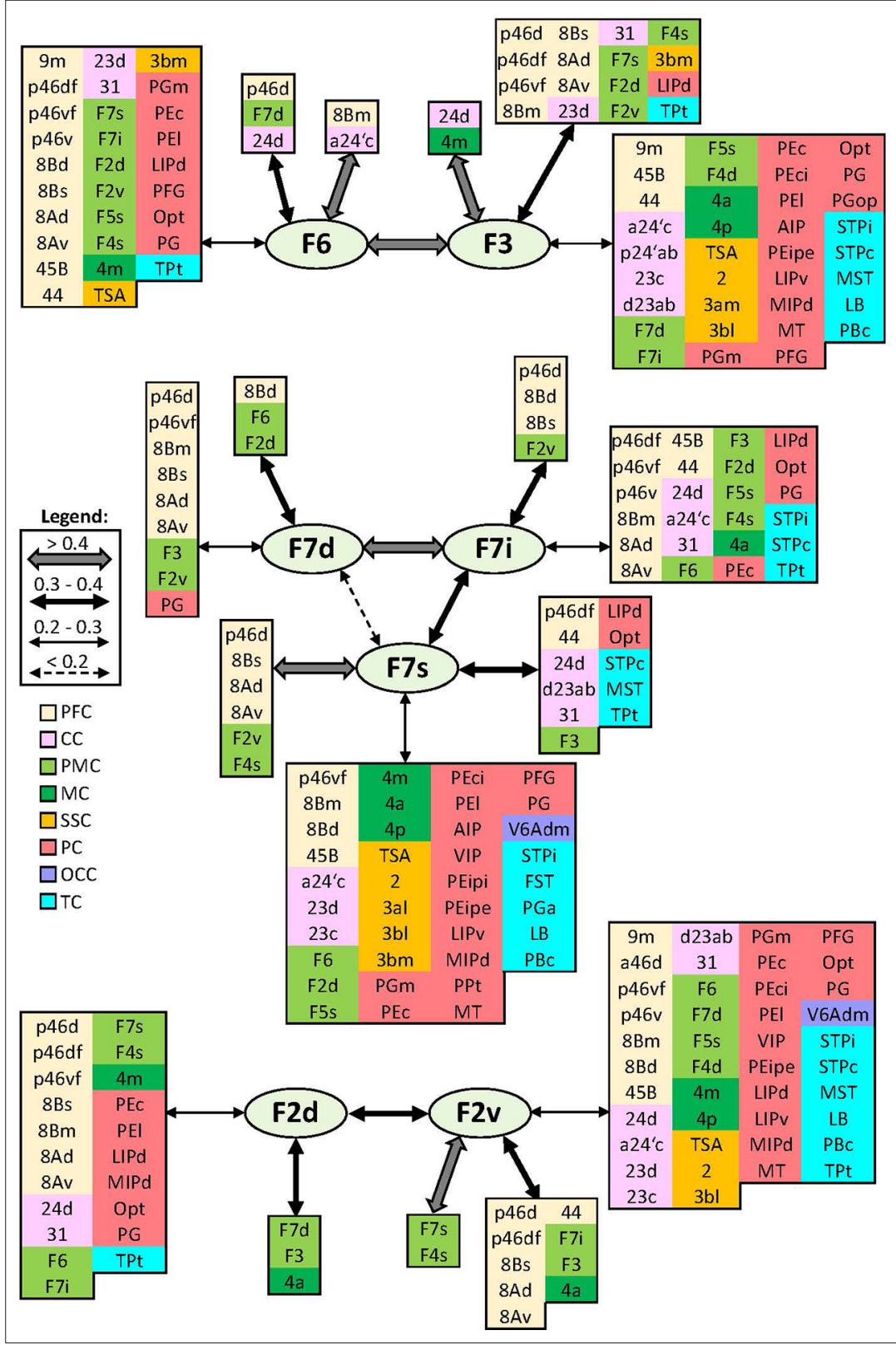

**Figure 16.** Schematic summary of the functional connectivity analysis between subdivisions of premotor areas F7 and F2, and areas F3 and F6. Legend shows the strength of the functional connectivity coefficient (z) is coded by the appearance (wider-thinner-doted) of the connecting arrows. Areas related to different brain region are marked on the scheme with distinct colours; prefrontal cortex (PFC) in light yellow, cingulate cortex (CC) in pink, premotor

*Figure 16 continued on next page*

*Figure 16 continued*

cortex (PMC) in light green, motor cortex (MC) in dark green, somatosensory cortex (SSC) in orange, parietal cortex (PC) in red, occipital cortex (OCC) in purple, and temporal cortex (TC) in light blue.

premotor and temporal areas. In contrast, areas 10, 14, 13, and 11 displayed functional connectivity limited within the prefrontal region, possibly suggesting that these areas are affected by a lower signal-to-noise ratio (*Yeo et al., 2011*). Thus, when available, we discuss the results of our functional connectivity analysis in the framework of tracer studies with injection sites within our region of interest (e.g. *Markov et al., 2014*; *Gerbella et al., 2010*; *Carmichael and Price, 1996*). Furthermore, areas located within and around spur of the arcuate sulcus, that is, F7s, F2v, F4s, and F5s, showed rather widespread connectivity pattern across the brain compared to their respective counterparts within the same premotor area. Primary motor areas 4m, 4a, and 4p revealed strongest connections with neighbouring premotor and somatosensory areas, as well as with the parietal cortex.

## Comparison with previous architectonic maps of macaque prefrontal region

### Medial and orbital prefrontal regions (areas 10, 11, 14, 13, and 12)

*Walker, 1940* identified five relatively large cytoarchitectonic areas on the medial and orbital prefrontal cortex, that is, area 10 located on the frontal pole and encroaching onto the orbital surface, area 11 on the rostral orbitolateral surface, caudal areas 13 and 12 on the medial and lateral orbital surface, and area 14 located on the ventromedial convexity. *Preuss and Goldman-Rakic, 1991* identified subdivisions in areas 13 (labelled as 13L and 13M) and 14 (defined as 14A, 14L, and 14M), whereas *Carmichael and Price, 1994* published a more detailed map, which also included cytoarchitectonic subdivisions of areas 10 and 11, and is in accordance with the connectional diversity of this region (*Carmichael and Price, 1996*). We were able to confirm all areas defined by *Carmichael and Price, 1994*, except for those located in the frontal pole region (area 10 of Walker). Their map of the rostral granular area 10 displays areas 10m, located on the medial and dorsal surface of the hemisphere, and 10o, occupying the orbital surface of the medioventral gyrus, and delimited caudally by area 14r (*Carmichael and Price, 1994*). Our cyto- and receptor analyses confirmed the location and extent of area 10o. But it revealed the existence of three subdivisions within 10m, that is, mediodorsal area 10md, medioventral 10mv, and area 10d on the dorsal surface of the frontal pole. Indeed, these novel areas differed not only in their cyto- and receptor architecture, but also in their functional connectivity. Medial areas 10md and 10mv contrasted from their lateral counterparts 10d and 10o by a strong connectivity with the cingulate cortex, that is, dorsally located area 10md with p32, and ventrally, 10mv with s32 and to a lesser extent with p32. Interestingly, macaque areas p32 and s32 have established homologies within the human brain, where they have been associated with the processing of emotion (*Palomero-Gallagher et al., 2013*; *Palomero-Gallagher et al., 2019*; *Vogt et al., 2013*). Comparison between the tracer study by *Markov et al., 2014* and our functional connectivity analysis revealed certain similarities regarding connectivity of area 10. Careful inspection of their *Figure 2* reveals that the injection sites are at a location comparable mainly to that of our area 10md and, to a lesser extent, of our area 10d. They describe connectivity with prefrontal areas 14, 9, 46d, 46v, and 9/46d as well as with cingulate areas 25, 32, and 24c (*Markov et al., 2014*), which is in accordance with our results for areas 10md, whereas our area 10d presents a more restricted functional connectivity than does 10md since it is not correlated with the cingulate cortex.

Within the OFC, the present analysis confirmed the position and extent of areas 11l, 11m, 13l, 13m, 13b, 13a, 14r, and 14c as identified by *Carmichael and Price, 1994*. We also identified four subdivisions of Walker's area 12, but their spatial relationship differs from that described by *Carmichael and Price, 1994*. In both maps areas 12r and 12m occupy the rostral portion of the lateral orbital cortex, while areas 12l and 12o cover its caudal part. Areas 12r and 12l extend onto the ventrolateral convexity below the *ps*. However, unlike in the map of *Carmichael and Price, 1994*, where 12m abuts areas 12r, 12l, *and* 12o, in our parcellation area 12m does not have a common border with 12l since our area 12r extends further posteriorly than that of *Carmichael and Price, 1994*. The OFC plays an important role in a reward processing (e.g. association of stimulus), as well as in emotional and motivational aspects of behaviour (*Mishkin and Manning, 1978*; *Rolls, 2000*; *Rolls et al., 1990*; *Rudebeck*

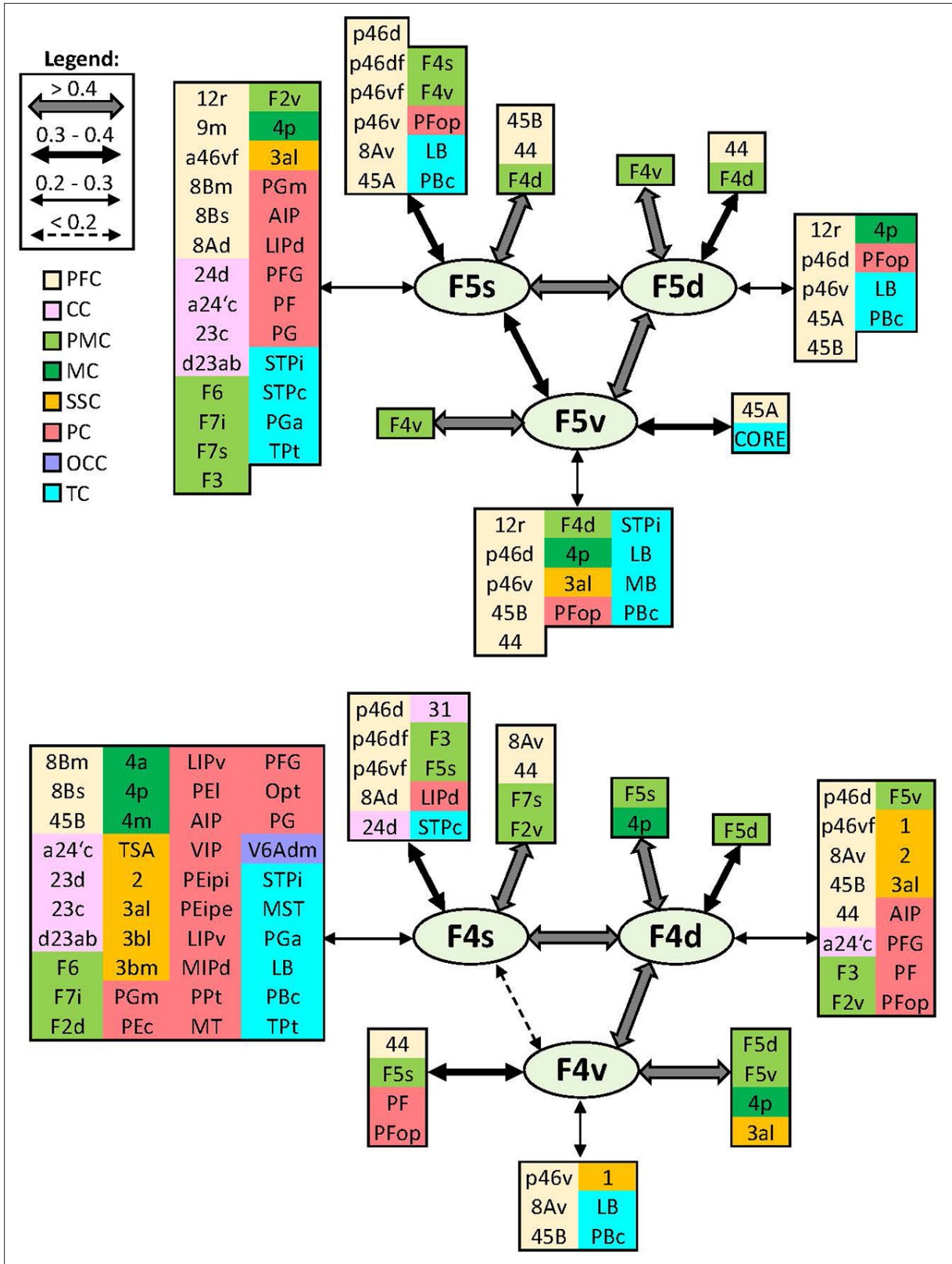

**Figure 17.** Schematic summary of the functional connectivity analysis between subdivisions of premotor areas F5 and F4. Legend shows the strength of the functional connectivity coefficient (z) is coded by the appearance (wider-thinner-doted) of the connecting arrows. Areas related to different brain region are marked on the scheme with distinct colours; prefrontal cortex (PFC) in light yellow, cingulate cortex (CC) in pink, premotor cortex (PMC) in light green, motor cortex (MC) in dark green, somatosensory cortex (SSC) in orange, parietal cortex (PC) in red, occipital cortex (OCC) in purple, and temporal cortex (TC) in light blue.

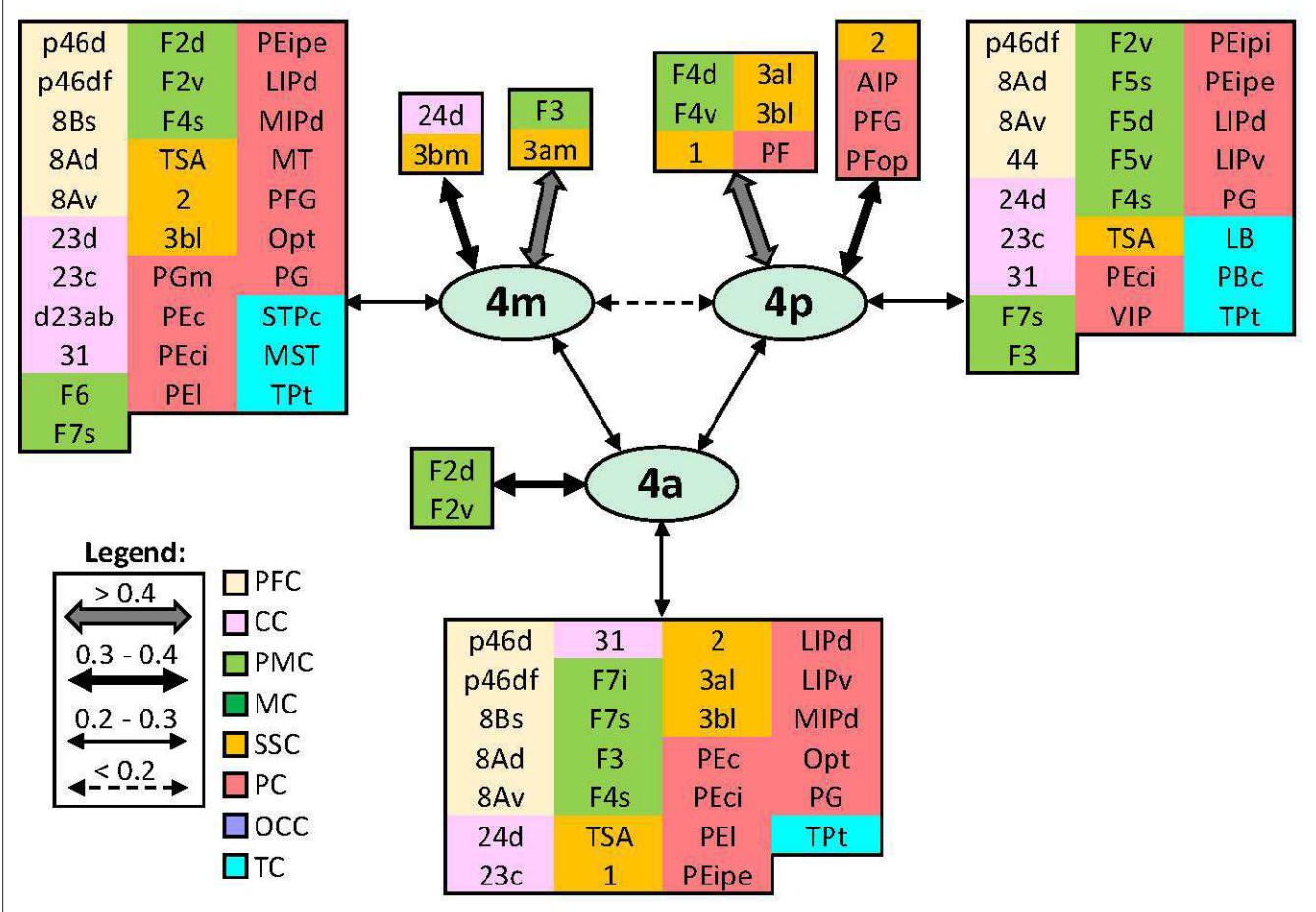

**Figure 18.** Schematic summary of the functional connectivity analysis between subdivisions of primary motor areas 4. Legend shows the strength of the functional connectivity coefficient (z) is coded by the appearance (wider-thinner-doted) of the connecting arrows. Areas related to different brain region are marked on the scheme with distinct colours; prefrontal cortex (PFC) in light yellow, cingulate cortex (CC) in pink, premotor cortex (PMC) in light green, motor cortex (MC) in dark green, somatosensory cortex (SSC) in orange, parietal cortex (PC) in red, occipital cortex (OCC) in purple, and temporal cortex (TC) in light blue.

and Murray, 2011b), whereas the ventrolateral region is associated with working memory for non-spatial tasks, as well as object memory retrieval (Wilson et al., 1993). In particular, the ventrolateral prefrontal cortex contains visual neurons specialized for the identification of object features (Asaad et al., 1998; Wilson et al., 1993). This brain region also encompasses our areas 12r and 12l, which express significantly lower $\alpha_2$ receptor densities than their medial counterparts 12m and 12o, respectively. Furthermore, we found areas 12r, 12m, and 12o to be strongly connected, while area 12l, which contained the lowest $\alpha_2$ receptor density of all subdivisions of area 12, was more strongly associated with area 45A than with the other subdivisions of area 12. Thus, the structural and functional organization of this region seems to be closely related to differences in the interareal levels of $\alpha_2$ receptors. This is an interesting finding since catecholamine neurotransmitters have been associated with cognitive decline in aged non-human primates (Arnsten and Goldman-Rakic, 1985), and in particular $\alpha_2$ receptor agonists have been shown to improve the delayed response performance test results in macaques (Arnsten et al., 1988).

## Dorsolateral prefrontal region (areas 9, 46, and 8B)

The analysis also resulted in a novel and more detailed subdivision within this region in regard to areas 9, 8B, and 46 than that described in previous maps (Petrides and Pandya, 1999; Preuss and Goldman-Rakic, 1991; Walker, 1940). Differences in the receptor architectonic organization of dorsolateral prefrontal areas are particularly obvious when looking at the normalized fingerprints, and

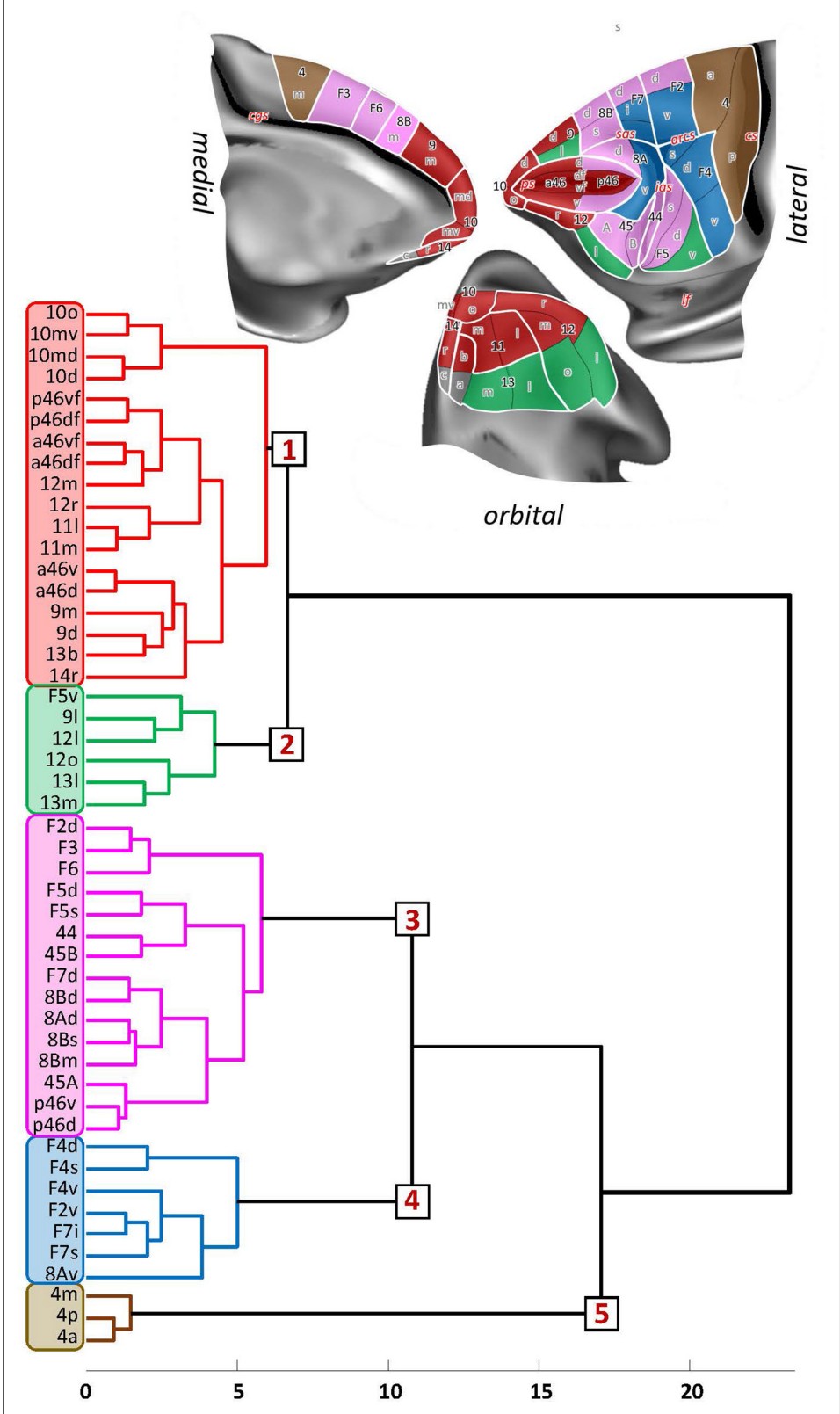

**Figure 19.** Receptor-driven hierarchical clustering of the receptor fingerprints in the macaque frontal lobe. The analyses include 33 of the 35 areas identified in this study (for areas 14c and 13a was not possible to extract receptor densities due to technical limitations), as well as 16 areas of the primary motor and premotor cortex identified in a previous study (*Rapan et al., 2021*) carried out on the same monkey brains. Above the hierarchical

*Figure 19 continued on next page*

*Figure 19 continued*
dendrogram, the extent and location of the five clusters are depicted on the medial, lateral, and orbital surface of the Yerkes19 atlas. Clusters are colour coded based on the corresponding colour on the dendrogram.

significant differences were found between rostral and caudal mediodorsal prefrontal areas 9 and 8B, respectively.

Although some authors confirmed Walker's area 9 (*Walker, 1940*; e.g. *Barbas and Pandya, 1989*; *Carmichael and Price, 1994*; *Morecraft et al., 2012*; *Petrides and Pandya, 1994*; *Petrides and Pandya, 2002*), others (e.g. *Caminiti et al., 2017*; *Preuss and Goldman-Rakic, 1991*) described a dorsal (9d) part, located on the convexity superior to the principal sulcus, and a medial (9m) subdivision on the medial surface of the hemisphere, dorsal to the cingulate sulcus. We confirmed the existence of 9m, but identified cyto- and receptor architectonic differences within their area 9d. Here only

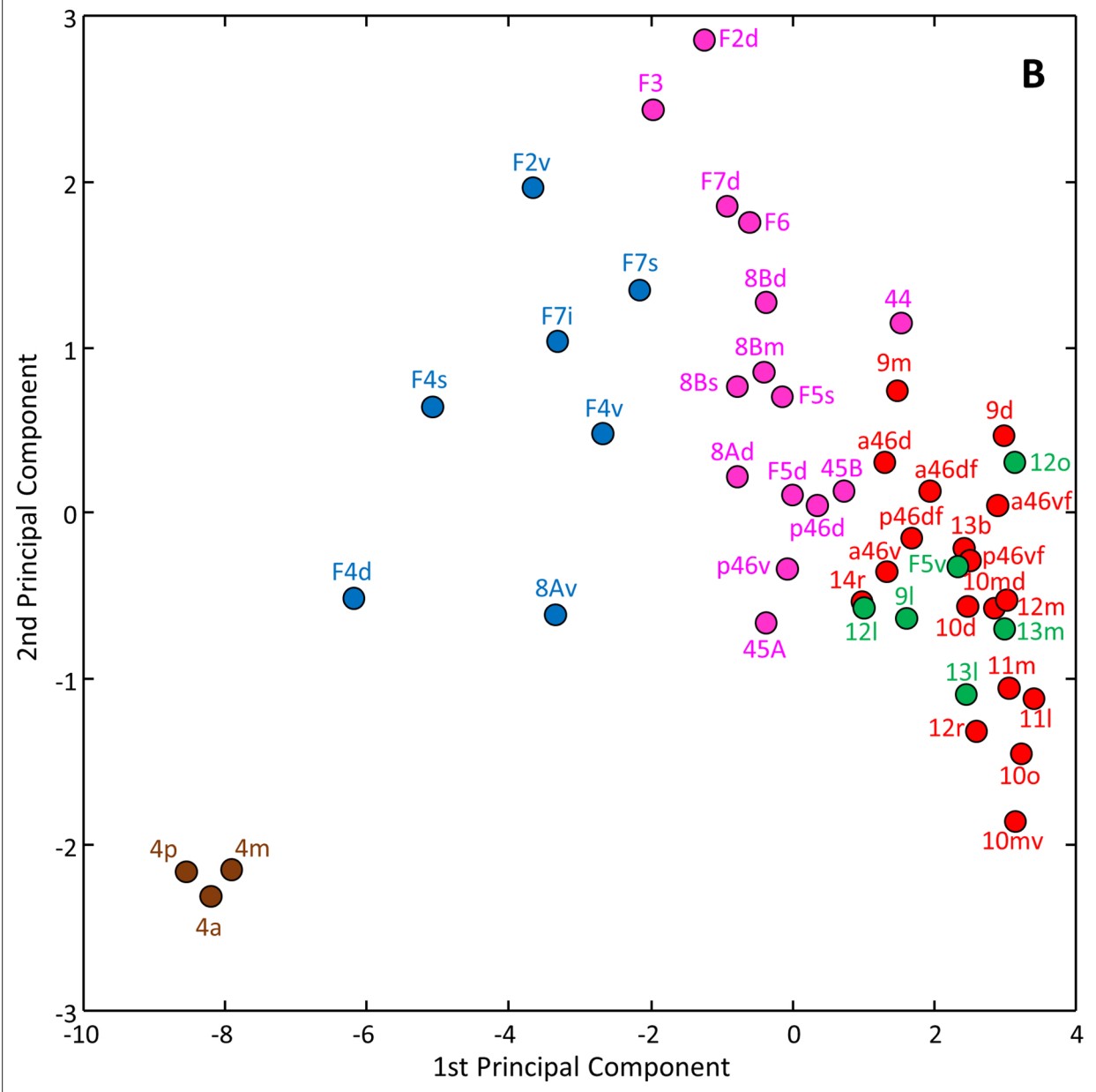

**Figure 20.** Principal component analysis (variance 79.8%) of the receptor fingerprints, where the k-means analysis showed five as the optimal number of clusters.

the most dorsal part was labelled as area 9d, whereas more laterally, we identified the distinct area 9l. Whereas area 9l presented a strong functional connectivity with laterally adjacent area a46d, this was not case for our areas 9d and 9m. These areas were more strongly associated with posterior area p46d. Moreover, dorsal areas 9d and 9l are strongly interconnected. Interestingly, medial area 9m, which has been included in the medial prefrontal network (*Carmichael and Price, 1996*), correlated with anterior cingulate area 24c more strongly than with the other subdivisions of area 9.

Further caudal on the mediodorsal prefrontal surface, a transitional region between granular prefrontal and agranular premotor areas was described, namely dysgranular area 8B of *Walker, 1940* and *Petrides and Pandya, 1994*, which encompasses areas 8Bm and 8Bd of *Preuss and Goldman-Rakic, 1991* and *Morecraft et al., 2012*. Similar to the situation described above for area 9, we were able to confirm the existence of area 8Bm, but we subdivided area 8Bd into a dorsal component located caudal to area 9d (our area 8Bd) and a ventral component 8Bs, which abuts area 9l. Previous maps (e.g. *Morecraft et al., 2012*; *Petrides and Pandya, 1994*; *Petrides and Pandya, 2002*; *Preuss and Goldman-Rakic, 1991*; *Walker, 1940*) depicted area 8B just rostral to the *sas*. However, the extent of our area 8B includes cortex above *sas* as well. Hence, area 8Bd was also identified on the most dorsal portion of the hemisphere rostral to and above the *sas*. Further lateral on the dorsal surface we identified area 8Bs, which extends onto the dorsal wall of the *sas*. Subdivisions of area 8B do not present a transitional region only by their structural features, but also based on their extensive functional connectivity since our analysis showed a widespread functional connectivity with prefrontal areas, as well as with the medial and dorsal premotor cortex. Dorsal prefrontal cortex, which is occupied by areas 9 and 8B, is involved in orientating processes and joint attention in primate brain (*Petrides and Pandya, 1999*), which is an important behavioural feature when animals need to integrate stimuli from different sensory modalities in order to select an adequate behavioural response. However, unlike area 9, more posteriorly adjacent mediodorsal area 8B is a prominent target region of the prestriate and the medial parietal cortex (*Petrides and Pandya, 1999*). In particular, neurons in area 8B fire during spontaneous ear and eye movement, as well as during the processing of auditory information (*Bon and Lucchetti, 1994*). Thus, it has been suggested that area 8B represents a macaque-specific region which is not present in humans, the so-called premotor ear-eye field (PEEF) (*Lucchetti et al., 2008*).

*Walker, 1940* defined area 46 within and around *ps,* and occupying large portion of the lateral prefrontal surface caudal to area 10, while on the most posterior end of principal sulcus, area 46 was replaced by area 8A. This location of area 46 in the macaque monkey has been confirmed in various anatomical studies (*Caminiti et al., 2017*; *Petrides and Pandya, 1994*; *Petrides and Pandya, 2002*; *Preuss and Goldman-Rakic, 1991*); however, it was widely acknowledged that this large region is not homogeneous, and distinct subdivisions with many discrepancies among parcellation schemes were made by different authors. *Preuss and Goldman-Rakic, 1991* identified four subareas along the principal sulcus. Two areas within the sulcus on the dorsal and ventral wall close to the fundus (inner subareas), areas 46d and 46v, respectively, and two areas on the dorsal and ventral shoulders of the sulcus and extending onto the free surface of the hemisphere (outer areas) areas 46dr and 46vr, respectively. Other authors identified rostro-caudal differences within Walker's area 46, but only described a dorsoventral segregation in the caudal portion, thus resulting in a parcellation with a rostral area 46 and caudal areas 9/46d and 9/46v located on the dorsal and ventral banks of the principal sulcus, respectively, and extending onto the free surface of the hemisphere (*Borra et al., 2019*; *Caminiti et al., 2017*; *Gerbella et al., 2013*; *Morecraft et al., 2012*; *Petrides and Pandya, 2006*).

The existence of dorsoventral subdivisions along the entire length of the principal sulcus, proposed by *Preuss and Goldman-Rakic, 1991*, could be corroborated by the present quantitative cyto- and receptor architectonic analysis. This study also confirmed the existence of rostro-caudal differences within the region and resulted in a new parcellation scheme for Walker's area 46 including a total of eight subdivisions – with areas 'a46' located within the anterior portion of *ps* and areas 'p46' occupying its most caudal. Receptor architectonic differences particularly highlighted borders between inner (subdivisions closer to the fundus, areas '46f') and outer (subdivisions extending onto surface, areas '46d' and '46v') portions of the principal region. We measured significantly higher levels of $\alpha_2$ receptors in the inner areas compared to their respective outer areas along the rostro-caudal *ps* axis. Area 46 plays an important role in higher-level cognitive processes, such as working memory (*Fuster, 2008*; *Goldman-Rakic, 1995*; *Petrides, 2000*), which has been reported to decline with age (*Arnsten and*

*Goldman-Rakic, 1985*). Similar to subdivisions of area 12, norepinephrine elicits different responses within area 46, depending on which type of receptor is stimulated. In particular, its binding to $\alpha_1$ and $\alpha_2$ receptors can have opposite effects on persistent activity during working memory. Stimulation of $\alpha_1$ receptors increases feedforward calcium-cAMP signalling, whereas stimulation of $\alpha_2$ receptors inhibits this process (*Arnsten et al., 1988*; *Arnsten et al., 2021*; *Arnsten and Jentsch, 1997*; *Hara et al., 2012*). Calcium-cAMP signalling must be kept within a tight range to support persistent activity, with excessive signalling leading to a shutdown of synaptic activity due to opening of potassium channels (*Arnsten et al., 2021*). The increase in $\alpha_2$ receptors in inner subdivisions of area 46 could help keep persistent activity in-check in these areas. In contrast, higher levels of kainate are measured in 'shoulder' areas of the *ps* than in the 'fundus' areas; however, only between anterior areas this difference has reached a significant level.

Our subdivision of Walker's area 46 into anterior/posterior and fundal/shoulder regions is further supported by the differences in the functional connectivity patterns of the areas we identified since posterior subdivisions of area 46 displayed a more widespread connectivity pattern than the anterior areas, and also in regard to all other prefrontal areas. Specifically, anterior areas showed the most prominent correlations with areas of the rostral prefrontal region as well as with their caudal 46 counterparts, while posterior areas strongly correlate with surrounding premotor areas in the lateral and medial frontal region, as well as with the parietal, temporal, and mid to posterior cingulate cortex. Our results are in accordance with previous connectivity analyses of area 46 (*Borra et al., 2019*; *Gerbella et al., 2013*), and may be indicative of the role of areas 'p46' in the visuospatial and visuomotor control of arm/hand reaching and eye movement, whereas areas 'a46' are more strongly involved in higher cognitive processes (*Borra et al., 2019*; *Gerbella et al., 2013*). Furthermore, the anterior part of *ps* is a major target of projections from the auditory and limbic cortex, whereas the posterior portion receives topographic sensory inputs from auditory, somatosensory, visual, and polysensory cortex (*Hackett et al., 1999*). Taken together, these findings clearly suggest that the anterior and posterior portions of cortex within the *ps* are involved in different aspects of behaviour, whereby areas 'p46' constitute a multimodal integration centre within the lateral PFC. Additionally, significant differences of kanite and $\alpha_2$ receptors between 'shoulder' and 'fundus' areas suggest an intermediate role of these receptors on working memory, a higher cognitive function associated with this region.

## Caudal region (areas 8Ad and 8Av)

Walker's area 8A has been subject of numerous architectonic analyses, resulting in maps that differ in the number and extent of areas depicted. A region defined as the granular part of area 8 (*Morecraft et al., 2012*; *Walker, 1940*) is associated with the frontal eye field (FEF) (*Bruce et al., 1985*; *Stanton et al., 1989*) and eye movement. However, eye movements are invoked only within a fundus of the arcuate sulcus, whereby the prearcuate surface is rather involved in the visual attention (*Germann and Petrides, 2020*). The present quantitative analysis encompasses a cortex rostral to premotor representation of the forelimb and mouth by the arcuate sulcus, from the ventral wall of the *sas*, across the portion of the prearcuate convexity located around the posterior portion of *ps* (where it borders posterior parts of area 46) and extending ventrally to the most caudal part of the anterior wall within the *ias* (where it abuts areas 44 and 45B) (*Morecraft et al., 2012*; *Walker, 1940*). The results of the present quantitative multimodal analysis are in accordance with the map of *Petrides and Pandya, 2006*, which identifies dorsal and ventral subdivisions within 8A, and not the tripartite subdivision of area 8A proposed by *Preuss and Goldman-Rakic, 1991*, or the rostro-caudal segregation of *Gerbella et al., 2007*. Furthermore, contrary to the map of *Preuss and Goldman-Rakic, 1991*, where their area 8Ar extends ventrally along the cortical surface adjacent to the *ias*, where it was delimited rostrally by area 12vl, our results are in accordance with the relative dorsoventral extent of area 8A described by *Petrides and Pandya, 2006* since area 8Av could be identified only on the cortical surface adjoining the most rostral portion of the *ias* and is replaced at this position by area 45A, so that it shares no common border with area 12. Moreover, the present receptor architectonic analysis also confirmed dorsoventral differences between subdivisions of area 8A since significantly higher kainate, $\alpha_1$, and 5-HT$_{1A}$ receptor densities were measured in 8Ad than in 8Av. Based on the qualitative cytoarchitectonic and receptor distribution pattern, we extended area 8Av onto the fundus of the arcuate sulcus, indicating that this area includes FEF. However, due to our material limitations in this study, this proposition was not tested by our quantitative approach. Both

subdivisions displayed a widespread connectivity pattern, with strongest correlations in the lateral frontal, parietal, and mid to posterior cingulate cortex, similar to the situation found for areas 'p46.' Interestingly, both areas 8Av and 8Ad display a strong connectivity with areas p46d, p46df, p46vf, but not with area p46v, whose connectivity pattern also differs from that of remaining 'p46' areas by its stronger correlation with the ventrolateral frontal region, but its weaker correlation with the inferior parietal and posterior cingulate cortex. Finally, it is noteworthy that areas 8Av and 8Ad (considered to constitute a key region regulating visual attention; *Germann and Petrides, 2020*; *Petrides, 2005*) were negatively correlated with areas of the occipital lobe, whereas p46v presented a positive correlation with this brain region, indicating that subdivisions of area 8A operate at a higher visual processing level than area p46v.

## Ventrolateral region (areas 45A, 45B, and 44)

Finally, the ventrolateral region also encompasses areas 44 and 45, which are thought to be the homologs of Broca's region in humans (*Petrides and Pandya, 2002*). In contrast with the parcellations proposed by *Walker, 1940* and *Preuss and Goldman-Rakic, 1991*, *Petrides and Pandya, 2002* found area 45 to extend rostrally onto the adjacent lateral surface of the hemisphere for a considerable distance, reaching as far as the *ipd*. Previous maps depicted area 45 mainly within the *ias*, and only encroaching onto the free surface, where it was replaced dorsally by area 46 and ventrally by area 12 (in the map of *Walker, 1940*), or rostrally by area 8Ar (in the map of *Preuss and Goldman-Rakic, 1991*). Furthermore, *Petrides and Pandya, 1999*; *Petrides and Pandya, 2002Petrides and Pandya, 1994* subdivided monkey area 45 into areas 45A and 45B. Area 45A occupies the ventral portion of the prearcuate convexity ventral to area 8Av, and extends rostrally into the *ipd,* where is substituted by 12r dorsally, and ventrally by 12l. Caudally 45A is delimited by 45B, which occupies the rostrodorsal wall of the *ias*. The subdivision of area 45 was based primarily on differences in the appearance of layer IV (*Petrides and Pandya, 1994*; *Petrides and Pandya, 1999*; *Petrides and Pandya, 2002*). The results of the present quantitative multimodal approach not only support the presence of an area 45, and not of area 12, on the prearcuate convexity, but also confirm the existence of areas 45A and 45B, with higher kainate densities in the former than the latter area.

While the present functional connectivity analysis shows that both areas 45 area correlated with polysensory areas STP and auditory-related temporal cortex (contrary to the findings of *Gerbella et al., 2010*), a suggestion that area 45A is associated with vocalization and communication behaviour, whereas area 45B rather plays a role in oculomotor frontal system (*Gerbella et al., 2010*), is in accordance with our analysis. We found that 45B is correlated to parietal areas, such as oculomotor area LIPd, and has much more extensive connectivity across the premotor cortex compared to 45A. Indeed, area 45A revealed a strong correlation only with premotor areas F5, which are involved in hand and mouth movements (*Fogassi et al., 2001*; *Maranesi et al., 2012*), which may have a function in communication.

In the past the existence of area 44 has been the subject of controversy. *Walker, 1940* and *Preuss and Goldman-Rakic, 1991* did not identify an area 44 in their maps because they considered that area 45 not only occupied the rostral, but also the caudal wall of the *ias*. Similarly, *Matelli et al., 1986* did not identify area 44 either since they thought that their area F5 continues rostrally into the *ias*, where it was followed by area 45. Petrides and Pandya (*Petrides et al., 2012*; *Petrides and Pandya, 1994*) identified a distinct dysgranular area between the caudally adjacent agranular premotor cortex and granular area 45, and this is supported by our structural (cyto- and receptor architecture) and functional connectivity analyses. Furthermore, tracer studies (*Cavada and Goldman-Rakic, 1989*; *Matelli et al., 1986*; *Petrides and Pandya, 1984*), which are in accordance with our functional connectivity results, showed that area 44 differs from the posteriorly adjacent ventral premotor cortex by its corticocortical projections to the parietal region. Whilst the ventral premotor region shares strong reciprocal connections with the most anterior areas of the inferior parietal lobule (IPL) (*Cavada and Goldman-Rakic, 1989*; *Matelli et al., 1986*; *Petrides and Pandya, 1984*), area 44 of the monkey brain is linked with the most posterior areas PFG and PG of the inferior parietal lobe (*Petrides and Pandya, 2009*). Thus, macaque area 44 may serve as an important region for the integration of different inputs in order to support the role of area 45B in oculomotor control (*Gerbella et al., 2010*) since the strongest correlations between frontal areas were found between area 44 and areas F5s and 45B, which also presented small Euclidean distances in the hierarchical clustering analysis. This finding further

supports the hypothesis that similarities in the size and shape of fingerprints constitute the molecular underpinning for related brain functions (*Zilles et al., 2015*; *Zilles and Palomero-Gallagher, 2017a*).

## Receptor-driven clustering of macaque frontal areas is associated with distinct functional connectivity patterns

Although functional connectivity often indicates direct anatomical connections (*Greicius et al., 2009*; *Thiebaut de Schotten et al., 2011*), it also reflects indirect connections, as well as an input from a common source area (*Smith et al., 2001*). Moreover, such analysis may be affected by the differences in local recurrent activity across areas (*Chaudhuri et al., 2015*). It is important to understand that while structural and functional aspects of brain organization are genuinely interconnected, they are not equal (*Rapan et al., 2021*). Contrary to the tract-tracing approach, functional connectivity can be easily assessed for novel parcellations of cortex, as shown in a present study, since it enables differentiation among areas with similar receptor profiles (e.g. newly identified subdivisions of area 10). Concerning neurotransmitters and their receptors, which constitute the molecular underpinning of signal transduction, we here analysed receptors with different mechanisms of action (ionotropic/metabotropic) and outcomes (excitatory/inhibitory). Activation of metabotropic receptors results in slower, longer lasting, and more widespread changes in membrane potential than does activation of ionotropic receptors. Therefore, if two areas differ in the relative balance of ionotropic versus metabotropic receptors, this will indeed result in different constraints on computational properties and could influence the temporal signature of neural activity. Taken together, functional connectivity facilitates the use of gold-standard anatomical data (e.g. the cytoarchitectonic boundaries and receptor data described here) by specialist in neuroimaging and enables a more systematic understanding of the macaque frontal cortex.

### Areas of cluster 1

Cluster 1 encompasses most of the rostrally positioned prefrontal areas, which share dense reciprocal connections with the limbic and auditory cortex (*Hackett et al., 1999*; *Romanski, 2007*), and also includes areas p46df and p46vf, which are located more posteriorly within the *ps*. The medial OFC is associated with value comparison since it shares reciprocal connections with brain regions involved in similar aspects of reward-guided behaviour (*Price, 2007*) and is a primary source of visceromotor inputs via reciprocal projections to the hypothalamus and brain stem (*Carmichael and Price, 1994*). Lesion studies of the medial OFC in the macaque brain, in particular to area 14, showed animals to be enticed into making incorrect choices, indicating that the decision-making process within the medial OFC is rather associated with motivation, than with action-like behaviour (*Noonan et al., 2010*; *Rudebeck and Murray, 2011b*). Since we found strong functional correlation between areas 10mv and 14r, it is interesting that most of the adjacent areas, such as 10o, 10mv, and 11m, showed significantly higher levels of inhibitory receptors (i.e. GABA$_A$ and GABA$_A$/BZ), but only area 10mv contained significantly higher levels of AMPA in regard to 14r. Additionally, similar to the medial frontopolar cortex, we found area 14r to have a strong functional connectivity with the anterior cingulate cortex, in particular to area 25. In contrast, connections of the lateral OFC to high-order sensory areas, such as the anterior temporal and perirhinal cortex (*Carmichael and Price, 1994*; *Price, 2007*), indicate that this region plays an important role in the reward-associated behaviour by assigning a value to stimuli. Animals with lesions in the rostrolateral OFC were unable to learn when to ascribe a different value when a new object is introduced, thus highlighting the importance of this region in value learning (*Noonan et al., 2010*). Although the medial and lateral orbitofrontal regions display distinct connectional patterns with distant cortical and subcortical structures, they also share numerous reciprocal connections which are thought to support the exchange and integration of information (*Carmichael and Price, 1994*). Specifically, areas 14r, 14c, 13a, 11m, and 12o serve as 'intermediary' areas connecting the lateral and medial OFC networks (*Carmichael and Price, 1994*; *Price, 2007*).

Microstimulation recordings revealed the presence of the auditory-responsive neurons within the caudal *ps* (*Hackett et al., 1999*; *Ito, 1982*; *Watanabe, 1992*), although most input from the auditory cortex targets the rostral portion of *ps* (*Barbas and Mesulam, 1985*) and, in particular, the frontopolar region (*Medalla and Barbas, 2014*). In this study, we found only a weak connectivity of the frontal polar region and orbital areas outside of the prefrontal cortex. However, our multivariate analyses grouped together subdivisions of area 10, anterior parts of area 46, as well as caudal fundal portions

of area 46, which are known to be targeted by the auditory cortex (*Barbas and Mesulam, 1985*; *Hackett et al., 1999*; *Medalla and Barbas, 2014*). Altogether, this suggests that the OFC provides an information on the object-value and motivation (*Carmichael and Price, 1994*; *Noonan et al., 2010*; *Price, 2007*; *Romanski, 2007*; *Rudebeck and Murray, 2011a*) which is then further processed by distinct regions in the medial and lateral PFC (*Goulas et al., 2014*). In addition, the dorsal prefrontal cortex, which is occupied by subdivisions of area 9 (also found in cluster 1), is involved in orienting processes and joint attention in the primate brain, which is an important feature when the animal processes and integrates stimuli from different sensory modalities in order to select the adequate behavioural response (*Petrides and Pandya, 1999*). Thus, PFC areas which we found to be grouped within cluster 1 based on similarities in their receptor fingerprints seem to be involved in distinct aspects of reward-guided behaviour.

## Areas of cluster 2

Cluster 2 is composed of closely grouped areas located in the posterior orbital PFC, that is, areas 13m, 13l, 12o, and 12l. It also contains dorsolateral prefrontal area 9l and premotor area F5v, located on the ventral portion of the postarcuate convexity, with which orbital areas do not share common borders. This is interesting since it demonstrates that frontal areas are not grouped simply on the basis of neurochemical similarities among neighbouring areas, but across the frontal cortex. Area F5v is mostly associated with mouth movements (*Maranesi et al., 2012*) and shares strong cortico-cortical connections with ventrally adjacent area ProM, as well as with the gustatory, orbitofrontal, insular, and somatosensory cortex (*Maranesi et al., 2012*), indicating an important role of this area in a feeding-related behaviour (*Cipolloni and Pandya, 1999*). While areas of the posterior orbital PFC, and in particular subdivisions of area 13, represent a multimodal region, which is targeted by the gustatory visual, auditory, somatosensory, and olfactory cortex, as well as by the amygdala, which assigns an emotional value to the integrated stimuli (*Barbas, 2007*).

Our functional connectivity analysis showed that newly identified area 9l has a strong correlation with multimodal area 46 (in particular with area a46d, and to a lesser extent with area p46v), as well as with polysensory area STPi and posterior cingulate areas d23a/b. Thus, area 9l may be a part of the multimodal region in the lateral PFC and serve as bridge with polysensory areas in the posterior orbital cortex. Furthermore, electrophysiological recordings of a brain region which topologically corresponds to our areas 9l and 9d (which are strongly correlated to each other) revealed that it contains neurons which are activated solely during voluntary head rotation, and neurons which are also activated when the head rotation is observed in another individual (mirror-like neurons), indicating that area 9 mediates head movements associated with certain social settings (*Lanzilotto et al., 2017*).

## Areas of cluster 3

Cluster 3 encompasses all subdivisions of area 8B, area 8Ad, ventrolateral areas 45A, 45B, and 44, areas occupying the posterior shoulder of *ps* (i.e. p46d, p46v), ventral premotor F5s and F5d, as well as medial and dorsal premotor areas F6, F3, F7d, and F2d. In accordance with our functional connectivity analysis, posterior prefrontal areas have strong correlation across the premotor cortex. With the exception of F7d and F5d, areas clustered here are also recognized by their widespread connectivity pattern with distant brain regions. Medial area F6 plays an important role in controlling when and how to execute complex motor plan (*Matelli et al., 1991*), but it lacks direct connections to the primary motor areas, as well as the spinal cord (*Dum and Strick, 2002*; *Luppino et al., 1993*), thus its contribution to movement is mediated via its dense connections with other premotor areas (e.g. F3, F2d). Thus, correlation found between area F6 and primary motor area 4m may reflect area's indirect connections (*Adachi et al., 2012*) rather than direct ones. On the other hand, posterior medial area F3 contains a complete somatotopic map of the body motor representation (*Woolsey et al., 1952*), and its direct anatomical connections with a primary motor cortex has been described (*Luppino et al., 1993*).

Area 8B is a prominent target region of the prestriate and the medial parietal cortex (*Petrides and Pandya, 1999*) and constitutes the cytoarchitectonic correlate of the functionally identified PEEF (*Lucchetti et al., 2008*), which is involved in auditory stimuli recognition and orientation processes (*Bon and Lucchetti, 1994*; *Lanzilotto et al., 2013*). Since neurons in area 8B fire during spontaneous

ear and eye movement, as well as during auditory information processing, indicating a role of this region in the integration of auditory inputs with ear and eye motor output, this area is thought to be monkey specific and have no homolog in the human brain (*Bon and Lucchetti, 1994*; *Lanzilotto et al., 2013*). In monkeys, ear movement improves localization of different sounds in the environment, whereas in humans this ability is rather shifted to eye-head coordination (*Bon and Lucchetti, 1994*).

Our novel architectonic subdivisions of area 8B presented different functional connectivity profiles. The functional connectivity profile of 8Bd is limited to adjacent areas on the dorsal portion of the PFC (e.g. areas 9d and F7d), whereas area 8Bs has a more widespread connectivity pattern which includes more ventrally located 8Ad and F7s. Furthermore, our cyto- and receptor architectonic results support the classification of area 8B as a transitional region between the prefrontal and the premotor cortex since the subdivisions of area 8B (which are dysgranular) showed a closer receptor architectonic relationship with premotor (agranular) than with the remaining prefrontal (granular) areas. This is particularly true for 8Bd and F7d, which are both (based on their position in our atlas) associated with the supplementary eye field (SEF) (*Schlag and Schlag-Rey, 1987*). Area 8Ad, which is partly associated to FEF, presents another region specialized for visual attention (*Amiez and Petrides, 2009*), but also, together with 8Bs, contributes to auditory responses (*Bruce and Goldberg, 1985*), as both areas have correlation with the auditory cortex, that is, parabelt areas PBr and PBc. The most prominent difference found between SEF and FEF is that saccades evoked from the latter region are of fixed vectors, whereas microstimulation recordings revealed evidence for the representation of eye position in SEF (*Mitz and Godschalk, 1989*; *Schlag and Schlag-Rey, 1987*). The present functional connectivity analysis revealed a strong correlation between areas 8Ad and p46d, which is in agreement with previous tracer studies (*Barbas and Mesulam, 1981*; *Barbas and Mesulam, 1985*; *Barbas and Pandya, 1989*). In general, input from the principalis region to the FEF may mediate regulatory control over gaze (*Schall, 1997*).

The posterior ventral cortex, which encompasses areas 45A (part of cluster 3) and 12l, shows evidence of overlapping auditory and visual responsive regions (*Romanski and Goldman-Rakic, 2002*; *Wilson et al., 1993*), indicating that convergent inputs allow response to both stimuli, especially when processing of information is related to face and vocalization communication, associated with the recognition of familiar and unfamiliar faces (*Romanski, 2007*). Finally, areas 45B and 44, located within the *ias*, are related with the oculomotor control (*Gerbella et al., 2010*). In addition, the present functional analysis showed that posterior area 44 has strong connection to neighbouring premotor area F5s, which, actually, presents the highest correlation found between two areas in our study. Therewith, ventral premotor areas F5s and F5d represent hand movements and are involved in object grasping (*Fogassi et al., 2001*). Specifically, area F5s (defined as area F5a by *Belmalih et al., 2009*) is associated with stereoscopic analysis of a 3D object (*Fogassi et al., 2001*). Thus, within cluster 3, we find caudal prefrontal areas associated with the attention and orientation based on the distinct visual and auditory inputs, whereas premotor areas grouped here are involved in arm reaching and orientation, with a main focus on a hand grasping (*Gerbella et al., 2017*).

## Areas of cluster 4

Cluster 4 contains area 8Av and premotor areas F7i, F7s, F2v, F4s, F4d, and F4v. As mentioned above, area 8Av is part of FEF, which is largely associated with saccades (*Bruce et al., 1985*). Due to the unique receptor architectonic features of the ventral portion of area 8A, indicated by the smallest receptor fingerprint of all prefrontal areas, we found a clear differentiation between 8Av and almost all surrounding prefrontal areas, where all significant receptor types were lower in 8Av. Thus, area 8Av was found to be more comparable to posteriorly adjacent premotor areas located within and around *arcs*, which are also characterized by relatively small fingerprints.

Furthermore, the functional connectivity analysis revealed that areas 8Av, F7s, F2v, and F4s, which are located within the spur of the arcuate sulcus, have strong connectivity with parietal areas associated with visual responses and control of saccadic and oculomotor movements, for example, intraparietal area LIP, and rostral areas Opt and PG of the inferior parietal lobule (*Niu et al., 2021*; *Andersen et al., 1990a*). In addition, we also found correlation with polysensory temporal areas STP and TPt, as well as with area MST, which is part of the temporal motion complex region (*Boussaoud et al., 1990*; *Kilintari et al., 2014*). This is interesting since fMRI studies of macaque behaviour involving voluntary saccadic eye movement reported a bilateral activation of both the rostral and caudal banks of *arcs*, as

well as of cortex within the spur of this sulcus (*Baker et al., 2006*; *Koyama et al., 2004*) That is, activations were found in a region which is thought to be part of an extended oculomotor region (*Amiez and Petrides, 2009*) associated with visual pursuit (*Fukushima et al., 2002*), and which is largely occupied by the areas composing our cluster 4. In particular, premotor areas of the extended oculomotor region are thought to play a role in blinking movement (*Bruce et al., 1985*) and in coordinating eye-arm movements within the peripersonal space (*Fujii et al., 1998*).

### Areas of cluster 5

Finally, primary motor areas 4m, 4a, and 4p demonstrated greater dissimilarity of their receptor fingerprints in regard to rest of the frontal areas and formed segregated cluster. Indeed, these areas are characterized by the one of the smallest receptor fingerprints among all areas identified in this study. Present and previous analysis of subdivisions of area 4 of our own group (*Rapan et al., 2021*) revealed differences in cyto- and receptor architecture as well as functional connectivity between area 4p, located mainly on the anterior bank of the central sulcus, and two other motor subdivisions, occupying the precentral convexity and medial surface of the hemisphere. In particular, area 4p showed strong functional correlation to the rostral areas PF, PFop, and PFG of the inferior parietal lobule, associated with somatosensory and body-related responses (*Andersen et al., 1990a*), whereas areas 4m and 4a showed higher correlations with caudal areas Opt, PG, and PGm, which are involved in visuomotor coordination (*Andersen et al., 1990a*; *Andersen et al., 1990b*). Unlike medial and dorsolateral areas, cortex occupied by area 4p has a higher packing density of the cortico-motor neurons (*Rathelot and Strick, 2009*), associated with the fine movements, such as the independent finger movement (*Porter and Lemon, 1995*). These neurons also play a role in the mapping of a new motor outline, which would enable performance of an additional skill (*Rathelot and Strick, 2009*). Since prefrontal area 44 revealed to be strongly connected with areas in premotor cortex associated with a hand movement, it is interesting that it also has strong functional connectivity with motor area 4p.

## Materials and methods
### Tissue processing

Both hemispheres of an adult macaque monkey (*M. mulatta*; male; brain ID DP1; 8 y; obtained as a gift from Professor Deepak N. Pandya) were used for cytoarchitectonic analysis in histological sections of a paraffin-embedded brain. Sodium pentobarbital was applied to deeply anesthetize the monkey, followed by a transcardial perfusion with cold saline and then 10% buffered formalin. The brain was removed and stored in a buffered formalin solution until further processing.

The brains of three adult macaques (*M fascicularis*; males; brain IDs 11530, 11539, 11543; 6 ± 1 y of age; obtained from Covance Laboratories, Münster, Germany) were processed for both cyto- and receptor architectonic analysis. Monkeys were sacrificed by means of a lethal intravenous injection of sodium pentobarbital. However, since receptor proteins are delicate in nature, only unfixed, deep frozen tissue can be used for receptor autoradiography (*Herkenham et al., 1990*; *Zilles et al., 2002*). Thus, the brains were immediately removed from the skull together with meninges and blood vessels to avoid further damage of superficial layers. The cerebellum, together with the brainstem, was separated from the rest of the brain. Each hemisphere was further divided into an anterior and a posterior slab at the level of the most caudal portion of the central sulcus. In this study, we examined all left hemispheres, except for brain 11539, where both hemispheres were analysed. The slabs were carefully placed on an aluminium plate to avoid any further deformation and slowly introduced into N-methylbutane (isopentane) at −40°C, where they were left for 10–15 min. Frozen slabs were stored in air-tight plastic bags at −80°C until used for sectioning. Animal care was provided in accordance with the NIH Guide for Care and Use of Laboratory Animals, and the European local Committee, and complied with the European Communities Council Directive.

### Identification of cortical areas

Starting point for the present parcellation was visual and microscopic inspection of our sectioned brains and previously published cytoarchitectonic literature of the macaque prefrontal cortex. Specifically, analysis of the OFC and ventrolateral areas 10, 11, 12, 13, and 14 was based on the parcellation scheme and nomenclature proposed by *Carmichael and Price, 1994*. Nomenclature of prefrontal

areas 9, 8B, 8A, 46, and 45 is based on Walker's (*Walker, 1940*) original parcellation scheme, though integrating later modifications (*Morecraft et al., 2012*; *Petrides, 2005*; *Preuss and Goldman-Rakic, 1991*).

Since the identification of neighbouring areas, based on a pure visual inspection, has previously resulted in maps that differ in terms of number, localization, and shape of cortical areas, in this study we applied a quantitative and statistically testable approach to test the localization and existence of all visually identified cytoarchitectonic borders (*Schleicher et al., 2000*; *Schleicher et al., 2009*; *Zilles et al., 2002*). Furthermore, cytoarchitectonically identified areas were further confirmed by differences in the regional and laminar distribution patterns of multiple neurotransmitter receptors, that is, by differences in receptor architecture.

## Processing postmortem brain and analysis of cytoarchitecture

DP1 brain was dehydrated in ascending graded alcohols (70–100% propanol), completed by a step-in chloroform. The brain was then embedded in paraffin and serially cut in the coronal plane with a large-scale microtome, resulting in 3305 20-µm-thick whole-brain sections. Every fifth section was mounted on gelatin-coated slides. Paraffin was removed and sections were rehydrated by a two-step washing (each of 10 min) with Xem-200 ('Xylol-Ersatz-Medium,' Vogel, Diatec Labortechnik GmbH) followed by graded washes in alcohol (10 min each in 100, 96, and 70% propanol) and finally a rinse in a pure water.

Sections were stained with a modified silver method (*Merker, 1983*; *Uylings et al., 1999*), which provides a high contrast between cell bodies and neuropil. In short, sections were pretreated 4 hr in 4% formic acid, then overnight in a 10% formic acid/30% peroxide solution. Sections were thoroughly washed, immersed twice for 5 min in 1% acetic acid, placed in a physical developer under constant movement until they become greyish, and then further developed with constant monitoring under the microscope until cell bodies were dark grey/black. The developer was prepared immediately before use by adding 30 ml of stock solution B (2 g $AgNO_3$, 2 g $NH_4NO_3$ and 10 g $SiO_2 \cdot 12WO_3 \cdot 26H_2O$ dissolved in 1 l distilled water; stored at room temperature) and then 70 ml of stock solution C (2 g $AgNO_3$, 2 g $NH_4NO_3$, 10 g $SiO_2 \cdot 12WO_3 \cdot 26H_2O$ and 7.3 ml of a 37% formaldehyde solution dissolved in 1 l distilled water; stored at room temperature) to 100 ml of stock solution A (50 g $Na_2CO_3$ dissolved in 1 l distilled water; stored at 4°C) under vigorous stirring, and development was terminated by two 5 min washes in 1% acetic acid. Sections were then fixed 5 min in a T-Max fixative (Kodak, two parts of T-Max and seven parts of distilled water), dehydrated in ascending grades of alcohol (70%, 96%, 100%) for 5 min in each dilution followed by two 5 min immersions in xylene before coverslipping with DePex mounting medium.

Sections were scanned with a light microscope (Axioplan 2 imaging, Zeiss, Germany) equipped with a motor-operated stage controlled by the KS400 and Axiovision (Zeiss) image analysing systems applying a 6.3 ×1.25 objective (Planapo, Zeiss) and a CCD camera (Axiocam MRm, Zeiss). Digitalized images are produced by stitching individual frames of 524 × 524 µm in size, 512 × 512-pixel spatial resolution, and in-plane resolution of 1 µm per pixel and 8-bit grey resolution.

The quantitative approach to cytoarchitectonic analysis relies on the volume fraction of cell bodies as estimated by the grey level index (GLI) in square measuring field, which is of fixed size (*Schleicher et al., 2009*). For each identified area, GLI images were generated from three neighbouring sections in the rostro-caudal direction, and ROIs were defined around each portion of the cortical ribbon where border had been identified by visual inspection by manually drawing an outer (at the interface between layers I and II) and an inner (at the border between layer VI and the white matter) contour. These contour lines were used to define equidistant traverses running perpendicularly to the cortical surface, along which the changes in grey values quantify the laminar pattern characteristic of a cortical area (*Schleicher et al., 2009*) and are measured as GLI-profiles (for details see *Palomero-Gallagher and Zilles, 2019*; *Zilles et al., 2002*). The shape of the profile can be parametrized, that is, presented as a frequency distribution of 10 features, which quantitatively describe the laminar distribution of the volume fraction of the cell bodies, constitute a feature vector of each profile, and can be standardized using different scales to set equal weight to each of the values used for multivariate analyses (*Schleicher et al., 2005*; *Zilles et al., 2002*).

Assuming that each area has a distinctive laminar pattern, areal borders would be located at the transition of the laminar pattern of one area to that of the neighbouring area. Therefore, the

Mahalanobis distance (MD; *Mahalanobis et al., 1949*) was applied to quantify differences in the shape of two profiles and enable detection of the position of borders (*Schleicher et al., 2005*; *Zilles et al., 2002*). Adjacent profiles were grouped into blocks to operate as a sliding widow shifting along the cortical ribbon by the distance of one profile, whereby the MD between immediately adjacent blocks was calculated and plotted as a distance function for all block positions. This process was repeated, but with systematically increasing block sizes from 10 to 24 profiles in order to control the stability of a distance function that changes with a number of profiles in a block. If two blocks belong to the same area, MD values are expected to be small since their laminar pattern coded by the profiles being compared is similar. To confirm and accept MD maxima as architectonically relevant borders, we applied Hotelling's $T^2$ test in combination with a Bonferroni adjustment of the p-values for multiple comparisons (*Schleicher et al., 2005*; *Zilles et al., 2002*). Finally, main maxima identified with numerous block sizes in one histological section were evaluated by comparison with corresponding maxima in three consecutive sections to exclude biologically meaningless maxima which may be caused by artefacts (e.g. ruptures, folds) or local discontinues in microstructure due to blood vessels or untypical cell clusters.

In order to visualize the relationship between identified areas and macroanatomic landmarks, we created a 2D flat map and a 3D model of the macaque prefrontal cortex. For the 2D flat map we generated a framework based on the sulcal anatomy of the DP1 brain, whereby every 40th section was represented as a line with indentations representing characteristic sulci and dimples and cytoarchitectonic borders were positioned relative to the corresponding macroscopic landmarks. Thus, the ensuing flat map enables visualization of borders even when they are located inside sulci (for more details see *Rapan et al., 2021*). To compute the 3D model, the positions of borders relative to macroanatomic landmarks (i.e. the fundus of sulci or dimples and the apex of gyri) were transferred by means of the connectome workbench software (https://www.human-connectome.org/software/connectome-workbench) to the surface representation of the Yerkes19 template brain (*Donahue et al., 2016*), thus also bringing our parcellation scheme into stereotaxic space.

## Processing unfixed brains and analysis of receptor architecture

We used quantitative in vitro receptor autoradiography to visualize binding sites of native receptors expressed on the cell membrane of neurons and glia cells. The advantage of this method is that it can be carried out on a large number of sections encompassing an entire hemisphere, alongside the possibility of precise quantification and a high specificity (*Palomero-Gallagher and Zilles, 2018*; *Zilles et al., 2002*).

Unfixed frozen slabs were serially sectioned in the coronal plane using a cryostat at –20°C, into 20-µm-thick sections, which were thaw-mounted on gelatin-coated glass slides. Sections were left to air dry and stored overnight in air-tight plastic bags at –20°C. Serial sections were used for the visualization of 14 distinct receptors types, that is, for glutamate (AMPA, kainate, NMDA), gamma-aminobutyric acid (GABA) (GABA$_A$, GABA$_B$, GABA$_A$-associated benzodiazepine binding sites [BZ]), acetylcholine (M$_1$, M$_2$, M$_3$), noradrenalin ($\alpha_1$, $\alpha_2$), serotonin (5HT$_{1A}$, 5HT$_2$), and dopamine (D$_1$), as well as for the visualization of cell bodies (see previous section) using previously published protocols (*Palomero-Gallagher et al., 2009*; *Zilles et al., 2002*; see *Table 5*), in three subsequent steps: a preincubation, a main incubation, and a rinsing step. The preincubation is carried out to rehydrate sections and to remove endogenous ligands that could block the binding sites. During the main incubation, two parallel experiments are conducted to test the specific binding ability of each ligand. In one, sections were incubated in a buffer solution with tritiated ligand to identify total binding of each ligand type. In the second, neighbouring sections were incubated in buffer solution containing the tritiated ligand and a receptor type-specific displacer in a 1000-fold higher concentration to visualize non-specific binding of the same ligand. Finally, the difference between total and non-specific binding demonstrates the specific binding ability for each ligand. In this study, specificity of ligands used resulted in a non-specific binding of less than 5% of the total binding. In the rinsing step, the binding process was stopped and free ligand and buffer salts removed. Air-dried, radioactive sections were then co-exposed with plastic tritium-standards (calibrated for protein density, and with known increasing concentrations of radioactivity) against β radiation-sensitive films (Hyperfilm, Amersham) for 4–18 wk depending on the analysed ligand. A densitometric analysis (*Palomero-Gallagher and*

**Table 5.** Receptor labelling protocols.
Square brackets indicate substances that are only included in the buffer solution for the main incubation.

| Transmitter | Receptor | Mechanism outcome | Ligand (nM) | Property | Displacer (µM) | Incubation buffer | Pre-incubation | Main incubation | Final rinsing |
|---|---|---|---|---|---|---|---|---|---|
| Glutamate | AMPA | Excitatory Ionotropic | [³H]-AMPA (10) | Agonist | Quisqualate (10) | 50 mM Tris-acetate (pH 7.2) [+100 mM KSCN] | 3 × 10 min, 4°C | 45 min, 4°C | 1. 4 × 4 s 2. Acetone/glutaraldehyde (100 ml + 2.5 ml), 2 × 2 s, 4°C |
| | NMDA | Excitatory Ionotropic | [³H]-MK-801 (3.3) | Antagonist | (+)MK-801 (100) | 50 mM Tris-acetate (pH 7.2) + 50 µM glutamate [+30 µM glycine +50 µM spermidine] | 15 min, 4°C | 60 min, 22°C | 1. 2 × 5 min, 4°C 2. Distilled water, 1 × 22°C |
| | Kainate | Excitatory Ionotropic | [³H]-Kainate (9.4) | Agonist | SYM 2081 (100) | 50 mM Tris-acetate (pH 7.1) [+10 mM $Ca^{2+}$-acetate] | 3 × 10 min, 4°C | 45 min, 4°C | 1. 3 × 4 s 2. Acetone/glutaraldehyde (100 ml + 2.5 ml), 2 × 2 s, 22° C |
| GABA | $GABA_A$ | Inhibitory Ionotropic | [³H]-Muscimol (7.7) | Agonist | GABA (10) | 50 mM Tris-citrate (pH 7.0) | 3 × 5 min, 4°C | 40 min, 4°C | 1. 3 × 3 s, 4°C 2. Distilled water, 1 × 22°C |
| | $GABA_B$ | Inhibitory Metabotropic | [³H]-CGP 54626 (2) | Antagonist | CGP 55845 (100) | 50 mM Tris-HCl (pH 7.2) + 2.5 mM $CaCl_2$ | 3 × 5 min, 4°C | 60 min, 4°C | 1. 3 × 2 s, 4°C 2. Distilled water, 1 × 22°C |
| | $GABA_A$/Bz | Inhibitory Ionotropic | [³H]-Flumazenil (1) | Antagonist | Clonazepam (2) | 170 mM Tris-HCl (pH 7.4) | 15 min, 4°C | 60 min, 4°C | 1. 2 × 1 min, 4°C 2. Distilled water, 1 × 22°C |
| | $M_1$ | Excitatory Metabotropic | [³H]-Pirenzepine (1) | Antagonist | Pirenzepine (2) | Modified Krebs buffer (pH 7.4) | 15 min, 4°C | 60 min, 4°C | 1. 2 × 1 min, 4°C 2. Distilled water, 1 × 22°C |
| | $M_2$ | Inhibitory Metabotropic | [³H]-Oxotremorine-M (1.7) | Agonist | Carbachol (10) | 20 mM HEPES-Tris (pH 7.5) + 10 mM $MgCl_2$ + 300 nM pirenzepine | 20 min, 22°C | 60 min, 22°C | 1. 2 × 2 min, 4°C 2. Distilled water, 1 × 22°C |
| Acetylcholine | $M_3$ | Excitatory Metabotropic | [³H]-4-DAMP (1) | Antagonist | Atropine sulfate (10) | 50 mM Tris-HCl (pH 7.4) + 0.1 mM PSMF +1 mM EDTA | 15 min, 22°C | 45 min, 22°C | 1. 2 × 5 min, 4°C 2. Distilled water, 1 × 22°C |
| | $α_1$ | Excitatory Metabotropic | [³H]-Prazosin (0.2) | Antagonist | Phentolamine mesylate (10) | 50 mM Na/K-phosphate buffer (pH 7.4) | 15 min, 22°C | 60 min, 22°C | 1. 2 × 5 min, 4°C 2. Distilled water, 1× 22°C |
| Noradrenaline | $α_2$ | Inhibitory Metabotropic | [³H]-UK 14,304 (0.64) | Agonist | Phentolamine mesylate (10) | 50 mM Tris-HCl + 100 µM $MnCl_2$ (pH 7.7) | 15 min, 22°C | 90 min, 22°C | 1. 5 min, 4°C 2. Distilled water, 1× 22°C |
| | 5-HT$_{1A}$ | Inhibitory Metabotropic | [³H]-8-OH-DPAT (1) | Agonist | 5-Hydroxy-tryptamine, (1) | 170 mM Tris-HCl (pH 7.4) [+4 mM $CaCl_2$ + 0.01% ascorbate] | | 30 min, 22°C 60 min, 22°C | 1. 5 min, 4°C 2. Distilled water, 3× 22°C |
| Serotonin | 5-HT$_2$ | Excitatory Metabotropic | [³H]-Ketanserin (1.14) | Antagonist | Mianserin (10) | 170 mM Tris-HCl (pH 7.7) | 30 min, 22°C | 120 min, 22°C | 1. 2 × 10 min, 4°C 2. Distilled water, 3 × 22°C |

*Zilles, 2018*; *Zilles et al., 2002*) was carried to measure binding site concentrations in the ensuing receptor autoradiographs.

Autoradiographs were digitized with an image analysis system consisting of a source of homogeneous light and a CCD-camera (Axiocam MRm, Zeiss) with an S-Orthoplanar 60 mm macro lens (Zeiss) corrected for geometric distortions, connected to the image acquisition and processing system Axiovision (Zeiss). Spatial resolution of the resulting images was 3000 × 4000 pixels; 8-bit grey value resolution. The grey values of the digitized autoradiographs code for concentrations of radioactivity. To transform grey values into fmol binding sites/mg protein, a linearization of the digitized autoradiographs had to be performed in a two-steps process, carried out with in-house-developed MATLAB (The MathWorks, Inc, Natrick, MA) scripts. First, the grey value images of the plastic tritium standards were used to compute the calibration curve, which defines the nonlinear relationship between grey values and concentrations of radioactivity. Then radioactivity concentration $R$ was then converted to binding site concentration $C_b$ in fmol/mg protein using *Equation 1*:

$$C_b = \frac{R}{E \cdot B \cdot W_b \cdot S_a} \cdot \frac{K_D + L}{L} \tag{1}$$

where $E$ is the efficiency of the scintillation counter used to determine the amount of radioactivity in the incubation buffer (depends on the actual counter), $B$ is the number of decays per unit of time and radioactivity (Ci/min), $W_b$ is the protein weight of a standard (mg), $S_a$ is the specific activity of the ligand (Ci/mmol), $K_D$ is the dissociation constant of the ligand (nM), and $L$ is the free concentration of the ligand during incubation (nM) (*Palomero-Gallagher and Zilles, 2018*; *Zilles et al., 2002*). For visualization purposes, a linear contrast enhancement and pseudo-colour coding of autoradiographs was applied using a spectre of 11 colours with equally spaced density ranges (red colour for highest and black for lowest receptor concentration levels).

Measurement of receptor binding sites (averaged over all cortical layers) was performed by computing the surface below receptor profiles, which were extracted from the linearized autoradiographs using in-house-developed scripts for MATLAB (The MathWorks, Inc) in a manner comparable to the procedure described above for GLI profiles. However, for receptor profiles the outer contour line was defined following the pial surface, and not the border between layers I and II. Thus, for each area (with the exception of areas 13m and 13l) and receptor type, we extracted profiles from three consecutive sections in each of the four hemispheres examined. Due to technical problems, we were only able to obtain this data for areas 13m and 13l from two hemispheres (11530 and 11539_R), and we could not measure receptor densities in areas 14c and 13a.

Densities (i.e. averaged over all cortical layers) of each of the 14 different receptors in 33 of the 35 cytoarchitectonically defined areas were calculated. Due to technical limitations associated with the cutting angle of the coronal sections, it was not possible to measure densities in areas 13a and 14c. The precise sampling for the measurements of each cytoarchitectonically defined area was verified by aligning autoradiographs with defined cytoarchitectonic borders in neighbouring silver-staining sections in the corresponding brain processed for the receptor architectonic analysis. For each of the examined areas and their subdivisions, the mean densities of all receptors averaged over all four hemispheres in that area were then visualized simultaneously as 'receptor fingerprints,' that is, as polar coordinate plots which reveal the specific balance of different receptor types within a cytoarchitectonic entity (*Zilles et al., 2002*).

## Statistical analysis of the receptor densities

To determine whether there were significant differences in receptor architecture between paired areas (in particular our analysis was focused on directly bordering areas within the prefrontal region), stepwise linear mixed-effect models were performed. A z-score normalization was performed for each receptor separately to ensure an equal weighting of all receptors in subsequent statistical analyses. All statistical analyses were conducted using the R programming language (version 3.6.3.; Team, 2013).

We conducted a statistical testing which included three levels. In the first level, an omnibus test was carried out to determine whether there were differences across all regions when all receptor types are considered simultaneously (*Equation 2*). The model consists of fixed effects for area and receptor type, and hemisphere was set as a random factor.

$$D_{a,r,h} = \alpha_o + \alpha_1 A_a + \alpha_2 R_r + \alpha_3 A_a R_r + \beta_1 H_h \qquad (2)$$

where $D$ represents the receptor density, $A$ is the prefrontal area, $R$ is the receptor type, and $H$ is the hemisphere.

If the interaction effect between area and receptor type at first level of testing was found to be significant, a second level of simple effect tests was applied for each receptor separately to determine whether there were significant differences across all areas for each receptor type. The p-values were corrected for multiple comparison using the Benjamini–Hochberg correction for false discovery rate (*Benjamini and Hochberg, 1995*).

Finally, the third-level post hoc tests were used to identify the paired areas driving the statistical difference in the second-level tests. For each receptor type, 528 post hoc tests were performed. To correct for multiple comparisons in the third step tests, we performed the false discovery rate correction (*Benjamini and Hochberg, 1995*) separately for each receptor type (i.e. p-values were corrected for 528 comparisons per receptor type).

## Visualization and analysis of functional connectivity

All datasets used here for analysis are openly available sources from the recently established PRIME-DE (http://fcon_1000.projects.nitrc.org/indi/indiPRIME.html; *Milham et al., 2018*). Resting-state fMRI data from 19 macaque monkeys (all males, age = 4.01 ± 0.98 y) was collected with no contrast agent on a 3T scanner with a four-channel coil in Oxford (*Noonan et al., 2014*). For each animal, one resting-state scan (6.67 min, 250 volumes) was used. These data were downloaded from the PRIME-DE database (*Milham et al., 2018*) and preprocessed using a Human Connectome Project-like pipeline for Nonhuman Primate as described previously (*Autio et al., 2020*; *Xu et al., 2015*; *Xu et al., 2018*; *Xu et al., 2019*). For each macaque, the structural preprocessing includes denoising, skull-stripping, tissue segmentation, surface reconstruction, and surface registration to align to Yerkes19 macaque surface template (*Donahue et al., 2016*). The functional preprocessing includes temporal compressing, motion, correction, global mean scaling, nuisance regression (Friston's 24 motion parameters, white matter, cerebrospinal fluid), band-pass filtering (0.01–0.1 Hz), and linear and quadratic detrending. The preprocessed data then were co-registered to the anatomy T1 and projected to the middle cortical surface. Finally, the data were smoothed (FWHM = 3 mm) on the high-resolution native surface, aligned, and downresampled to a 10k surface (10,242 vertices per hemisphere). The preprocessed BOLD activity time courses for each monkey brain were demeaned and then concatenated in time. This enabled us to estimate the group functional connectivity maps for each seed region in a single analysis.

The connectivity of each identified prefrontal areas was investigated in regard to 76 cortical areas, previously defined by Palomero-Gallagher group, that is, 16 areas of (pre)motor cortex, 15 areas of cingulate cortex, 6 areas of somatosensory cortex, 23 areas of parietal cortex, and 16 areas of occipital cortex (*Impieri et al., 2019*; *Niu et al., 2021*; *Rapan et al., 2021*; *Rapan et al., 2022*). A representative time course was calculated for each of the 35 prefrontal areas and the 76 (pre)motor, cingulate, somatosensory, parietal, and occipital areas, giving 111 areas in total. For each of the 111 areas, a principal components analysis was performed on activity across all vertices within the area, where the first principal component was taken as the representative activity time course for each area.

The representative time courses of each of the 35 prefrontal areas were used as seeds for functional connectivity analysis. Since they were correlated with the activity time courses for each vertex on the surface using a Pearson correlation. A Fisher's r-to-z transformation was then applied to each of the correlation coefficients. This was visualized on the Yerkes19 cortical surface. Code used for the implementation and visualization of the functional connectivity analysis has been made publicly available (https://github.com/seanfw/macaque-pfc-func-conn, copy archived at *Rapan, 2023*).

## Multivariate analyses of receptor fingerprints

To reveal structure–function relationship between areas of the frontal lobe, we not only used receptor fingerprints of the here identified 33 prefrontal areas (except areas 13a and 14c, see above), but also included those of previously identified 16 motor and premotor areas (*Rapan et al., 2021*). Receptor densities were extracted from the same macaque brains. Hierarchical clustering and principal component analyses were carried out to enable grouping of areas based on receptor architectonic similarities (*Palomero-Gallagher et al., 2009*). We used a receptor fingerprint of each area as a feature

vector characterizing the area of interest. The Euclidean distance, which takes into account difference in the size and shape of fingerprint, was applied as a measure of (dis)similarities between receptor fingerprints.

Before any statistical analysis was conducted, it was necessary to normalize all absolute receptor values due to large differences in absolute densities across receptor types. Receptors with high absolute density values (i.e. GABAergic receptors) would dominate the calculation of the Euclidean distance between areas, as well as of the principal component analysis, cancelling intended multi-modal approach in the present analysis. Whereas normalized receptor values enable for each receptor type to contribute with equal significance to the statistical analyses. Here, z-scores calculation was applied since this approach maintains the relative differences in receptor densities among areas, that is, the mean density of a given receptor across all examined areas was subtracted from the mean density of the same receptor in a defined area and obtained value was divided by the standard deviation of that receptor over all areas. The Ward linkage algorithm was chosen as the linkage method in combination with the Euclidean distances. It yielded a higher cophenetic correlation coefficient than any other combination of alternative linkage methods and measurements of (dis)similarity. The cophenetic correlation coefficient quantifies how well the dendrogram represents the true, multidimensional distances within the input data. The k-means analysis was applied to identify the highest acceptable number of clusters and confirmed by the k-means permutation test.

## Acknowledgements

Open Access publication costs are funded by the Deutsche Forschungsgemeinschaft (DFG, German Research Foundation) 491111487. Katrin Amunts and Nicola Palomero-Gallagher: founder: European Union's Horizon 2020; grant reference number: 945539 (Human Brain Project SGA3); founder: Helmholtz Association's Initiative and Networking Fund; grant reference number: InterLabs-0015. Nicola Palomero-Gallagher: founder: Federal Ministry of Education and Research (BMBF); grant reference number: 01GQ1902. Xiao-Jing Wang: founder: National Institute of Health (NIH); grant reference number: R01MH122024-02. The founders had no role in study design, data collection and interpretation, or the decision to submit the work for publication. The authors declare that the research was conducted in the absence of any commercial or financial relationships that could be acknowledged as a potential conflict of interest.

## Additional information

### Funding

| Funder | Grant reference number | Author |
| --- | --- | --- |
| Horizon 2020 Framework Programme | 945539 (HBP SGA3) | Katrin Amunts<br>Nicola Palomero-Gallagher |
| Helmholtz Association | Initiative and Networking Fund InterLabs-0015 | Katrin Amunts<br>Nicola Palomero-Gallagher |
| Bundesministerium für Bildung und Forschung | 01GQ1902 | Nicola Palomero-Gallagher |
| National Institutes of Health | R01MH122024-02 | Xiao-Jing Wang |

The funders had no role in study design, data collection and interpretation, or the decision to submit the work for publication.

### Author contributions

Lucija Rapan, Validation, Investigation, Visualization, Writing - original draft; Sean Froudist-Walsh, Conceptualization, Software, Writing – review and editing; Meiqi Niu, Conceptualization, Data curation, Software, Formal analysis, Visualization, Writing – review and editing; Ting Xu, Ling Zhao, Resources, Data curation, Formal analysis, Visualization, Writing – review and editing; Thomas Funck, Data curation, Formal analysis, Visualization, Writing – review and editing; Xiao-Jing Wang, Formal

analysis, Writing – review and editing; Katrin Amunts, Writing – review and editing; Nicola Palomero-Gallagher, Conceptualization, Resources, Data curation, Supervision, Project administration, Writing – review and editing

**Author ORCIDs**
Lucija Rapan ⓘ http://orcid.org/0000-0002-6582-5826
Meiqi Niu ⓘ http://orcid.org/0000-0001-7937-5814
Xiao-Jing Wang ⓘ http://orcid.org/0000-0003-3124-8474
Katrin Amunts ⓘ http://orcid.org/0000-0001-5828-0867
Nicola Palomero-Gallagher ⓘ http://orcid.org/0000-0003-4463-8578

**Decision letter and Author response**
Decision letter https://doi.org/10.7554/eLife.82850.sa1
Author response https://doi.org/10.7554/eLife.82850.sa2

## Additional files

**Supplementary files**
• MDAR checklist

**Data availability**
The files with the parcellation scheme will be available via EBRAINS platform of the Human Brain Project     (https://search.kg.ebrains.eu/instances/Project/e39a0407-a98a-480e-9c63-4a2225ddfbe4) and the BALSA neuroimaging site (https://balsa.wustl.edu/study/7xGrm). As well as the code used for the implementation and visualization of the functional connectivity analysis (https://github.com/seanfw/macaque-pfc-func-conn, copy archived at *Rapan, 2023*).

The following datasets were generated:

| Author(s) | Year | Dataset title | Dataset URL | Database and Identifier |
|---|---|---|---|---|
| Rapan L, Froudist-Walsh S, Niu M, Xu T, Zhao L, Funck T, Wang XJ, Amunts K, Palomero-Gallagher N | 2023 | Cytoarchitectonic, receptor distribution and functional connectivity analyses of the macaque frontal lobe | https://balsa.wustl.edu/study/7xGrm | BALSA, 7xGrm |
| Rapan L, Froudist-Walsh S, Niu M, Xu T, Zhao L, Funck T, Wang XJ, Amunts K, Palomero-Gallagher N | 2023 | MEBRAINS Multilevel Macaque Brain Atlas | https://search.kg.ebrains.eu/instances/Project/e39a0407-a98a-480e-9c63-4a2225ddfbe4 | EBRAINS, e39a0407-a98a-480e-9c63-4a2225ddfbe4 |

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
