## [Editor Report]

Rapan et al. report a new multi-modal parcellation of the macaque frontal cortex based on cytoarchitectural division complemented with functional connectivity and neurochemical data. This builds on prior highly influential maps that subdivide the cortex based on anatomical fingerprints, both confirming these prior reports and defining new subdivisions. As such, this is a fundamental contribution with compelling results that can guide future neuroscientific research into the function of the frontal lobes.

---

## [Decision Letter]

**Decision letter after peer review:**

Thank you for submitting your article "Cytoarchitectonic, receptor distribution and functional connectivity analyses of the macaque frontal lobe" for consideration by *eLife*. Your article has been reviewed by 3 peer reviewers, and the evaluation has been overseen by a Reviewing Editor, David Badre, and Chris Baker as the Senior Editor. The following individual involved in the review of your submission has agreed to reveal their identity: Michel Thiebaut de Schotten (Reviewer #3).

The Reviewers were universally positive about the contribution and quality of these findings. However, they did have some points of revision that can help improve the clarity and the impact of the work. I refer you to the specific comments from the reviewers below for details. However, their comments highlight some essential revisions.

1) Reviewer 1 raises the point that connectivity to the temporal lobe is important to include in your analysis of the prefrontal cortex.

2) Reviewer 2 points out that the relationship to the Walker map could be made clearer.

3) All three reviewers highlighted several points to be clarified and tightened both in the introduction and discussion, as well as clarifying some aspects of the design and methods.

*Reviewer #1 (Recommendations for the authors):*

The design of the study is not completely clear. Indeed, the reason for analyzing prefrontal and frontal areas together in the cluster analysis is not explained. In the introduction, the authors state that quantification of heterogeneous receptor distribution throughout the cerebral cortex enables the identification of subdivisions among primary sensory, primary motor, and multimodal areas. However, they analyzed prefrontal data only with those of caudal frontal areas. The authors have also, from previous studies, receptor density data of parietal and cingulate areas, thus they could have also included these areas in the cluster analysis. Furthermore, in the introduction, the authors said that functional connectivity data could "resonate" with those of receptor density, but most of the analyzed receptors reflect the presence of subcortical input, not cortical connectivity. Finally, in the discussion, the results of cluster analysis are interpreted in a way that is not supported by their data ("functional clustering", e.g., oculomotor interpretation of the entire cluster 4).

The introduction could be more focused. Specifically, the issue of homologies between macaque and human cortical areas is introduced but then not addressed, as well as the issue of variability of parcellations. Furthermore, the authors introduce the issue of variability between brains, but from the methods, I understand that the observer-independent cytoarchitectonic analysis was performed on only one subject (DP1) and that the other brains are then qualitatively subdivided as brain DP1.

The number and dimension of the samples analyzed for autoradiography in each brain for each area are not clear. Accordingly, the assessment of variability between samples and brains is not clear. In the observer-independent cytoarchitectonic analysis, the concept of ROIs is not well defined.

The authors excluded parts of areas 14 (14c) and 13 (13a) because of the plane of cut. What about area 8 within the most posterior part of the inferior arcuate sulcus?

I suggest using "area" and "subdivision" more consistently within the text.

*Reviewer #2 (Recommendations for the authors):*

In this impressive study, Rapan and colleagues did perform a multi-modal parcellation of the macaque frontal cortex based on cytoarchitectonic, receptor architecture data in fascicularis macaques and resting-state functional fMRI data in anaesthetized rhesus macaques.

Compared with the classic Walker map of the macaque frontal cortex, the authors produced a more refined map.

Some areas they identified had been already reported in previous studies and atlases (Petrides et al. 2012 Cortex; Reveley et al. 2017 Cerebral Cortex; Saleem and Logothetis 2012), although arguably with different names. How does the proposed parcellation fit with previous ones? More specifically how do the authors consider areas 9/46d and 9/46v?

How does the clustering based on receptor fingerprints correlate with a clustering based on functional connectivity fingerprints (see also Goulas et al. 2017 J. Neurophysiology)? Clusters based on receptors densities appear to be distinct from published connectivity parcellation (Goulas et al. 2017 J. Neurophysiology; Giarracco and Averbeck 2021 J. Neurophysiology; Hutchinson and Everling 2014 Neuroimage). The inputs/outputs of a given brain area constrain the computation it can perform (Passingham and Kotter 2002 Nat Neurosc), could the intrinsic properties of a brain area reflect a distinct constraint on computational properties? For instance, by impacting the temporal signature of neural activity (Fontanier et al. 2022 *eLife*).

If I well understood the fMRI analysis, a chosen vertex was considered a seed for calculating the connectivity fingerprint of a given region. Is it correct?

In their discussion, the authors discussed that the limited functional connectivity of area 10 compared with for instance area 8A could be due to a lower SNR. Classic tracer studies could help the authors strengthen their point. For instance, retrograde tracer injections were performed in both areas 10 and 8A by Markov et al. 2014. The authors could simply compare the number of areas projecting to 10 and 8A.

---

## [Author Response]

Essential revisions:Reviewer #1 (Recommendations for the authors):The design of the study is not completely clear. Indeed, the reason for analyzing prefrontal and frontal areas together in the cluster analysis is not explained. In the introduction, the authors state that quantification of heterogeneous receptor distribution throughout the cerebral cortex enables the identification of subdivisions among primary sensory, primary motor, and multimodal areas. However, they analyzed prefrontal data only with those of caudal frontal areas. The authors have also, from previous studies, receptor density data of parietal and cingulate areas, thus they could have also included these areas in the cluster analysis. Furthermore, in the introduction, the authors said that functional connectivity data could "resonate" with those of receptor density, but most of the analyzed receptors reflect the presence of subcortical input, not cortical connectivity.

We apologize for the lack of clarity. The present study builds on our parcellation of the (pre)motor cortex (Rapan et al., 2021) as part of a project to parcellate and characterize the macaque frontal lobe. Thus, the primary aim of the present study was to characterize the cyto- and receptor architectonic segregation of the macaque prefrontal cortex and to produce a statistically testable parcellation. A second aim was to determine clustering of the identified areas based on similarities in their receptor fingerprints. A further aim was to characterize the functional connectivity of the identified areas, and this analysis revealed, among other things, a tight correlation between posterior prefrontal and premotor areas. Since receptors constitute key elements in signal transduction, we hypothesized that this trend would be reflected in the result of the multivariate analyses if they also included our previously published data on the (pre)motor areas. Thus, our current Figures 19 and 20 present the result of such an analysis, with which we provide a comprehensive overview of the structural and functional relationships between all frontal areas. To clarify our motivation for this study design, we have modified following sentence in the Introduction (lines 136-145):

“The primary aim of the present study was to. First, to identify and characterize prefrontal areas based a quantitative cyto- and receptor architectonic approach, and to create a 3D statistically validated parcellation scheme in stereotaxic space. Since the functional connectivity analysis revealed a tight coupling between posterior prefrontal and premotor areas, and receptors play a key role in signal transduction, we hypothesized that this tight relationship would be associated with similarities in neurochemical composition. Thus, we decided to also include our previously published receptor fingerprints of (pre)motor areas (Rapan et al. 2021) in the multivariate analyses. Importantly, the densities of prefrontal and (pre)motor areas were all obtained from the same brains.”

We are not sure we understood the reviewer’s comment concerning the fact that most of the receptors we analyzed reflect the presence of subcortical input, but not of cortico-cortical connectivity. Although it is true that cholinergic, adrenergic, serotonergic and dopaminergic innervation arises from subcortical structures, glutamate and GABA are ubiquitously produced throughout the brain. I.e., both the superficial cortical layers (source of long-range feedforward cortico-cortical connectivity) and the deeper cortical layers (source of long-range feedback cortico-cortical connectivity) contain glutamate and GABA synthesizing neurons. As now mentioned in the aims, we hypothesize that there is a tight correlation between the neurochemical segregation and functional connectivity pattern of areas of the macaque frontal lobe.

Finally, we agree that it would make sense to perform multivariate analyses which include receptor data from all the areas mapped by our group. However, we would like to remind the reviewer that the present study already encompasses a very large portion of the macaque brain, and provides detailed, multimodal, and quantitative data of a total of 51 frontal areas. Consequently, the discussion of an analysis as described by the reviewer would go outside the scope of the study. However, in the future, once we have finalized mapping the entire macaque brain, we plan to conduct such a global analysis of our multimodal data.

Finally, in the discussion, the results of cluster analysis are interpreted in a way that is not supported by their data ("functional clustering", e.g., oculomotor interpretation of the entire cluster 4).

Reviewer’s comments are correct. In order to clear this, we included functional connectivity data of all premotor and motor areas in Figures 16-18. Our results are in accordance with the previous suggestion that areas within and around spur of the arcuate sulcus belong to the oculomotor brain complex.

With this, we update our text in the Discussion, which now reads (lines 1359-1366):

“Furthermore, the functional connectivity analysis revealed that areas 8Av, F7s, F2v and F4s, which are located within the spur of the arcuate sulcus, have strong connectivity with parietal areas associated with visual responses and control of saccadic and oculomotor movements, e.g., with intraparietal area LIP, and rostral areas Opt and PG of the inferior parietal lobule (Andersen et al., 1990; Niu et al., 2021). In addition, we also found a correlation with polysensory temporal areas STP and TPt, as well as with area MST, which is part of the temporal motion complex region (Boussaoud et al., 1991; Kilintari et al., 2014).”

The introduction could be more focused. Specifically, the issue of homologies between macaque and human cortical areas is introduced but then not addressed, as well as the issue of variability of parcellations.

We thank the reviewer for this constructive suggestion, and have shortened the introduction by not mentioning Brodmann’s map of the human brain and also by removing the sentences concerning the issue of homologies. Regarding the variability of parcellations, we have dedicated an entire chapter in the Discussion to this issue (pages 35-43). Since we are aware that this is a very complex problem, we have created an additional table (Table 1) in which we provide a comparison of the various existing parcellations with that of Rapan.

Furthermore, the authors introduce the issue of variability between brains, but from the methods, I understand that the observer-independent cytoarchitectonic analysis was performed on only one subject (DP1) and that the other brains are then qualitatively subdivided as brain DP1.The number and dimension of the samples analyzed for autoradiography in each brain for each area are not clear. Accordingly, the assessment of variability between samples and brains is not clear. In the observer-independent cytoarchitectonic analysis, the concept of ROIs is not well defined.

We found only minor differences in macroanatomy, which were restricted mainly to the presence of dimples and of the spur of the arcuate sulcus, but this does not influence the cyto- or receptor architectonic features. We used the quantitative cytoarchitectonic approach on the DP1 brain, and also on the sections from brains 11530, 11539 and 11543 which were processed for the visualization of cell bodies. When a given receptor type revealed a border, it coincided with the position of the cytoarchitectonically identified borders at that location. However, the most exhaustive quantitative cytoarchitectonic analysis was performed in the DP1 brain, since we could use between three and six sections (depending on the size of the area) to define borders for each area. This was not always possible in the receptor brains, since they were processed for more modalities and thus had less silver-stained sections per cortical area. Interindividual variability in receptor densities is given by the standard deviation (s.d.) values which we provide in Table 3.

Concerning the number of samples analyzed for autoradiography, we now provide this information in the Material and Methods section (lines 1586-1591):

“Thus, for each area (with the exception of areas 13m, 13l, 14c and 13a) and receptor type, we extracted profiles from three consecutive sections in each of the four hemispheres examined. Due to technical problems, we were only able to obtain this data for areas 13m and 13l from two hemispheres (11530 and 11539_R), and we could not measure receptor densities in areas 14c or 13a.”

In order to clarify the concept of the ROI used in the quantitative cytoarchitectonic method, the following sentences have been added in the manuscript text (lines 1478-1487):

“For each identified area, GLI images were generated from three neighbouring sections in the rostro-caudal direction, and regions of interest (ROI) were defined around each portion of the cortical ribbon where a border had been identified by visual inspection by manually drawing an outer (at the interface between layers I and II) and an inner (at the border between layer VI and the white matter) contour. These contour lines were used to define equidistant traverses running perpendicularly to the cortical surface, along which the changes in grey values quantify the laminar pattern characteristic of a cortical area (Schleicher et al. 2009) and are measured as GLI-profiles (for details see Palomero-Gallagher and Zilles 2019; Zilles et al. 2002).”

The authors excluded parts of areas 14 (14c) and 13 (13a) because of the plane of cut. What about area 8 within the most posterior part of the inferior arcuate sulcus?

Unfortunately, we had a similar problem when it comes to extracting densities from the most posterior part of the inferior arcuate sulcus (*ias*) due to the plane of sectioning. Based on simple visual inspection of both the cytoarchitecture and receptor architecture of cortex in the most posterior part of the inferior arcuate sulcus, we believe that it contains area 8Av. To demonstrate what exactly this means, we created an exemplary Author response image 1. Series of NMDA section are shown, more rostrally, areas 14c and 13a are located, whereas area 8A is positioned more posteriorly. Arrows indicate questionable parts of cortex, where cortical surface is stretched without clear laminar structure (as in the case of areas 14c and 13a) or hidden beneath the transitional cortical surface (as seen in the posterior portion of *ias*).

**Author response image 1. sa2fig1:** 

I suggest using "area" and "subdivision" more consistently within the text.

We have identified 35 “entities” with distinct cytoarchitecture, receptor architecture and functional connectivity patters, and we consider each of these “entities” to be a cortical area. When we refer to ‘subdivision’ it is in the context that several of our areas can be encompassed by a single area as defined by previous mapping studies. E.g., our areas 9m, 9d and 9l are subdivisions of Walker’s area 9. We have gone through the entire manuscript to ensure that our use of the words “area” and “subdivision” comply with this definition.

Reviewer #2 (Recommendations for the authors):In this impressive study, Rapan and colleagues did perform a multi-modal parcellation of the macaque frontal cortex based on cytoarchitectonic, receptor architecture data in fascicularis macaques and resting-state functional fMRI data in anaesthetized rhesus macaques.Compared with the classic Walker map of the macaque frontal cortex, the authors produced a more refined map.Some areas they identified had been already reported in previous studies and atlases (Petrides et al. 2012 Cortex; Reveley et al. 2017 Cerebral Cortex; Saleem and Logothetis 2012), although arguably with different names. How does the proposed parcellation fit with previous ones? More specifically how do the authors consider areas 9/46d and 9/46v?

We value the reviewer’s appreciation of our work. As mentioned above, we dedicate an entire chapter in the Discussion (pages 35-43) to discuss our findings in the framework of previously published maps. We have now modified the Introduction to include work by Reveley et al. (2017) and Saleem and Logothetis (2012). Furthermore, in the revised manuscript we have created an additional table (Table 1) in which we specify the topographical relationship between our areas and those identified by Walker (1940), Petrides and Pandya (1994, 2002), Carmichael and Price (1994), Preuss and Goldman-Rakic (1991), Morecraft et al. (2012) and Caminiti et al. (2017).

Regarding areas 9/46d and 9/46v, the former corresponds in location and extent with our areas p46d and p46df, whereas 9/46v would be comparable to our areas p46v and p46vf.

How does the clustering based on receptor fingerprints correlate with a clustering based on functional connectivity fingerprints (see also Goulas et al. 2017 J. Neurophysiology)? Clusters based on receptors densities appear to be distinct from published connectivity parcellation (Goulas et al. 2017 J. Neurophysiology; Giarracco and Averbeck 2021 J. Neurophysiology; Hutchinson and Everling 2014 Neuroimage).

We performed a clustering analysis based only on the receptor fingerprints, not on the functional connectivity fingerprints. However, the result of the functional connectivity analysis was discussed in the framework of the clusters identified by the multivariate analysis of receptor densities. Thus, we only discuss whether a certain trend can be noted by both methods simultaneously. Our motivation for including functional connectivity was to facilitate the use of gold-standard anatomical data by specialists in in-vivo imaging, and facilitate the understanding of structural and functional organization. The connectivity data also allowed us to differentiate some areas with similar receptor profiles, such as between the newly identified subdivisions of area 10. This provides further support to the separation of these areas based on cytoarchitecture. In order to clarify this point, we have changed the title of the corresponding section in the discussion. It now reads:

“4.2 Receptor-driven clustering of macaque frontal areas is associated with distinct functional connectivity patterns”.

It is very difficult to compare the result of our functional connectivity analysis with the results of Goulas et al. (Goulas et al., 2017), because we have a much more detailed parcellation of the prefrontal cortex, and also include our previously published premotor and motor areas (Rapan et al. 2021). In contrast to our present study, Goulas and colleagues did not include the entire frontal lobe in their analysis. First, their lateral frontal cortical mask did not extend into the central sulcus, which means that our area 4p was not part of their analysis. Furthermore, Goulas et al. (2017) wrote in the Results section that “The results from the clustering in the principal sulcus were not interpretable in terms of prior parcellation schemes and therefore not satisfactory”. Therefore, they did not include the clusters from this region in their subsequent analyses. Interestingly, their Figure 3 reveals a rostro-caudal subdivision of cortex within and around the principal sulcus, which is in agreement with our definition of anterior and posterior subdivisions of Walker’s area 46. Furthermore, cortex in the fundus of the sulcus and on the more superficial part of the sulcal walls are located in different clusters, which could reflect our medio-lateral subdivisions of Walker’s area 46. Thus, our areas a46df, a46vf, p46df and p46vf are found at a location occupied by cluster C10 of Goulas et al. (2017).

It is also problematic to directly compare the result of our functional connectivity analysis with the study by Giarrocco and Averbeck (Giarrocco and Averbeck, 2021) because they created a connectivity matrix by examining the primary literature of anatomical tract-tracing studies, which do not use the parcellation scheme described in this study. E.g., Giarrocco and Averbeck (2021) describe the connectivity of area F5, but within this portion of cortex, we identify areas F5s, F5d and F5v. And we found that F5s has a much wider functional connectivity pattern than do F5d or F5v. Thus, it is difficult to quantify the true anatomical connectivity of the newly described regions, without a laborious and time-consuming analysis, which would go beyond the scope of this study. Furthermore, we did not include cingulate nor agranular insular areas in the present analysis.

We find that comparison with Hutchison and Everling (Hutchison and Everling, 2014) is not entirely compatible since they only include prefrontal region. However, they analysed the intrinsic connectivity of voxels located rostral to the genu of the arcuate sulcus, and identify 5 clusters which they label as “Rostral”, “Ventral”, “Lateral”, “Caudal” and “Dorsal” (Figure 6 in Hutchinson and Everling (2014)). They mention in their discussion that a cut-off at a Euclidean distance between 184-193 would result in the merging of the rostral and ventral as well as of the lateral and caudal clusters. This tripartition would be in agreement with our observation that the areas we identified within the “Rostral-Ventral” cluster of Hutchinson and Everling (2014) have the most restricted functional connectivity pattern, whereas we found the most widespread functional connectivity pattern across the brain for areas we defined within cortex covered by their “Lateral-Caudal” cluster.

The inputs/outputs of a given brain area constrain the computation it can perform (Passingham and Kotter 2002 Nat Neurosc), could the intrinsic properties of a brain area reflect a distinct constraint on computational properties? For instance, by impacting the temporal signature of neural activity (Fontanier et al. 2022 eLife).

We thank the reviewer for this insightful comment, which has now led us to include the following text at the beginning of the part of the Discussion addressing the comparison of our functional connectivity and multivariate clustering analyses of receptor densities (lines 1185-1206):

“Although functional connectivity often indicates direct anatomical connections (Greicius et al., 2009; Thiebaut de Schotten et al., 2011), it also reflects indirect connections, as well as an input from a common source area (Smith et al., 2013). Moreover, such analysis may be affected by differences in local recurrent activity across areas (Chaudhuri et al., 2015). It is important to understand that while structural and functional aspects of brain organization are genuinely interconnected, they are not equal (Rapan et al. 2021). Contrary to the tract-tracing approach, functional connectivity can be easily assessed for novel parcellations of cortex, as shown in a present study, since it enables differentiation among areas with similar receptor profiles (e.g., newly identified subdivisions of area 10). Concerning neurotransmitters and their receptors, which constitute the molecular underpinning of signal transduction, we here analysed receptors with different mechanisms of action (ionotropic/metabotropic) and outcomes (excitatory/inhibitory). Activation of metabotropic receptors results in slower, longer lasting, and more widespread changes in membrane potential than does activation of ionotropic receptors. Therefore, if two areas differ in the relative balance of ionotropic vs metabotropic receptors, this will indeed result in different constraints on computational properties and could influence the temporal signature of neural activity. Taken together, functional connectivity facilitates the use of gold-standard anatomical data (e.g., the cytoarchitectonic boundaries and receptor data described here) by specialist in neuroimaging and enables a more systematic understanding of the macaque frontal cortex.”

Additionally, in Table 5 we now include, for each receptor type analysed, a column providing information on their mechanism of action and outcome of their activation.

If I well understood the fMRI analysis, a chosen vertex was considered a seed for calculating the connectivity fingerprint of a given region. Is it correct?

No, we did not use vertexes as seed points for the computation of the connectivity fingerprint of a specific area. The seed was an entire cortical area, and we used the 1^st^ principal component of the BOLD-signal across all vertices within that area as the seed for connectivity throughout the cortex.

In their discussion, the authors discussed that the limited functional connectivity of area 10 compared with for instance area 8A could be due to a lower SNR. Classic tracer studies could help the authors strengthen their point. For instance, retrograde tracer injections were performed in both areas 10 and 8A by Markov et al. 2014. The authors could simply compare the number of areas projecting to 10 and 8A.

We thank the reviewer for this helpful suggestion. Careful inspection of Figure 2 of Markov et al. (2014) reveals that the injection location of area 10 mostly corresponds to our dorsal subdivisions 10md and 10d. We have included the following text in the revised Discussion (lines 901-903):

“Thus, when available, we discuss the results of our functional connectivity analysis in the framework of tracer studies with injection sites within our region of interest (e.g., Markov et al. 2014; Carmichael and Price, 1996).”

(lines 936-944):

“Comparison between the tracer study by Markov et al. (2014) and our functional connectivity analysis revealed certain similarities regarding connectivity of area 10. Careful inspection of their Figure 2 reveals that their injection sites are at a location comparable mainly to that of our area 10md and, to a lesser extent, of our area 10d. They describe connectivity with prefrontal areas 14, 9, 46d, 46v and 9/46d as well as with cingulate areas 25, 32 and 24c (Markov et al., 2014), which is in accordance with our results for areas 10md, whereas our area 10d presents a more restricted functional connectivity than does 10md, since it is not correlated with the cingulate cortex.”

Also, what is also visible from Markov et al. (2014), is that areas 8m and 8l (location of those injections corresponding to our area 8Ad) have more areas across the brain projecting to them than has area 10.